# Particulate reshapes surface jet dynamics induced by a cavitation bubble

Xianggang Cheng [1], Xiao-Peng Chen [1,2] ✉, Zhi-Ming Yuan [3] &
Laibing Jia [3] ✉

Liquid jet formations on water surfaces serve as a cornerstone in diverse scientific disciplines, underpinning processes in climatology, environmental science, and human health issues. Traditional models predominantly focus on pristine conditions, an idealisation that overlooks common environmental irregularities such as the presence of particulate matter on water surfaces. To address this shortfall, our research examines the dynamic interactions between surface particulate matter and cavitation bubbles using floating spheres and spark bubbles. We unveil five novel jet modes, advancing beyond classical models and demonstrating enhanced variability in jet dynamics. We observe that particulates significantly lower the energy threshold for jet formation, showing the enhanced sensitivity of jet dynamics to their presence. The phase diagram and analyses illustrate how the interplay between the dimensionless immersion time of the particulate and the spark bubble's dimensionless depth influences jet mode development, from singular streams to complex cavity forms. These insights not only advance our understanding of jet formation, but also unlock the potential for refined jet manipulation across a broad range of physical, environmental, and medical applications.

Liquid jets, masses of liquid propelled into streams, play critical roles in a diverse array of natural phenomena and industrial applications, and have been extensively studied[1–5]. Liquid jets influence natural processes such as cloud formation[6,7] and aerosol generation[8,9]. They are also crucial in inkjet printing technologies[10–12] and medical aerosol drug delivery systems[13,14], showing their significance in a broad range of climatological and environmental sciences, advanced manufacturing, and human health.

Numerous studies have explored how jets are formed from clean water surfaces, particularly in the context of bubble bursting and splashing phenomena near a free surface[15–18]. Previous studies have examined the contributions of bubbles in jet formations, considering finite-time singularities, inertial flow focusing, and Rayleigh-Taylor instability[19–21]. Despite this, a comprehensive understanding of liquid jets in particulate-rich environments[22–25], a scenario commonly encountered in real-world settings, remains limited.

Recently, liquid jets from contaminated water surfaces have been recognised as contributing to global human health issues related to the dispersal of pollutants and pathogens[26,27]. A few researchers have explored the impact of nano- and micro-particles mimicking ocean pollutant particles such as microplastics, bacteria, and viruses on bursting bubbles[28–30]. They focused on the particles' transport and accumulation at the jet tip and in the subsequently ejected airborne droplets. Nonetheless, these investigations often overlook the dynamic and sensitive interplay between particulate matter and jet formations. These studies operate under the assumption that nano- or micro-scale particles have minimal influence on jet dynamics. However, this assumption may not be applicable to particles larger than hundreds of microns. Unlike smaller particles, these larger particles substantially contribute to surface defects and create unique interactions within these defects, potentially leading to a profound impact on jet dynamics. Thus, the lack of attention to larger particles in current models[16,18]

[1]School of Marine Science and Technology, Northwestern Polytechnical University, Xi'an, China. [2]Research & Development Institute of Northwestern Polytechnical University in Shenzhen, Shenzhen, China. [3]Department of Naval Architecture, Ocean & Marine Engineering, University of Strathclyde, Glasgow, UK.
✉e-mail: xchen76@nwpu.edu.cn; l.jia@strath.ac.uk

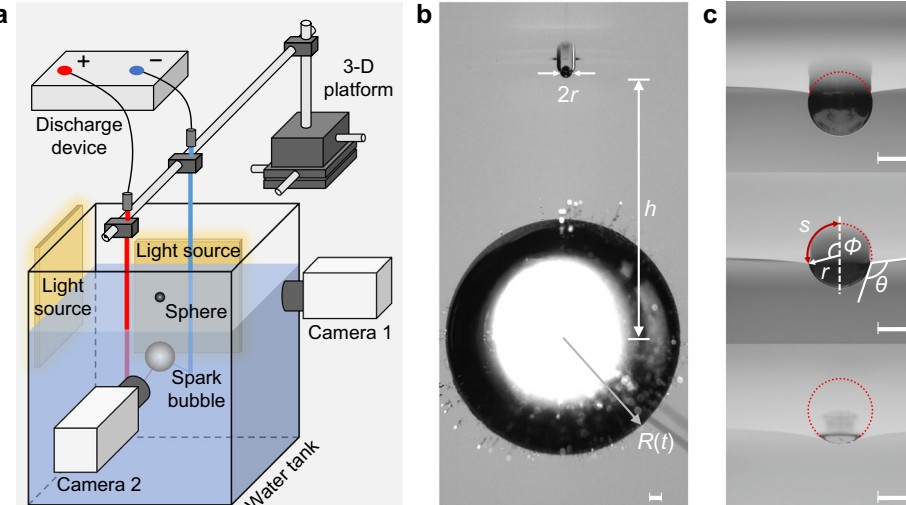

**Fig. 1 | Experimental setup. a** A sketch of the experimental setup. The spark bubble was generated by a low-voltage discharge method, and a surface defect was created by floating a sphere on the water surface. Two high-speed cameras were used to capture the evolution of the water surface and the underwater spark bubble, respectively. **b** A solid sphere with a radius of $r$ creates a defect when floated on the water surface. The underwater spark bubble is at a distance $h$ below the initial water line, with its instantaneous radius $R(t)$. The photograph captures the moment ($t = \tau$) as the bubble reaches its maximum radius $R_m$. The interaction causes an upward deformation of the water surface, forming a pit over the sphere. **c** Underwater close-up of three copper spheres ($r = 500$ μm) resting on still water with varying contact angles ($\theta = 81°$, $108°$, and $154°$). The red dotted lines outline their emerged parts above the water surface. The azimuthal angle of the contact line, $\phi$, and the arc length of the unimmersed portion, $s$, are also indicated. All scale bars represent 1 mm.

represents a notable gap in our understanding of jet dynamics, especially under realistic and often imperfect environmental conditions.

Existing studies have investigated underwater particles driven by a cavitation bubble[31,32]. These studies found that the driving force originates from either liquid inertia or bubble contact on one side of the particle. The studies provide valuable insights for our research into how cavitation bubbles interact with particles on the water surface.

To address the impact of larger floating particulates on surface jet dynamics, we designed an approach using solid spheres to create surface imperfections and underwater spark cavitation bubbles to drive jet formations (see Fig. 1a). A systematic study was conducted using spheres with varied properties and spark bubbles positioned at different depths. In some experiments, the spark bubble directly contacted the sphere, introducing a distinct driving mechanism different from those governed by liquid inertia[32]. These cases are therefore excluded from the analysis in this study.

Here, we find that the jet formation is reshaped by the presence of particulates on the water surface. The particulate matter increases the variability in jet dynamics. It also significantly lowers the energy requirement for jet formation, highlighting the system's sensitivity to particulate presence. By modelling the reaction of the water surface and

sphere to the spark bubble's impact using two key dimensionless numbers, we capture the fluid's adaptive response to particulate disturbances. Our findings have implications for refined jet manipulation across a broad range of applications, including aerosol production, pollution management, and the control of pathogen transmission.

## Results

### Characteristics of the five jet modes

We use underwater spark bubbles to drive jet formations on the water surface and employ floating solid spheres to create surface defects (Fig. 1a and 'Methods'). Figure 1b outlines the key geometric relationships in the experimental setup. A sphere, serving as a model for particulate matter, is placed on the water surface to create a surface defect. In our experiments, the sphere's radius $r$ ranges from 100 to 2500 μm. Their physical parameters are detailed in Table 1.

Before the spark bubble ignites, the sphere remains afloat on the water surface due to the balance among gravity, surface tension, and buoyancy forces[22,33,34], which can be described by,

$$6 \sin\phi \sin(\phi - \theta) - (\cos^3\phi - 3\cos\phi - 2)(\rho - 1)Bo = 0. \quad (1)$$

Here, $\theta$ is the contact angle of the sphere with water, $\rho$ is the density ratio between the sphere and water, $\rho = \rho_s/\rho_l$. $Bo$ is the Bond number, $Bo = \rho_l r^2 g/\sigma$, where $\sigma$ is the water's surface tension and $g$ is the gravitational acceleration. $\phi$ is the azimuthal angle of the contact line, implicitly given by $\phi = \phi(Bo, \rho, \theta)$. Figure 1c shows how $\phi$ and the corresponding arc length of the unimmersed portion, $s$, vary with $\theta$.

In this study, the maximum radius of the spark bubble, $R_m$, remains fixed at $9.2 \pm 0.5$ mm, where the value of $\pm$ denotes the standard deviation of the mean. Using $R_m$ as the characteristic length, the dimensionless depth of the spark bubble is,

$$\hat{h} \equiv \frac{h}{R_m}, \quad (2)$$

where $h$ denotes the initial depth of the spark bubble centre. Additionally, four critical moments during the oscillations of a spark bubble were defined:

## Table 1 | Physical properties of spheres

| Material | $\rho$ | $\theta$ (°) | | |
|---|---|---|---|---|
| | | hydrophilic | hydrophobic | super-hydrophobic |
| H62 copper | 8.5 | 81.2 ± 3.7 | 108.8 ± 3.4 | 155.3 ± 9.9 |
| 304 stainless steel | 7.9 | - | 111.1 ± 2.2 | 153.4 ± 3.9 |
| ZrO₂ | 5.9 | - | 106.2 ± 1.4 | 152.7 ± 5.4 |
| TC4 Ti | 4.4 | 80.8 ± 1.4 | 111.2 ± 1.8 | 150.8 ± 8.3 |
| 1060 Al | 2.7 | - | 108.7 ± 1.3 | 155.8 ± 6.8 |
| SiO₂ | 2.5 | - | 105.0 ± 2.3 | 156.6 ± 1.0 |
| POM | 1.4 | 82.1 ± 6.0 | - | 150.5 ± 4.2 |

$\rho$ is the density ratio between sphere and water, and $\theta$ denotes contact angle. The values of ± represent the standard deviation.

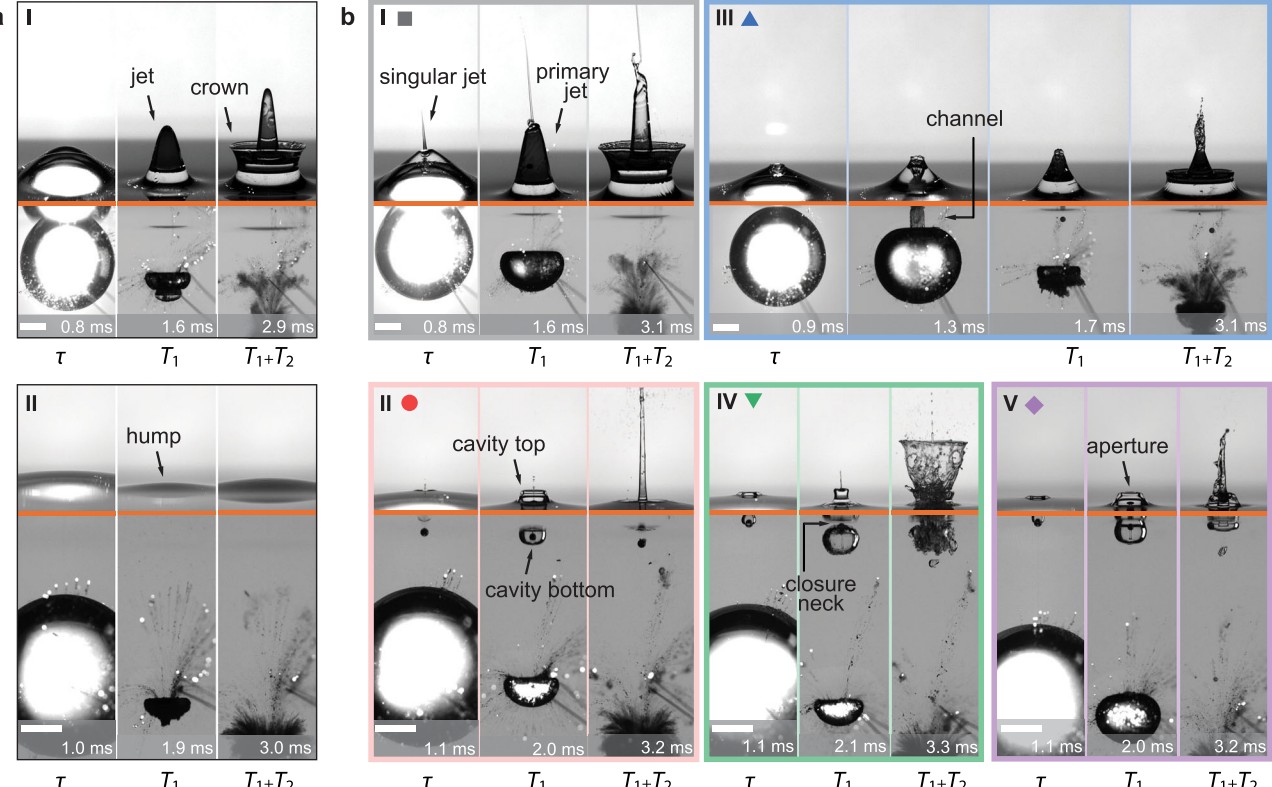

**Fig. 2 | The snapshots for different jet modes.** $\tau$ is the duration from the spark bubble's inception to its maximum radius, $T_1$ and $T_2$ are the periods of the spark bubble's first and second oscillation, respectively. The orange lines separate the views above and below the waterline. All scale bars represent 5 mm. **a** Reference group showing the deformation of flat water surfaces. I: Jets development from a flat water surface ($\hat{h} = 0.82$). II: A hump development from a flat water surface ($\hat{h} = 1.57$). **b** Five jet modes with spheres setting on water surfaces. The spheres radii, $r = 500$ μm. I: Tiered Jet Mode ($\hat{h} = 0.81$, $\rho = 4.4$, $\theta = 80.8°$). II: Jet Cavity Mode ($\hat{h} = 1.62$, $\rho = 4.4$, $\theta = 80.8°$). III: Cavity Venting Mode ($\hat{h} = 1.00$, $\rho = 1.4$, $\theta = 82.1°$). The second snapshot shows a channel formed between spark bubble and atmosphere. IV: Sealed Cavity Mode ($\hat{h} = 1.93$, $\rho = 4.4$, $\theta = 111.2°$). V: Open Cavity Mode ($\hat{h} = 2.25$, $\rho = 4.4$, $\theta = 111.2°$). Coloured symbols indicate the five jet modes. Supplementary Figs. 1–5 and Supplementary Movies 1–5 provide further details, showcasing the five jet modes alongside their corresponding particle-free reference cases.

- $t = 0$: The moment when the spark bubble is ignited.
- $t = \tau$: The moment when the spark bubble reaches $R_m$.
- $t = T_1$: The moment marking the end of the spark bubble's first oscillation.
- $t = T_1 + T_2$: The moment marking the end of the spark bubble's second oscillation.

We selected $\tau$ as the characteristic time of the spark bubble. According to the Rayleigh-Plesset equation for spherical bubbles[35], $\tau \propto R_m$, and $T_1 = 2\tau$. For a fixed $R_m$, both $\tau$ and $T_1$ remain constant. The presence of the water surface accelerates the collapse of the spark bubble[17]. In our experiments, $T_1 = (1.9 \pm 0.1)\tau$.

With a sphere floating on the water surface, we observed diverse and distinct jetting phenomena compared to those from flat water surfaces (Fig. 2). Figure 2a serves as a reference, which depicts jet formations from flat surfaces. In scenarios with a flat water surface, underwater spark bubbles induce dynamic pressure variations, leading to distinct water surface responses at various values of $\hat{h}$[17,18]. Experiments show that within $0.4 \le \hat{h} \le 1.2$, the water surface deforms outward, forming a pronounced jet at $T_1$ (Fig. 2a-I). The bubble rebounds after $T_1$ and a crown forms from the base of the jet. When $\hat{h} > 1.2$, the water surface rises slightly (Fig. 2a-II). A hump forms without significant jet generation. For $\hat{h} < 0.4$, the bubble breaks the surface, leading to water splashing[18], a phenomenon not considered in this study.

We identified five distinct jet modes resulting from the interactions between floating spheres and underwater spark bubbles (Fig. 2b). These modes are named: Tiered Jet (I), Jet Cavity (II), Cavity Venting

(III), Sealed Cavity (IV), and Open Cavity (V). The names are based on the features of jets showing in the second snapshots of each sub-figure in Fig. 2b.

In Mode I, a rapid singular jet and an underlying water bulge form as the spark bubble expands ($t = \tau$). A tiered jet structure is observed with a fine singular jet[19] above the primary jet ($t = T_1$). After the spark bubble rebounds, a crown forms, similar to scenarios without spheres as shown in Fig. 2a-I.

In Mode II, a concavity develops at $t = \tau$, surrounding a singular jet at its centre. The concavity enlarges into a cavity as the spark bubble collapses, then it extends to the sphere at $T_1$. This configuration, where the sphere is surrounded by an open cavity with a singular jet above it, is defined as the Jet Cavity Mode. Upon the bubble's rebound, the open cavity collapses, ejecting a focused primary jet.

In Mode III, the concavity on the water surface expands as the spark bubble collapses. Its bottom reaches the upper wall of the spark bubble, then forms a channel that bridges the bubble with the atmosphere. This channel facilitates aerodynamic interaction and creates a ventilation effect. The channel then pinches off, trapping some air in the primary jet and fragmenting it into tiny bubbles ($t = T_1$). After the spark bubble rebounds, a crown forms from the base of the primary jet. The main feature of Mode III is the presence of the channel and the corresponding cavity venting phenomenon. In Mode III, cases both with and without singular jets were observed before $t = \tau$. These cases are defined as Mode III.a (with singular jets) and Mode III.b (without singular jets). Fig. 2b-III illustrates a Mode III.b case.

In Modes IV and V, no singular jet forms before $t = \tau$. In Mode IV, as the spark bubble collapses, the expanding cavity seals at its rim. An

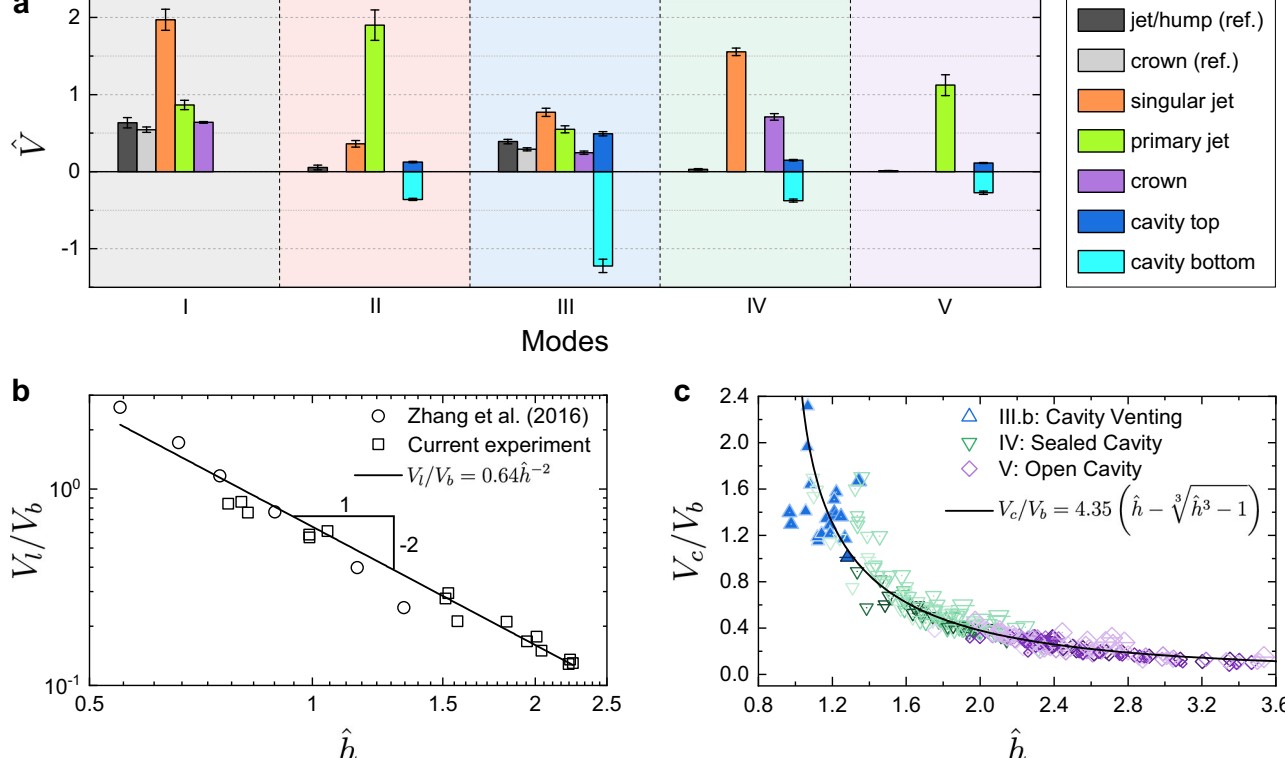

**Fig. 3 | Velocities of jets and cavities in five jet modes. a** Dimensionless velocities of featured geometries during jet formation, including the tips of singular and primary jets, the up edge of crowns, and the cavities' top and bottom. For reference, the velocities of jet/hump from flat water surfaces at the same $\hat{h}$ are also included. Velocities are normalised by the characteristic velocity of the spark bubble ($V_b = R_m/\tau \approx 10.8$ m/s). The velocities and their standard deviations are calculated from the position measurements corresponding to the cases presented in Fig. 2b. The corresponding experimental parameters are as follows: spheres radii, $r = 500$ μm. I: Tiered Jet ($\hat{h} = 0.82 \pm 0.01$, $\rho = 4.4$, $\theta = 80.8°$). II: Jet Cavity ($\hat{h} = 1.57 \pm 0.05$, $\rho = 4.4$, $\theta = 80.8°$). III: Cavity Venting ($\hat{h} = 1.03 \pm 0.04$, $\rho = 1.4$, $\theta = 82.1°$). IV: Sealed Cavity ($\hat{h} = 1.92 \pm 0.07$, $\rho = 4.4$, $\theta = 111.2°$). V: Open Cavity ($\hat{h} = 2.25 \pm 0.04$, $\rho = 4.4$, $\theta = 111.2°$). **b** The dimensionless water surface velocity on flat water surfaces, $V_l/V_b$, follows a power-law relationship with $\hat{h}$, $V_l/V_b \sim \hat{h}^{-2}$. **c** The dimensionless cavity expansion velocity, $V_c/V_b$, versus $\hat{h}$. The solid line represents theoretical predictions. Source data are provided as a Source Data file.

underwater air bubble is trapped around the sphere. The closure neck is marked in Fig. 2b-IV. A water column and a singular jet above it are formed as the cavity seals. Following the spark bubble rebounds, a crown forms from the base of the water column, and the underwater air bubble oscillates strongly. In Mode V, the cavity remains open at $t = T_1$. Its aperture is marked in Fig. 2b-V. Similar to Mode II, the open cavity collapses and generates a focused primary jet as the spark bubble rebounds.

The structures of the five jet modes in Fig. 2b represent significant departures from the reference jet profiles observed in particle-free cases in Fig. 2a. The results highlight the enhanced variability in liquid jet dynamics due to particulate interactions. Supplementary Figs. 1–5 and Supplementary Movies 1–5 provide further details, showcasing the five jet modes alongside their corresponding particle-free cases for reference.

### Kinematics of jets and cavities

We measured the velocities of the jet and cavity geometries in the cases shown in Fig. 2b. For comparison, results from jet or hump formation on flat surfaces at the same $\hat{h}$ are also presented. The trajectories of the singular jet, primary jet, and crown appear approximately linear (Supplementary Figs. 1–5). We therefore applied linear fits to these trajectories to determine corresponding velocities. The velocity of humps is averaged over the interval $t = 0$ to $T_1$. Velocities of the cavity top and bottom are determined by performing linear fits to their trajectories from $t = \tau$ to $T_1$. All velocities are normalised by the characteristic velocity $V_b$ ($V_b = R_m/\tau \approx 10.8$ m/s, see 'Methods').

The velocities of these features are illustrated in Fig. 3a. In Mode I, jet kinematics closely resemble those of the corresponding reference case, but with the addition of a fast singular jet unique to this mode. Mode II features a high-speed primary jet. This jet originates from the collapsing cavity bottom after $T_1$ (Fig. 2b-II and Supplementary Fig. 2). Its high velocity results from flow-focusing at the curved interface, where kinetic energy is focused[20,36,37]. In Mode III, the cavity bottom exhibits the highest speed among all cases shown in Fig. 3a. This rapid movement leads to the formation of a channel between the spark bubble and the atmosphere. Mode IV shows both a fast singular jet and a crown. In contrast, the corresponding reference case shows only a hump without any jet ejection. In Mode V, a primary jet forms similarly to Mode II, but no singular jet is present.

In addition, we measured the average water surface velocity $V_l$ on flat surfaces from $t = 0$ to $\tau$, and the characteristic cavity expansion velocity $V_c$, defined as the slope of a linear regression fitted to the cavity bottom position $z(t)$ over the interval $\tau \leq t \leq T_1$, across different values of $\hat{h}$ (Fig. 3b, c). The results show that both $V_l$ and $V_c$ are primarily governed by $\hat{h}$.

Figure 3b shows that the normalised water surface velocity, $V_l/V_b$, follows a power-law relationship with $\hat{h}$. Assuming the spark bubble expands spherically, $V_l$ can be estimated through $V_l h^2 \sim V_b R_m^2$, which yields $V_l \sim \hat{h}^{-2} V_b$. The theoretical predictions align well with the experimental data, with a best-fitting prefactor of 0.64.

To estimate $V_c$, we analyse the physical process during the spark bubble's collapse. The spark bubble is centred at a depth $h$. At time $t = \tau$, the bubble reaches its maximum radius $R_m$. We consider a spherical shell of water extending from $R_m$ to $h$ at the water surface. As the

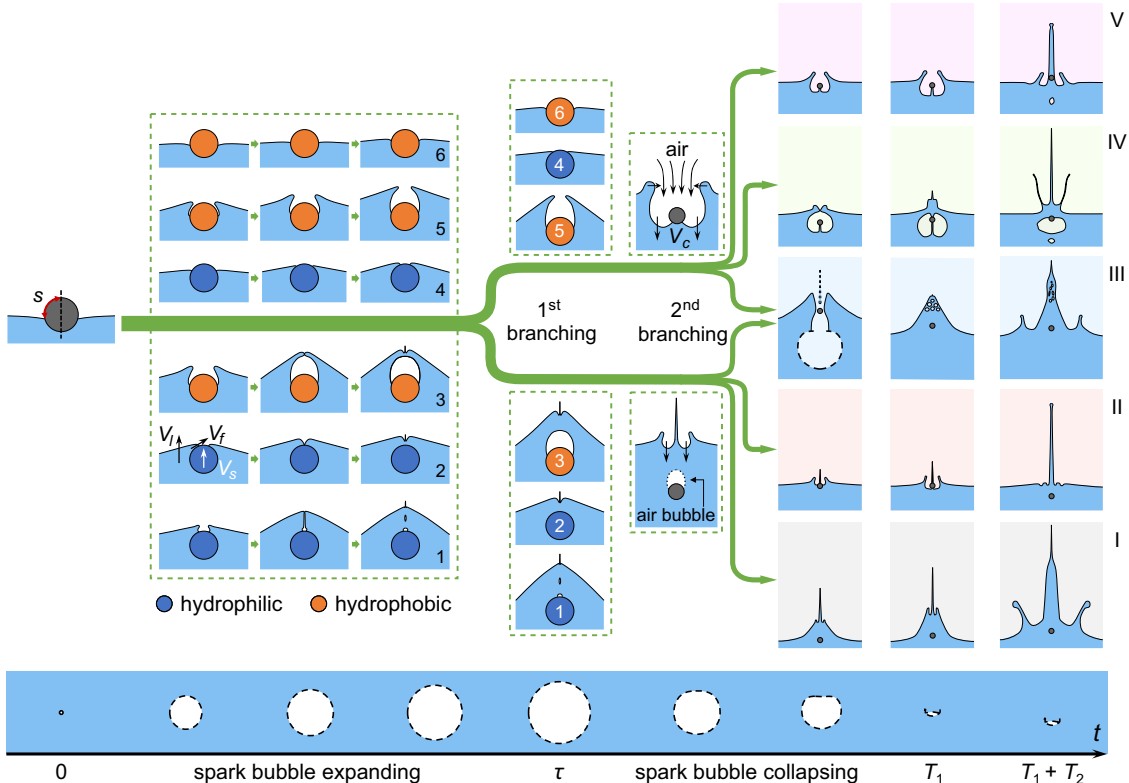

**Fig. 4 | Schematic of the temporal evolution for five jet modes.** The bottom row with a light blue background illustrates the evolution of the spark bubble over its first two oscillation periods, with dashed lines outlining the spark bubble. The upper plots depict the branching processes from a sphere resting on the water surface to five distinct jet modes. Circles indicate the spheres, and solid lines represent the air-water interfaces. The first branching process occurs during the spark bubble's expansion phase, determined by whether a singular jet appears at $t = \tau$. The sketches enclosed by dashed boxes illustrate the potential immersion processes for spheres with hydrophilic (blue circles) or hydrophobic (orange circles) surfaces during this phase. As the spark bubble collapses, the second branching process arises, resulting in five distinct jet modes observed at $t = T_1$, which then evolve into more complex interface phenomena at $T_1 + T_2$.

spark bubble collapses over the duration $\tau_{coll}$ (approximated as $\tau$[17,18,35]), the cavity begins to expand downward at a velocity $V_c$ from the water surface to a depth of $V_c \tau_{coll}$. This collapsing causes the spherical shell to contract, reducing its inner radius from $R_m$ to zero, and outer radius from $h$ to $h - V_c \tau_{coll}$. With the conservation of water volume within this spherical shell, we establish the relationship: $h^3 - R_m^3 \sim (h - V_c \tau_{coll})^3$, which yields $V_c \sim (\hat{h} - \sqrt[3]{\hat{h}^3 - 1})V_b$. As shown in Fig. 3c, the theoretical predictions closely match the experimental data, with a pre-factor of 4.35.

## Two branching processes of jet modes

Figure 4 illustrates five distinct jet modes that evolve from a sphere initially resting on the water surface, governed by two sequential branching processes. The first branching process occurs during the spark bubble's expansion phase ($0 \leq t \leq \tau$), according to whether or not a singular jet forms. As the spark bubble collapses, the second branching process occurs ($\tau \leq t \leq T_1$), with the cavity expanding and evolving further into five jet modes.

The first branching process involves the interaction between the partially immersed floating sphere and the upward surface jet driven by spark bubble expansion. This interaction creates a rising liquid layer that climbs along/near the sphere (Supplementary Fig. 6). If the rising liquid layer closes over the sphere's top before $t = \tau$, the sphere becomes fully immersed; otherwise, it remains partially immersed at this stage.

The physical scenario differs from the classical water-entry problems, where the initial contact between the sphere and the liquid surface induces strong radial spreading of a splashing liquid layer, and

the sphere's immersion primarily results from cavity pinch-off following the sphere's penetration into the liquid[38-40].

To determine the physical conditions in the present system under which the liquid layer encloses the sphere or remains open, we now turn to the impact dynamics during the first branching process. As the spark bubble expands, the water surface rises at a characteristic velocity $V_l$. The surface jet impacts and wets the initially floating sphere, which accelerates to a velocity $V_s$. If the relative speed between the water surface and the sphere, $v = V_l - V_s$, is below a critical velocity $U^*$, the liquid layer climbs along the sphere to its north pole. If $v$ exceeds $U^*$, the liquid layer becomes unstable and detaches from the sphere, resulting in wetting failure[39,41,42]. $U^*$ depends on the surface's wettability[41] (Methods).

Various immersion regimes will appear for spheres with different wettabilities. For hydrophilic spheres, three wetting regimes can occur. In the first case, $v > U^*$, a cavity forms above the sphere. The rim of the liquid layer eventually converges above the sphere and ejects a singular jet, leaving air bubbles between the jet root and the sphere[43] (Fig. 4-1). In the second case, $v < U^*$, the liquid layer climbs along the sphere[41]. The layer converges at the north pole of the sphere and ejects a singular jet before $t = \tau$ (Fig. 4-2). The third case occurs when $v$ is low enough that the wetting liquid layer cannot fully immerse the sphere at $t = \tau$, and no singular jet is ejected, as illustrated in Fig. 4-4.

For hydrophobic spheres, three distinct wetting regimes will appear. When $v > U^*$, wetting failure occurs, and the contact line tends to pin near the sphere's equator[38,41,43,44]. This results in a larger cavity above the sphere (Fig. 4-3,5), in contrast to the narrow air channel formed in the hydrophilic case[40,43] (Fig. 4-1). If $v$ is high enough, the rim of the detached liquid layer converges above the sphere and ejects a

singular jet, leaving behind an air bubble comparable in size to the sphere[40,41] (Fig. 4-3). When $v$ is not fast enough (still larger than $U'$), a pit forms above the sphere with its aperture remaining open (Fig. 4-5). Another case is that when $v < U'$, the liquid layer wets the sphere more slowly, and the sphere remains unimmersed at $t = \tau$ (Fig. 4-6).

In the experiment, we observed singular jets appearing before $t = \tau$ in Modes I, II, and III.a. This suggests that by this time, the sphere was fully submerged, aligning with immersion regimes 1, 2, or 3 in Fig. 4. Conversely, no singular jets were observed until $t = \tau$ in Modes III.b, IV, and V, indicating that the sphere was not yet fully submerged at $t = \tau$, corresponding to immersion regimes 4, 5, or 6 in Fig. 4.

The second branching process occurs during the collapse phase of the spark bubble (Fig. 4). In the branch where the sphere is fully submerged, a singular jet is emitted (lower branch). A concavity emerges at the junction between the singular jet and the water bulge, and it expands into a cavity as the spark bubble collapses. This branch further divides into three distinct modes: I, II, and III.a. In the branch where the sphere remains unimmersed, the pit over the sphere evolves into an expanding cavity as the spark bubble collapses (upper branch). This routine also branches into three modes: III.b, IV, and V. At time $T_1$, and later at $T_1 + T_2$, these processes evolve into the snapshots shown in Fig. 2.

## Governing parameters of jet modes

As illustrated in Fig. 4, two primary branching processes of jet modes have been identified. The evolution of these modes involves distinct geometries, driving mechanisms, and time scales that differ fundamentally from those in classical water-entry scenarios. Therefore, it is necessary to identify appropriate dimensionless parameters beyond those traditionally used.

We focus first on the initial branching process and define an 'immersion time', $t_c^u$, which represents the duration required for the sphere to become fully submerged. To theoretically determine $t_c^u$, two cases are considered: one where wetting failure does not occur and one where it does. During the sphere's immersion, the liquid layer and contact line advance at a velocity $V_f \approx \zeta v$, where $\zeta$ is a prefactor on the order of unity[41,45] (Fig. 4). In the case without wetting failure, the wetted liquid layer and contact line move at a speed comparable to $v$. Using the arc length $s$ as the length scale, the immersion time can be estimated as $t_c^u \approx s/v$.

In the wetting-failure case, the detached liquid layer advances faster due to reduced viscous dissipation at the sphere's surface (with $\zeta = 2$ in Duez et al.[41]), and the distance it travels before converging is also increased. We introduce a factor $\zeta'$ to estimate this travel distance as $\zeta' s$, where $\zeta' > 1$. Although the exact value of $\zeta'/\zeta$ could not be determined experimentally due to optical distortion at the curved air-water interface, our observations confirm qualitatively that the upward travel distance before convergence remains on the same order as the sphere dimension (Supplementary Fig. 6). Given these considerations, the net impact of increased layer velocity and travel distance on the immersion time is difficult to quantify precisely. Nevertheless, since no drastic increase or decrease in immersion time magnitude is anticipated from these competing effects, we adopt $t_c^u \approx s/v$ as a first-order estimation. This simplification proves effective in classifying the experimentally observed jet modes, as demonstrated in Fig. 5.

Note that the immersion time $t_c^u$ depends on the relative speed, $v (= V_l - V_s)$. The characteristic velocity of the sphere is (Methods),

$$V_s \sim \left( C_h + 4\sin^2\frac{\theta}{2} We^{-1} \right) \rho^{-1} V_l. \quad (3)$$

Here, the Weber number, $We = \rho_l V_l^2 r/\sigma$, represents the ratio of liquid inertia to capillary force. $C_h$ is the hydrodynamic force coefficient[46–48]. In this study, the Reynolds number $Re = \rho_l V_l r/\mu > 10^2$. The flow can be considered as potential flow, and $C_h$ is of order unity[47–49]. The first term

in the brackets of Eq. (3) corresponds to the influence of liquid inertia, and the second term represents the effect of surface tension.

The critical condition for the first branching process is that the sphere fully submerged at $t = \tau$. Normalising $t_c^u$ by $\tau$, one obtains the dimensionless immersion duration: $t_c = t_c^u/\tau$. Physically, $t_c$ also characterises the ratio between the unimmersed arc length of the initially floating sphere and the distance travelled by the liquid layer during $\tau$. By substituting $V_l (\sim V_b/\hat{h}^2)$ and $V_s$ into $t_c$, which yields,

$$t_c = \frac{t_c^u}{\tau} = \frac{s}{v\tau} \sim \frac{\phi(Bo, \rho, \theta)\hat{r}}{\alpha} \hat{h}^2, \quad (4)$$

where $\hat{r}$ is the radius ratio between the sphere and the spark bubble, $\hat{r} = r/R_m$. The factor $\alpha$ is expressed as $\alpha \sim 1 - (C_h + 4\sin^2\frac{\theta}{2} We^{-1})\rho^{-1}$.

The dimensionless immersion time $t_c$ is determined by the initial conditions, including the parameter groups $We$, $Bo$, $\rho$, $\theta$, $\hat{r}$, and $\hat{h}$. It characterises the relative timing between the sphere's immersion and the expansion of the spark bubble. A transition in behaviour is expected to occur across a critical threshold of $t_c$, separating cases where the sphere is fully immersed at $t = \tau$ from those where it remains unimmersed.

The second branching process is mainly governed by the cavity dynamics. Driven by the negative radiated pressure from the spark bubble, the concavity at the junction between the singular jet and the water bulge in Modes I, II, and III.a, as well as the pit above the sphere in Modes III.b, IV, and V, expands into downward cavities (Fig. 4). Additionally, as shown in Fig. 3c, the cavity expansion velocity is primarily determined by $\hat{h}$. Therefore, $\hat{h}$ is the governing parameter characterising the second branching process.

## Distribution of the five jet modes

The dimensionless immersion time $t_c$ and the dimensionless depth $\hat{h}$ are used to characterise the two branching processes of jet modes. Since $t_c$ is influenced by $\hat{h}$ (Eq. (4)), to decouple these two parameters, we constructed the phase diagram based on $\hat{h}$ and $\phi\hat{r}/\alpha$ (where $\phi\hat{r}/\alpha = t_c/\hat{h}^2$, with $C_h = 1$ in all cases to calculate $\alpha$). As shown in Fig. 5, the five jet modes cluster distinctly.

When a sphere is present on the water surface, a jet forms even as $\hat{h}$ reaches 3.5 (Mode V in Fig. 5), exceeding the typical upper boundary of $\hat{h} \approx 1.2$ for jet formation on a flat water surface[17,18]. With the sphere, the energy density on the water surface required to generate a jet is roughly $(3.5/1.2)^{-2} \approx 12\%$ of that required without a sphere, which is approximately an order of magnitude lower (Methods). The presence of spheres significantly lowers the energy threshold required for jet formation, enhancing the system's sensitivity.

In Fig. 5, the oblique line marks the boundary between regimes with and without singular jets at $t = \tau$. It corresponds to the first branching process and applies to experiments using spheres with varying wettabilities. Along this line, a critical dimensionless immersion time of $t_c^* = 0.24$ was identified from experimental data. While dimensional analysis defines the structure of $t_c$, its numerical threshold must be determined experimentally. The value $t_c^* = 0.24$ reflects the combined influence of system-specific parameters, such as the characteristic length and velocity scales, whose numerical prefactors are not captured in the scaling argument. The phase diagram delineates the first branching process effectively using this critical value, indicating that $t_c$ serves as a robust and universal parameter for distinguishing jetting modes.

The remaining boundaries in Fig. 5 correspond to the second branching process and will be discussed in two categories. Modes III.b, IV, and V appear above the oblique boundary, corresponding to cases where the sphere remains unimmersed at $t = \tau$. As shown in Fig. 3c, the cavity expansion velocity $V_c$ decreases as $\hat{h}$ increases. Additionally, two critical values, $\hat{h} \approx 1.3$ and $2.1$, were identified to differentiate Modes III.b and IV, and Modes IV and V, respectively. When $V_c$ is sufficiently

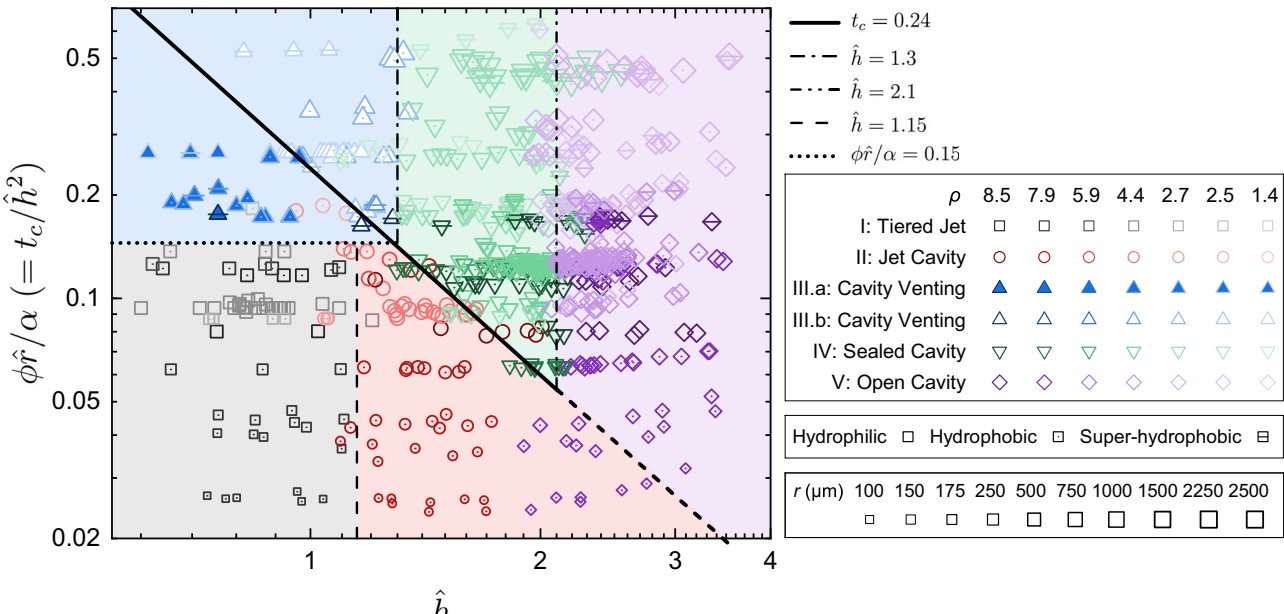

**Fig. 5 | Distribution of the five jet modes.** The plot of $\phi\hat{r}/\alpha$ ($=t_c/\hat{h}^2$) versus $\hat{h}$ categorises the five jet modes. $\hat{h}$ is the spark bubble's dimensionless depth. $t_c$ is the dimensionless immersion time, defined as the ratio of the sphere's immersion time to the spark bubble's expansion time. The formula $t_c = \phi\hat{r}\hat{h}^2/\alpha$ incorporates the azimuthal angle $\phi$ of the contact line, and the radius ratio $\hat{r}$ between the sphere and the spark bubble. The factor $\alpha$ is determined by the density ratio $\rho$, Weber number

$We$, and contact angle $\theta$, expressed as $\alpha = 1 - (1 + 4\sin^2\frac{\theta}{2}We^{-1})\rho^{-1}$. The oblique line marks the boundary at $t_c = 0.24$, distinguishing jet modes with singular jets (Modes I, II, and III.a below the line) from those without singular jets (Modes III.b, IV, and V above the line). Additional lines are included to indicate the boundaries between different jet modes. Source data are provided as a Source Data file.

high, the cavity reaches the spark bubble before $t = T_1$, a channel forms, resulting in Mode III.b. As $V_c$ decreases at larger values of $\hat{h}$, the cavity seals at its neck, leading to Mode IV. Further decreases in $V_c$ prevent cavity sealing, resulting in Mode V.

The critical condition distinguishing Modes III.b and IV is whether the cavity reaches the spark bubble at $t = T_1$. This leads to $V_c\tau_{coll} \approx h$, neglecting the uplift of the water surface at $t = \tau$ and the migration of the spark bubble's centroid at $t = T_1$. The resulting critical value is $\hat{h} \approx 1.23$, which closely matches the experimental result of $\hat{h} \approx 1.3$ (Fig. 5).

The distinction between Modes IV and V depends on whether the cavity rim seals at $t = T_1$. As the spark bubble collapses, the cavity expands, allowing air to flow into and fill the void (Fig. 4). During the sealing of the cavity rim, the aerodynamic pressure drop $\Delta p$ induced by the high-speed airflow and the Laplace pressure $p_s$ induced by surface tension take effect. We introduced a Weber number $We^*(=\Delta p/p_s)$ to assess the relative importance between these two pressures (Methods and Supplementary Fig. 7). $We^*$ ranges from $O(10^1)$ to $O(10^5)$ at the boundary between Modes IV and V. Based on this scaling, we consider aerodynamic pressure to be the primary contributor in determining this mode transition, and surface tension is not required to explain the observed boundary.

A modified Rayleigh-Plesset equation can be obtained for characterising this sealing process[38,50]: $\rho_l\left(b\ddot{b} + \frac{3}{2}\dot{b}^2\right) = \Delta p$, where $b$ denotes the distance from the cavity axis to its rim. The collapse duration of the rim is further derived, $\tau_{rim} \sim b_0\sqrt{\rho_l/\Delta p}$, where $b_0$ is the initial distance from the rim to the cavity axis. If $\tau_{rim} \leq \tau_{coll}$, the cavity seals before the spark bubble rebounds, resulting in Mode IV. Otherwise, the cavity remains open at $t = T_1$, leading to Mode V. The value of $\tau_{rim}$ is primarily governed by $\Delta p$, which depends on $\hat{h}$. This suggests a critical $\hat{h}$ that determines cavity sealing, consistent with experimental observations showing that the vertical line at $\hat{h} \approx 2.1$ marks the boundary between Modes IV and V (Fig. 5).

Modes I, II, and III.a appear below the oblique boundary (Fig. 5), corresponding to the lower branch where the singular jet forms

before $t = \tau$. In this branch, differences that appear before the collapse of the spark bubble include the water bulge height and whether a large air bubble is sealed above the sphere. These early differences subsequently influence the second branching into Modes I, II, and III.a.

For Modes I and II, there are three observable differences: water bulge height at $t = \tau$, concavity-sphere contact at $t = T_1$, and crown appearance at $t = T_1 + T_2$. Among them, concavity-sphere contact is used as the classification criterion, as it provides a clear binary indicator of sphere-surface interaction (Supplementary Fig. 8).

At $t = \tau$, two distances affect the transition between Modes I and II: the distance between the concavity and the sphere, $d_1 \approx (V_l - V_s)\tau$, and the distance between the concavity and the spark bubble, $d_2 \approx V_l\tau + h$. Here, $d_1$ represents the initial gap the concavity needs to close to reach the sphere, and $d_2$ determines the downward speed of the concavity, $V_c$. Ignoring the sphere's motion during the spark bubble's collapse phase, we find $V_c\tau_{coll} \approx d_1$, yielding a critical value of $\hat{h} \approx 1.04$ for $C_h = 1$, $We \gg 1$, and $\rho \to \infty$. This theoretical value is close to the experimental result of $\hat{h} \approx 1.15$ that separates Modes I and II.

The boundary separating Modes I and III.a appears as a horizontal line at $\phi\hat{r}/\alpha \approx 0.15$, independent of $\hat{h}$ (Fig. 5). These two modes correlate with the sphere's wettability and density: Mode III.a tends to occur with larger contact angles and lower densities, while Mode I is more likely to appear for more wettable and denser spheres (Supplementary Fig. 9). We hypothesise that these properties influence the formation and rupture of a sealed air bubble atop the sphere. On less wettable surfaces, wetting failure may lead to bubble formation, and larger contact angles can promote contact-line pinning near the equator[39–41,43,44]. A sealed bubble may then enable cavity venting if ruptured by the expanding concavity. Lower-density spheres may reduce the liquid layer thickness at $t = \tau$, increasing rupture likelihood. While these mechanisms reflect our current understanding, they do not explain why the transition boundary is horizontal. This remains an open question that will be investigated in future work.

In the phase diagram, the oblique boundary at $t_c = 0.24$ separates jet modes with and without singular jets before $t = \tau$. However, several experiments, classified as Mode V without singular jets, fall below this boundary (Fig. 5). These exceptions occur with sphere radii $r$ ranging in 100–175 µm and $\hat{h}$ between 1.9 and 2.8.

We hypothesise that in these experiments, the spheres are fully submerged at $t = \tau$ (Supplementary Fig. 10). The singular jet radius $r_{jet}$ is roughly an order of magnitude smaller than the sphere's radius (Fig. 2b-I, II), with $r_{jet} = O(10)$ µm for spheres with radii between 100 and 175 µm. By balancing the jet's kinetic energy per unit length ($\sim \pi \rho_l r_{jet}^2 V_r^2 / 2$) with its surface energy per unit length (-$2\pi\sigma r_{jet}$), the reduction in jet velocity due to surface tension is $V_r \sim [4\sigma/(\rho_l r_{jet})]^{1/2}$[36,51]. For $r_{jet} = 10$ µm, $V_r$ ~ 5.4 m/s. In the experiments, the singular jet velocity at $\hat{h} \approx 1.7$ is $V_{jet} = O(1)$ m/s. $V_r$ is of the same order as $V_{jet}$, suggesting that surface tension significantly suppresses jet formation.

The dominance of surface tension in the observed jet dynamics is further evident from the relevant dimensionless numbers. The Weber number based on jet parameters, $We_{jet} = \rho_l V_{jet}^2 r_{jet}/\sigma$, is of order $O(10^{-1})$, indicating that inertial forces are relatively weak compared to surface tension. Similarly, the capillary number, $Ca_{jet} = \mu V_{jet}/\sigma$, is on the order of $10^{-2}$, suggesting that viscous effects are also minor. The Ohnesorge number, $Oh = \mu/\sqrt{\rho_l \sigma r} \approx 0.01$, calculated using the sphere radius $r$ as the characteristic length scale, aligns with conventions in bubble-bursting studies that use the bubble radius rather than the jet radius[3,15,52]. Previous work has shown that viscosity plays a significant role in suppressing jet formation when $Oh > 0.037$, with complete inhibition at $Oh = 0.1$[3,52], well above the range encountered here. These estimates collectively indicate that, for small spheres and large $\hat{h}$, surface tension governs the dynamics, effectively suppressing the emergence of singular jets.

## Discussion

In summary, this study reveals that surface jet dynamics are more complex and more sensitive to particulates than previously understood. The presence of particulates on the water surface induces surface defects. The interaction between the underwater spark bubble, the water surface defect, and the sphere reshapes the jet dynamics. We observed five new jet modes, showing enhanced variability in jet types. Moreover, the jet velocities are increased, and the effective range of the spark bubble's dimensionless depth for jet formation is extended. These observations illustrate that the presence of particulates reduces the energy required for jet formation, and enhances the responsiveness of the water surface.

We identified two primary branching processes leading from a sphere resting on the water surface to the formation of five distinct jet modes. To characterise these processes, we introduced two key dimensionless numbers: the sphere's dimensionless immersion time, $t_c$, and the spark bubble's dimensionless depth, $\hat{h}$. The immersion time $t_c$ characterises the first branching process, depending on whether the sphere becomes fully submerged during the spark bubble's expansion phase. The second branching process occurs during the spark bubble's collapse phase, where the cavity dynamics are crucial and mainly governed by $\hat{h}$. Despite the complexity of the phenomena, our proposed model, based on these two dimensionless numbers, is applicable across most experimental conditions in this study.

In the current experiments, distortions induced by curved interfaces, coupled with the intense light emitted by the spark bubble, present significant challenges for accurate measurement. These effects restrict our ability to capture detailed quantitative data. In particular, they limit measurements of the sphere's motion and the surrounding air-liquid interfaces, which are important for analysing the transition between Modes I and III.a. As a result, the physical mechanisms that govern this transition remain unclear. Although the underlying physics remains not fully understood, we highlights the influence of sphere properties and wetting failure on jet mode transitions. Future work will use numerical simulations to investigate the dynamics of the sphere, the motion of contact lines, and the cavity venting behaviour characteristic of Mode III.a.

This study uncover new physics in jet dynamics driven by spark bubble-particulate interactions. Tiny surface defects introduced by particulates act as energy-focusing sites, lowering the jet-formation threshold by an order of magnitude and giving rise to five distinct jet modes. These complex phenomena stem from the simple interplay between a spark bubble and a surface defect, showing how small-scale heterogeneities can reshape free-surface dynamics. This work not only bridges a gap in classical fluid dynamics by highlighting the enhanced sensitivity and variability of jet behaviour in the presence of particulates, but also offers practical strategies: removing particles to suppress aerosol generation, or introducing them to promote jetting. These advancements in understanding have practical implications across diverse fields.

In environmental science, these findings offer a new perspective on how particulate matter can affect water jet behaviour. For example, microplastics floating on the ocean surface may promote the formation of spray aerosols, which in turn could carry pollutants or pathogens into the atmosphere. Gaining a better understanding of these processes is important for assessing environmental risks and protecting aquatic ecosystems. In medical engineering, this study also provides useful insights for improving aerosol drug delivery. The presence of particulates increases the variability and responsiveness of jet formation, which could be used to generate aerosols more efficiently and with less energy. Moreover, the formation of fast, focused jets opens up possibilities for needle-free drug delivery methods.

## Methods
### Experimental setup
The experimental apparatus was constructed for observations of the interplay between an underwater spark bubble and a floating sphere on the water surface (Fig. 1a). The principal components of the setup comprised the following elements:

- **Water tank:** A transparent acrylic tank was employed, with dimensions of $25 \times 25 \times 30$ cm³. It is filled with deionised water to a depth of 25 cm.
- **Discharge electrodes:** Copper wires with a diameter of 60 µm were used as electrodes to create the spark bubbles. These were manipulated with a three-dimensional platform for positioning.
- **Sphere positioning:** The spheres were initially picked up with tweezers and gently placed on the water surface. To fine-tune their position, a rubber suction bulb was used, which allowed for control by gently blowing air. We used the feedback from the high-speed camera to monitor and adjust the position so that it was directly above the contact point of the copper wires. This method minimised the drift of the sphere from the desired location before each experiment commences.
- **High-speed imaging:** Two high-speed cameras (Phantom VEO 711, Vision Research Inc., USA) equipped with parallel back lighting were utilised to capture the interaction process. The cameras were set orthogonal to each other to record the front and side views of the surface defect evolution and the underwater spark bubble behaviour, respectively. The frame rate was set at between 7500 and 60,000 frames per second, with an exposure time ranging from 5 to 20 µs.
- **Ambient condition:** The water tank was placed in an environment maintained at a consistent temperature of about 25 °C for 24 h before conducting experiments.
- **Data acquisition and analysis:** Matlab built-in image processing and edge detection algorithms were used in measuring the physical quantities. Accuracy in the image analysis was maintained to a single pixel, corresponding to a spatial resolution of 35 µm.

## Materials

The experiment employed spheres with radii ($r$) ranging from 100 to 2500 μm. These spheres, made from different materials, were selected to cover a range of relative densities, $\rho = \rho_s/\rho_l = 1.4$–8.5, where $\rho_s$ and $\rho_l$ represent the densities of the solid sphere and water, respectively. The materials used for the spheres included H62 copper, 304 stainless steel, zirconia ($ZrO_2$), titanium alloy (TC4 Ti), 1060 aluminium (1060 Al), silicon dioxide ($SiO_2$), and polyoxymethylene (POM).

The spheres were prepared with a cleaning process for experiments. They were ultrasonically cleaned in acetone, ethanol, and deionised water, lasting 30 min for each step. Additionally, spheres were coated with different materials to alter their contact angles. To obtain hydrophobic spheres, a fluorinated silane with low surface energy: 1H, 1H, 2H, 2H-perfluorodecyltriethoxysilane (Beijing HWRK Chemical Co., Ltd, China), was deposited onto the sphere surface via evaporative deposition. The super-hydrophobic spheres were fabricated by coating the surface with a thin layer of hydrophobic nano-particles. The cleaned spheres were immersed in the coating liquid (Ultra Glaco, Soft99 Co., Japan). After coating, the spheres were taken out and left to dry naturally for 12 h, resulting in super-hydrophobic surfaces.

The contact angle, $\theta$, was measured by using a sessile drop method. The values were validated by calculating the force balance in a quiescent floating state (Fig. 1c). Table 1 summarises the density and contact angle data for spheres used in the experiments.

Deionised water was used in the experiment, with a density of $\rho_l = 998$ kg/m$^3$, a viscosity of $\mu = 1$ mPa · s, and a surface tension of $\sigma = 72.8$ mN/m.

## Spark bubble generation and characterisation

The low-voltage underwater spark-discharge method was used to generate the spark cavitation bubble[53]. Upon activation of the circuit at $t = 0$, an initial spark at the electrodes' contact point rapidly vaporises the adjacent water, releasing a substantial amount of heat. It leads to the formation of an expanding bubble, driven by the high internal pressure and temperature. Following the spark discharge, the bubble expands to its maximum radius at $t = \tau$, then collapses to its minimum volume at $t = T_1$, and rebounds, resulting in oscillations. In our experiments, the maximum radius of the spark bubbles is $R_m = 9.2 \pm 0.5$ mm.

$\tau$ is chosen as the characteristic time of the spark bubble, which can be theoretically determined using the Rayleigh-Plesset equation for spherical bubbles[35],

$$\tau \cong 0.915 R_m \sqrt{\frac{\rho_l}{p_{atm} - p_v}}, \qquad (5)$$

where $p_{atm}$ and $p_v$ are the atmosphere pressure and the saturated vapour pressure of water, respectively. $T_1$ is theoretically twice $\tau$ for

spherical bubbles[35]. The presence of the free surface accelerates the collapse of the spark bubble[17]. In our experiments, $T_1/\tau = 1.9 \pm 0.1$.

Furthermore, the characteristic velocity of the spark bubble is determined as

$$V_b = \frac{R_m}{\tau} \cong \frac{1}{0.915} \sqrt{\frac{p_{atm} - p_v}{\rho_l}}. \qquad (6)$$

$V_b$ is independent of the spark bubble's size. By substituting $p_{atm} = 101325$ Pa, and $p_v = 3169$ Pa at 25 °C into Eq. (6), we obtain $V_b \approx 10.8$ m/s.

## Non-axisymmetric wavy feature in Mode III

A non-axisymmetric wavy feature was observed in the primary jet of Mode III after $t = T_1 + T_2$ in the Supplementary Movie 3. The instability of a jet and its subsequent breakup into droplets or waves are complex interactions involving surface tension, aerodynamic forces, and initial disturbances[54]. In Mode III, high-speed airflow rushes into the spark bubble, followed by the pinch-off of the channel, introducing strong disturbances on the channel wall. Additionally, random factors such as fragmented air bubbles and the horizontal motion of the sphere exist. These result in a sinuous wave shape of the jet.

## Critical values for wetting transition

The wetting transition occurs when the relative speed of liquid-solid surpasses the maximum contact line speed allowed[41]. For hydrophilic surfaces ($\theta < 90°$), $U^* \approx 0.1 \sigma/\mu$, which is basically independent of the contact angle. For hydrophobic surfaces ($\theta > 90°$), $U^*$ is lower and dependent on the contact angle, $U^* \approx (7/270)\sigma/\mu(\pi-\theta)^3$.

In this study, three types of spheres with different wetting abilities were used (Tab. 1). Based on the wetting transition model by Duez et al.[41], the values of $U^*$ for the hydrophilic, hydrophobic, and super-hydrophobic spheres used in our experiments are 7.3, 3.8, and 0.2 m/s, respectively. By neglecting the sphere's speed at the initial phase ($V_s \approx 0$), the critical values of $\hat{h}$ for wetting transition can be determined by setting $V_l (= 0.64\hat{h}^{-2} V_b)$ equal to $U^*$. The critical values of $\hat{h}$ are 1.0, 1.4, and 6.3 for hydrophilic, hydrophobic, and super-hydrophobic spheres, respectively. In our experiments, $\hat{h}$ ranges from 0.6 to 3.5. This suggests that wetting transition will occur for both hydrophilic and hydrophobic spheres, while super-hydrophobic spheres will consistently undergo wetting failure.

## Dimensional analysis

Through dimensional analysis, we simplify the system in this study into seven dimensionless parameters, which are the top seven dimensionless numbers listed in Table 2. They are the Weber number $We$, representing the ratio of liquid inertia to capillary force, the Froude number $Fr$, which compares liquid inertial to gravitational force, the Reynolds number $Re$, indicating the ratio of liquid inertia to viscous force, the density ratio $\rho$, the contact angle $\theta$, the radius ratio between the sphere and the spark bubble, $\hat{r}$, and a dimensionless depth, $h^*$. The table also includes the Bond number $Bo = We/Fr^2$, which compares gravitational to capillary force, as well as the spark bubble's dimensionless depth, $\hat{h} = h/R_m = h^*\hat{r}$, a parameter commonly used in studies on cavitation bubble interactions with flat water surfaces[17,18]. In this study, we use $\hat{h}$ instead of $h^*$.

The ranges of these dimensionless parameters are summarised in Tab. 2. The Reynolds number ranges from $10^2$ to $10^4$, indicating that the inertial force significantly outweighs the viscous force, and the viscous force can be neglected. The Froude number $Fr > 30$ suggests that the inertial force dominates over the gravitational force. The Weber number ranges from $10^0$ and $10^3$. At lower Weber numbers, surface tension effects become pronounced and should be considered, whereas at higher values, inertial forces dominate and surface tension can be neglected. The Bond number $Bo < 1$, suggesting that

**Table 2 | Relevant dimensionless parameters and their characteristic values**

| Dimensionless numbers | Symbol | Definition | Range |
|---|---|---|---|
| Weber number | $We$ | $\frac{\rho_l V_l^2 r}{\sigma}$ | 2–6126 |
| Froude number | $Fr$ | $\frac{V_l}{\sqrt{gr}}$ | 30–412879 |
| Reynolds Number | $Re$ | $\frac{\rho_l V_l r}{\mu}$ | 112–14938 |
| Density ratio | $\rho$ | $\frac{\rho_s}{\rho_l}$ | 1.4–8.5 |
| Contact angle | $\theta$ | $\theta$ | 81°–154° |
| Radius ratio | $\hat{r}$ | $\frac{r}{R_m}$ | 0.01–0.27 |
| Dimensionless depth | $h^*$ | $\frac{h}{r}$ | 6–268 |
| Bond number | $Bo$ | $\frac{\rho_l r^2 g}{\sigma} = \frac{We}{Fr^2}$ | 0.001–0.8 |
| Spark bubble's dimensionless depth | $\hat{h}$ | $\frac{h}{R_m} = h^*\hat{r}$ | 0.6–3.5 |

Italicised symbols denote physical quantities and dimensionless numbers.

gravitational and buoyancy forces are negligible compared to capillary force[46,55–57].

## The sphere velocity

The sphere's motion equation in the vertical direction can be expressed as,

$$(m_s + m_a)a_s = f_d + f_c + f_v + f_g + f_b, \tag{7}$$

where $m_s$ is the sphere's mass, $m_a$ is the added mass, and $a_s$ is the sphere's acceleration. The terms $f_d, f_c, f_v, f_g,$ and $f_b$ represent the form drag, capillary force, viscous force, gravitational force, and buoyant force, respectively.

Based on the dimensional analysis, we find that the liquid inertia and capillary force are dominant forces for the sphere. Therefore, by neglecting $f_v$, $f_g$, and $f_b$, and defining a hydrodynamic force $f_h = f_d - m_a a_s$ following previous studies[46–48], we obtain:

$$m_s a_s = f_h + f_c. \tag{8}$$

The hydrodynamic force $f_h$ can be represented as[46–48]: $f_h = \frac{\pi}{2}C_h\rho_l V_l^2 r^2$, where $C_h$ is the hydrodynamic force coefficient. For liquid flow around the sphere with $Re > 10^2$, it can be assumed as potential flow, with $C_h$ being of order unity[47–49,58]. The capillary force is influenced by the wettability of the sphere[57], with a characteristic value of $f_c = \pi r(1 - \cos\theta)\sigma$.

By substituting $f_h$ and $f_c$ into Eq. (8), we can give the scale of sphere acceleration: $a_s \sim (C_h + 4\sin^2\frac{\theta}{2}We^{-1})\rho^{-1}V_l^2 r^{-1}$. Further, with the time scale for sphere acceleration, $\Delta t \sim rV_l^{-1}$, the characteristic velocity of the sphere is obtained:

$$V_s \sim a_s\Delta t \sim (C_h + 4\sin^2\frac{\theta}{2}We^{-1})\rho^{-1}V_l. \tag{9}$$

## Energy density on the water surface

The mechanical energy of a spark bubble is determined by its volume and the driving pressure, given as $E = \frac{4}{3}\pi R_m^3(p_{atm} - p_v)$[17]. This energy, $E$, is initially stored as pressure potential energy resulting from the work done by the radiated pressure of the spark bubble, which deforms the water surface and initiates jet formation. The water surface is located a distance $h$ above the spark bubble, and the energy density radiated by the spark bubble onto the water surface can be characterised as

$$e \sim \frac{E}{4\pi h^2} \propto R_m(p_{atm} - p_v)\hat{h}^{-2}. \tag{10}$$

In this study, the driving pressure $(p_{atm} - p_v)$ remains constant, and $R_m$ is fixed at $9.2 \pm 0.5$ mm. As a result, $e$ depends solely on $\hat{h}$ and is inversely proportional to $\hat{h}^2$.

## Aerodynamic pressure vs. Laplace pressure in Modes IV and V

We evaluate the aerodynamic pressure drop across the cavity rim, $\Delta p$, with the Laplace pressure $p_s$ induced by surface tension. According to Bernoulli's principle, $\Delta p \sim \rho_a V_a^2$, where $\rho_a$ is the air density and $V_a$ denotes the airflow speed entering the cavity. The Laplace pressure is given by $p_s = \sigma(\frac{1}{a} - \frac{1}{b})$, where $a$ is the radius of the rim itself and $b$ is its distance to the cavity axis. Based on the geometric characteristics of the liquid rim in Modes IV and V, both $a$ and $b$ are on the order of $r$, leading to $p_s \sim \sigma/r$. A Weber number is defined as,

$$We^* = \frac{\Delta p}{p_s} \sim \frac{\rho_a V_a^2 r}{\sigma}. \tag{11}$$

$We^*$ quantifies the ratio of aerodynamic to Laplace pressure, with $We^* \gg 1$ indicating the dominance of aerodynamic effects.

As the spark bubble collapses, the cavity expands, drawing air into the cavity from the surrounding atmosphere. The rate of cavity volume increase is equal to the airflow rate through its aperture. Neglecting compressibility effects, $V_a$ can be estimated through $r_c^2 V_c \sim b^2 V_a$, which gives $V_a \sim (r_c/b)^2 V_c$. The cavity radius $r_c$ can be approximated as $r_c \sim V_c\tau$, where $V_c$ is the cavity expansion velocity and $\tau$ is the characteristic time of the spark bubble. Substituting $V_a \sim V_c^3/(b/\tau)^2$ and $b \sim r$ into Eq. (11) gives

$$We^* \sim \frac{\rho_a \tau^4 V_c^6}{\sigma r^3}. \tag{12}$$

In our experiments, $\tau \approx 1$ ms. $V_c$ ranges from 3 to 18 m/s in Mode IV and from 0.8 to 5 m/s in Mode V (Fig. 3c). Using these values, we evaluated $We^*$ across the experimental data in Modes IV and V, as presented in Supplementary Fig. 7.

## Data availability

The processed data for all main and Supplementary figures are provided in the Source Data file. Raw experimental data are available via the University of Strathclyde KnowledgeBase at https://doi.org/10.15129/f7280d66-1735-4dd3-b3d2-7c1d473e1f2c. All other data that support the plots within this paper and other findings of this study can be obtained by contacting the corresponding authors. Data will be shared for non-commercial academic use via the institutional file transfer service. Source data are provided with this paper.

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

## Acknowledgements

X. Cheng and X. Chen acknowledge the support provided by the National Natural Science Foundation of China (grant no. 11872315), the Guangdong Basic and Applied Basic Research Foundation (grant no. 2022A1515011201). L.J. thanks the support from the Royal Society Research Fund (RGS\R2\222218). We thank A-Man Zhang and Yunlong Liu at Harbin Engineering University; Erqiang Li at University of Science and Technology of China; Haibao Hu, Hengdong Xi, Xiao Huang and Jun Luo at Northwestern Polytechnical University for helping with our experiments. We acknowledge Chao Sun at Tsinghua University for helpful discussions.

## Author contributions

X. Cheng and X. Chen developed the original concept for this study. The design of the experiments was a collaborative effort by X. Cheng, X. Chen, and L.J. X. Cheng carried out the experiments. The analytical phase was conducted jointly by X. Cheng, X. Chen, and L.J. Discussion and interpretation of the results and the manuscript were contributed to the collaborative efforts of X. Cheng, X. Chen, Z.Y., and L.J.

## Competing interests

The authors declare no competing interests.
