## [Transparent Peer Review file · Nature Communications]

Particulate Reshapes Surface Jet Dynamics Induced by a Cavitation Bubble

Corresponding Author: Dr Laibing Jia

Version 0:

Reviewer comments:

Reviewer #1

(Remarks to the Author)

The authors have performed several experiments to study the jet dynamics on the water surface in the presence of particles (impurities). The paper presents results from a suite of experiments carried out and highlights the formation of 5 different types of jets depending on the initial dimensionless properties chosen. The author emphasized that the structure of the jet and its type is primarily determined by two dimensionless numbers γ (dimensionless height) and t_c (dimensionless chasing time).

While the experimental setup along with results are mentioned there are certain clarifications required which I have listed below.

Additionally, a major piece missing in this work is to unravel the physical mechanism related to the dynamics of water jets.

Here are three major points which the authors needs to address::

1. Choice of sphere radius and its influence in the jet dynamics :

The authors in line 131-132 have mentioned that they have used spheres of varying sizes from 100 micron to 2.5 mm, however their experiments are primarily focussed with a sphere of radius 500 micron. The authors have also commented in the Methods (Material) section that this choice is primarily due to constraints of capturing the image of the resulting jet in the camera. In this regard, few clarification is required:

Firstly, the points in Figure 2 that show distribution of jet modes, it is not clear if these points are done using the spherical radius of 500 micron only or even other spherical radius have been included. If this was done using a 500 micron sphere only, then it would be prudent to check if such a phase plot trend also holds for another larger radial sphere. This is essential as the authors have mentioned that the radiative force has a r^2 dependence, so doubling the radial size of the sphere would increase the radiative force by a factor of 4. In which case, would the resulting jets show all these 5 types and what would be the threshold value of t_c in the case of larger spherical radius.

2. Oscillation period of the spark bubble.

The authors have mentioned that they study the jet behavior up to $t = T_1$ first oscillation period of the spark bubble which they have mentioned to be fundamental in their study (Lines 160-161) as their expansion and collapse drive jet formation (line 156), however, the authors have never discussed what's the value of T_1 and if this would same for all their experiments. Also it's not clear if the value of τ same for all experiments. If they are the same then τ/T_1 would have a unique value, perhaps the author should mention that as characteristic of the bubble properties. If these time scales are different for each experiment then authors need to comment on how this ratio determines the formation of different jet modes.

3. Non-axisymmetric modes and height of the water jets

Going through the movies, a comparison of Mode III : Ventilation with reference shows formation of the jet, while the reference jet is symmetric and the jet formed in the Ventilation mode shows non-axisymmetric instabilities (From 5.3 to 5.9

ms).

Can the authors comment on their physical origin?

Further, the authors should present an analysis on the height attained by the jets formed in different modes and its dependence on t_c and γ .

A detailed physical analysis of the non-axisymmetric nature of the jet and its maximum height would provide this study an added value beyond the current academic exercise and cater to the various plausible application scenarios.

Reviewer #2

(Remarks to the Author)
attached in the word file.

Reviewer #3

(Remarks to the Author)

Report to Manuscript NCOMMS-24-09227-T entitled "Particulates Reshape Jet Dynamics on Water Surface" by Cheng et al.

The authors reported the jet dynamics of a cavitation bubble near the water surface under the effect of a particle rest at the water surface. Depending on the distance of the cavitation bubble from the free surface and the particle density and wettability, the authors identified five different jetting modes, provided the corresponding regime map, and discussed the evolution of the jet, considering two dimensionless numbers, the dimensionless bubble depth and the dimensionless chasing time. The presence of the rest particle at the liquid surface significantly modifies the jet dynamics from the cavitation bubble, which not only yields interesting phenomena as reported by this manuscript, but indicates unexplored physics regarding the interplay between the particle dynamics and the cavitation bubble dynamics.

This manuscript is organized well, and presents new phenomena with significant impact over a wide range of natural and industrial processes. However, the authors mainly described and discussed different jet phenomena without enough physical insights and theoretical analyses. I thus suggest the authors to improve the theoretical part of the manuscript, as I suggest in the below comments.

1. I suggest to revise the title of the manuscript to describe this study more precise. Here is a suggestion: "Particulates reshape the jet dynamics of a cavitation bubble near a water surface".
2. There are some studies discussing the interplay between the cavitation bubble and the particle, such as Ren, PRL, 2022, 128: 044501 and Wu, PRL, 2017, 119: 084501. Though these studies did not involve a free water surface, the theoretical analyses on the interplay between the cavitation dynamics and the particle dynamics may help.
3. Line 131-133: "...spheres of varying sizes, ranging from 100 μm to 2.5mm in radius, exhibited distinct phenomena compared to a clean water surface". I am confused with this sentence. Which phenomenon is distinct? The authors stated that they used 500 μm particles in the following sentence. Did they really fixed the particle size?
4. Can the authors indicate the particle materials for each data points in Fig. 2, using different point shapes or colors?
5. Some important information in the Methods part should be removed to the main text. For example, the fixed maximum bubble size. The definition and the physical meaning of the two dimensionless numbers are crucial for the readers to understand the regime map and jet evolution schematics.
6. The authors identified the jetting modes based on the jet feature at $t = T_1$. However, I think the global features of the jet are more important and more reasonable. For example, in figure 3, it seems the cavity in mode V will collapse after T_1 and may generate a jet, showing a global feature same as Mode IV. I suggest the authors to check the jet dynamics during the whole jetting process instead of only an oscillation period.
7. I suggest the authors to give some typical time sequences from experiments to support the schematics in Fig. 3.
8. The dimensionless chasing time is not a parameter that apparently decided by the initial parameters of the system, which makes it inconvenience to use in practical situations for the prediction and control of the jet morphology. I prefer dimensionless numbers based on the initial parameters of the system, with clear physical meanings. Why not using dimensionless numbers that represent the comparison of different forces, such as the inertial, viscous, and capillary forces, etc.?
9. I am wondering that if the two dimensionless numbers are enough to describe the dynamics of the jet dynamics under the influence of particle. For example, the initial configuration of the particle are considered in the dimensionless chasing time, by involving h_{im} . However, the particle density and wettability also significantly affect the motion of the particle at interface (Ji, Chemical Engineering Science, 2019, 207: 17-29; Ding JFM, 2015, 783: 504-525), and thus may change the dynamics of the cavity and jet.
10. My major concern is that there are no quantitative analyses (even simple scaling) on the experimental findings. Taking the regime map as an example, the authors can derive some theoretical scaling to describe the boundaries between different modes. The characteristic parameters of the jet, such as size, height, and velocity are also deserved to study, and the related theoretical models can be proposed. Of course, all the above suggestions rely on critical physical insights of the behind physics regarding the interplay between the particle motion and the flow field generated by the cavitation bubble, which is lacked in the present form and should be included in the revision.

Reviewer #4

(Remarks to the Author)

The authors investigate liquid jet formation in presence of a solid particle floating on the liquid surface. While previous studies have been conducted with particles of nano or micrometric size, the authors aim to extend the problem to

macroscopic particles. By means of an oscillating bubble triggering the jet formation and spheres of varying properties (density and contact angle), the authors show that the jet shape and dynamics are affected by the floating sphere. In particular, they describe the five type of jets that can emerged depending on the initial configuration and the energy transferred by the spark bubble. The key parameters governing the problem are compiled in two characteristics quantities whose ratio determines the type of jet. In addition, a tentative qualitative explanation for the mechanisms leading to each type of jet is provided.

While these experimental observations are interesting, the paper lacks clarity and quantitative results that would support its claims, which limits the broad impact of this study. In the present form, the paper can not be recommended for publication.

Below are detailed major concerns as well as minor comments.

1) The paper is confusing and needs a thorough reorganisation before being accepted for publication. Descriptions and definitions are given in different parts of the paper and methods, and one need to look for information in many places to make sense of the paper. In particular:

- the different jet modes need to be described as they are presented. Fig. 3 is a good guide but should arrive before the diagram Fig.2.
- line 183, it is unclear what "ventilation" is; in general more explanations are needed when terms are introduced.
- the definition of the chasing time t_c , which contains all the physics of the problem, needs to be better described and in the main text.
- line 158: it is unclear at that point why the temporal parameters are fundamentals in the study. Furthermore, while the characteristic time of the spark bubble τ appears in t_c , it is not the case of the first oscillation period T_1 , which is then absent from the phase diagram Fig.2.

2) The authors claim that the presence of the particle lowers the energy needed for jet formation. There are no measurement in the paper that support this claim. In general, there are no quantitative measurements. In particular:

- it is said that the amplitude of the jet varies with the dimensionless depth of the spark bubble γ (line 168), but no graph is provided. Giving a plot of the amplitude vs γ , both for a particle-free interface and with a particle would be interesting. Furthermore, such a plot would be useful to support the fact that "A pivotal finding of our study is the notable impact of spheres on the water surface, significantly extending the effective range of γ from typically up to 1.2 on a flat water surface to values surpassing 4" and that particles "substantially lowers the energy requirement for jet generation". Here only qualitative comparison are provided in the supplementary videos.
- Measurements (e.g. evolution of the radius of the cavity or jet height with time) could be provided to better describe the different modes, in particular to support sentences like "the modes III to V depend substantially on the bubble depth γ . A smaller γ results in more rapid expansion of the cavity" (line 303).

3) In many places there is too much emphasis without substantial results.

- line 76: why is the approach "unique and innovative" ? How does this experimental setup allows to do something that has never be done before ? The spark bubble is not a novel method to produce jets (please refer to ref. 17 here).
- line 321: "not merely academic, it has tangible implications in various fields", line 333 "contributes to the advancements of fluid dynamics", line 335 "unlocks the potential for jet manipulations", "have far-reaching impacts" are all overselling; either develop, or provide at least one clear substantial result that support this. Currently the paper, while being a nice experimental study, remains a qualitative description of an interesting phenomenon in a specific range of parameters without full insight into physical mechanisms or modelling.

Minor comments:

- l. 115 : The dotted lines are hardly visible (maybe better in red ?)
- l. 132 : It says that particles of different sizes are used while both in methods and figures only particles of 500 microns are used.
- l. 176 : "While this study primarily focuses on the dynamics of liquid jet formation within the spark bubble's first oscillation period." Rephrase/ remove "while" ?
- l. 201 : Caption of fig. 2: Add the meaning of the dashed lines. Recall what γ and t_c represent ?
- l. 201 : Fig. 2 Are there any error bars on the measurements ? Are the experiments repeated ?
- l. 433 : "Studies on cavitation bubbles reveal that the force is significantly greater at the onset of expansion and at the end of collapse compared to other times". Could you add some references ?

Reviewer #5

(Remarks to the Author)

Version 1:

Reviewer comments:

Reviewer #1

(Remarks to the Author)

The revised manuscript have addressed the comments that I have given. Further, I can see that new figures with results from additional experiments have really improved the presentation and also showcases a deeper analysis which was missing in their previous version.

The authors have commented on the impact of not accounting for surface tension in their analysis which has impacted their results particularly for small radius sphere.

While, I would have ideally preferred a detailed analysis with adding the surface tension terms in their analysis, which would have modified the boundaries in Figure 5 but they have presented an argument on dimensionless numbers which also present a form of explanation highlighting the impact of surface tension. Overall am fine with the modifications.

Also the authors have taken time to explain the different time lines in their experiment which is indeed welcome.

Reviewer #2

(Remarks to the Author)

I think the authors have done substantial revisions to the revised manuscript. I am happy with the changes incorporated this time. I think the article can be accepted now.

Reviewer #3

(Remarks to the Author)

Please see attachment

Reviewer #4

(Remarks to the Author)

The presentation of the results has been improved, in particular with the addition of Fig. 3 and 5 and the reorganization of the text. However, the paper still lacks a clear message linked to quantitative analysis. We believe the measurements are not exploited enough and could be used to obtain a precise characterization of the regimes and a better understanding of the physical mechanisms.

- In the phase diagram Fig. 5, there is a clear dependence on the particule's radius which is not taken into account in neither dimensionless number (it cancels in t_c). This should be discussed with the physical description of the regimes.

- From the description of the regimes it seems like the relevant time scale is T_1 rather than τ . Since T_1 is roughly 2τ the limit at $t_c=0.2$ would be closer to $t_{\tilde{u}_c}=T_1$.

The two parameters t_c and h are not independent. The slope of the separation between regimes III and I simply reflects this dependence as clearly seen from all data points. There is more to understand to have an efficient collapse of the data and separation of the regimes.

- A loglog plot and power law scalings are hard to interpret when there is less than half a decade.

- While it is good to have some quantitative description of the regimes, Fig. 3 is hard to understand.

- There are more to extract from these measurements. For example, the jet velocity with or without particle seems to depend on the dimensionless depth h . Could you plot velocity vs h for the different modes ? Would it help understand the evolution of the depth of the cavity ?

Fig. 3f is not discussed thoroughly in the main text and it is hard to use the information it provides to gain an understanding of the physical mechanisms.

Minor remark: ventilation is still an odd name that does not describe well the fact that there is an airflow from the cavity.

Reviewer #5

(Remarks to the Author)

Version 2:

Reviewer comments:

Reviewer #3

(Remarks to the Author)

The authors provide an explanation using the water entry theory to support the criterion for the first branch, but this theory still cannot well explain the formation of all the final five jet modes. I am still worried that the dimensionless immersion time is not an initial parameter and lacks universality. Without rigorous theoretical arguments, as well as a broad enough impact, my feeling is, therefore, the manuscript would be better suited to a specialized journal in its present form. I believe it could be improved further by considering the following questions.

- 1) It is oversimplified for the derivation of t_c . The travel distance ζ 's here should be the pinch-off depth of sphere in water entry problem (Aristoff & Bush 2008 JFM). It was reported that this value can be much higher than the sphere size (ζ can be much larger than 1 in deep seal mode). Is it reasonable to consider ζ'/ζ as order of unity?
- 2) When describing the particle's immersion, "liquid film" is odd and I cannot understand what it refers to. The closure of the cavity above the particle can be resulted by the pinch off of the cavity instead of a "liquid film".
- 3) How do the authors measure the velocity V_c in Fig. 3h? Is it the average velocity during the whole expansion process?
- 4) Line 499: "the phase diagram effectively delineates the first branching process without requiring an additional fitting factor". 0.24 itself is a fitting parameter that smaller than 1 by one order of magnitude.
- 5) I cannot understand the derivations when discussing the boundary between Modes IV and V. Did the author neglect the contribution of Laplace pressure?
- 6) To explain the boundary between Modes III.a and I, the authors discuss the influences of particle density and wettability. It is not clear why a horizontal line divides these two modes. Can the authors explain it more directly?
- 7) When it comes to the failure of $t_c = 0.24$ to describe the occurrence of Mode V for small particles, the authors attribute it to the influence of surface tension. I am wondering that if the viscosity plays a role here. Viscosity becomes important and inhibits the jetting significantly at small length scales with a large Ohnesorge number as reported for bubble bursting jets. Besides, I suggest the authors to prove this argument by presenting the experimental images that indicate the cases of Mode V below $t_c = 0.24$ are from branch 4 in Fig. 4.

Reviewer #6

(Remarks to the Author)
See attached.

Version 3:

Reviewer comments:

Reviewer #3

(Remarks to the Author)
The authors have carefully revised the manuscript in response to my previous concerns and I am happy to recommend publication now.

Best,
Bingqiang Ji
School of Astronautics
Beihang University

Reviewer #6

(Remarks to the Author)
I thank the authors addressing my concerns, and I appreciate the authors' clarification regarding the relative magnitudes of aerodynamic pressure and Laplace pressure. The provided analysis convincingly demonstrates that the surface tension can be neglected in the current modeling framework. However, I still have one problem regarding the supplementary Figure 8. Based on the jet morphology observed during the first period (i.e., prior to the time T_1) the two sets of results presented in Supplementary Figure 8 appear to represent an identical jet mode. However, a discrepancy arises when examining Figure 4 in the main manuscript: the distinction between Mode I and Mode II seems to originate before the cavitation bubble collapses during the first period. This observation raises concerns about the consistency between Figure 4 and Supplementary Figure 8. I kindly request the authors to clarify or address this apparent inconsistency.

Version 4:

Reviewer comments:

Reviewer #6

(Remarks to the Author)
The authors have addressed my concerns clearly and provided sufficient explanation for the key differences between Modes

I and II. The revisions and the additional images make the manuscript clear and complete. I recommend the manuscript for publication.

Responses to Reviewers' Comments for Manuscript NCOMMS-24-09227-T

Particulate Reshapes Surface Jet Dynamics Induced by a Cavitation Bubble

Addressed Comments for Manuscript to

by

Xianggang Cheng, Xiao-Peng Chen, Zhi-Ming Yuan and Laibing Jia

Authors' Response to Reviewer 1

General Comments: The authors have performed several experiments to study the jet dynamics on the water surface in the presence of particles (impurities). The paper presents results from a suite of experiments carried out and highlights the formation of 5 different types of jets depending on the initial dimensionless properties chosen. The author emphasized that the structure of the jet and its type is primarily determined by two dimensionless numbers γ (dimensionless height) and t_c (dimensionless chasing time).

While the experimental setup along with results are mentioned there are certain clarifications required which I have listed below.

Additionally, a major piece missing in this work is to unravel the physical mechanism related to the dynamics of water jets.

Here are three major points which the authors needs to address.

Thank you for your valuable feedback and constructive suggestions. In the revised manuscript, we have made the following major changes:

1. The title was changed from 'Particulates Reshape Jet Dynamics on Water Surface' to 'Particulate Reshapes Surface Jet Dynamics Induced by a Cavitation Bubble' to emphasise the role of cavitation bubbles in the research.
2. A more thorough theoretical analyses on the key dimensionless numbers, the surface defect evolution underlying the five jet modes, and the boundaries between different jet modes have been analysed.
3. The key dimensionless parameters \hat{h} (dimensionless spark bubble depth) and t_c (dimensionless chasing time) were introduced in the main text, explaining their influence on the formation and dynamics of jets. (Notation \hat{h} replaced γ to avoid confusion with surface tension)
4. More experimental data was added in the updated phase diagram (Figure 5a) to systematically categorise the jet modes based on the dimensionless parameters.
5. Quantitative measurements and comparisons of results with and without particulates were presented in Figure 3 to show quantitative differences among jet modes, highlighting the differences in their kinematics.
6. Expanded details on the size, physical properties, and preparation methods of the spheres are provided in the Results and Methods sections.
7. Experimental snapshots of different jet modes were added as Figure 2, providing a direct view of the experimental results in different modes.

8. The jet shapes at $t = T_1 + T_2$ are added in Figures 2 and 4, and descriptions are added in the main text of the revised manuscript to reflect a longer-term evolution of the jets.
9. New references to studies on underwater particles driven by cavitation bubbles and sea spray aerosol (Wu et al. [2017], Ren et al. [2022], Sha et al. [2024]) were added in the introduction section.

Following are our point-by-point responses to your comments.

Comment 1: Choice of sphere radius and its influence in the jet dynamics.

The authors in line 131-132 have mentioned that they have used spheres of varying sizes from 100 micron to 2.5 mm, however their experiments are primarily focused with a sphere of radius 500 micron. The authors have also commented in the Methods (Material) section that this choice is primarily due to constraints of capturing the image of the resulting jet in the camera. In this regard, few clarification is required:

Firstly, the points in Figure 2 that show distribution of jet modes, it is not clear if these points are done using the spherical radius of 500 micron only or even other spherical radius have been included. If this was done using a 500 micron sphere only, then it would be prudent to check if such a phase plot trend also holds for another larger radial sphere. This is essential as the authors have mentioned that the radiative force has a r^2 dependence, so doubling the radial size of the sphere would increase the radiative force by a factor of 4. In which case, would the resulting jets show all these 5 types and what would be the threshold value of t_c in the case of larger spherical radius.

In our previous experiments, we tested some spheres with radii ranging from 100 μm to 2.5 mm to ensure the phenomena were applicable to a wide range of spheres. However, our primary focus was using spheres with a radius of $r = 500 \mu\text{m}$. The data shown in Fig. 2 of the previous version were drawn solely from experiments using $r = 500 \mu\text{m}$ spheres.

To thoroughly investigate the influence of sphere radius on jet dynamics, we conducted additional experiments using spheres with various properties. These spheres had radii ranging from 100 to 2500 μm , relative densities between 1.4 and 8.5, and wetting abilities from hydrophilic to super-hydrophobic (with contact angles from 81° to 154°). The results are presented in Fig. 5 of the revised manuscript. Following a reviewer's suggestion, we have replaced the notation γ with \hat{h} to represent the dimensionless depth of the spark bubble, to avoid confusion with the symbol for surface tension.

With the additional experimental data, the resolution of the phase plot for the two key dimensionless numbers \hat{h} and t_c has been highly improved, and the range of parameters

has been broadened. The distributions of five jet modes and their boundaries are now clearly presented in the phase plot. We identified six boundary lines compared to two in the previous version. We found that the horizontal line in the previous manuscript actually consists of an oblique line and a horizontal line. These boundaries have been analysed with respect to the evolution of the water surface in the revised manuscript.

Comment 2: Oscillation period of the spark bubble.

The authors have mentioned that they study the jet behaviour up to $t = T_1$ first oscillation period of the spark bubble which they have mentioned to be fundamental in their study (Lines 160-161) as their expansion and collapse drive jet formation (line 156), however, the authors have never discussed what's the value of T_1 and if this would same for all their experiments.

Also it's not clear if the value of τ same for all experiments. If they are the same then τ/T_1 would have a unique value, perhaps the author should mention that as characteristic of the bubble properties. If these time scales are different for each experiment then authors need to comment on how this ratio determines the formation of different jet modes.

Thank you for your comments and suggestions. The spark bubble ignites at $t = 0$, then expands to its maximum radius R_m at $t = \tau$, and collapses to its minimum volume at $t = T_1$. T_1 denotes the spark bubble's first oscillation period, and τ is the spark bubble's expansion duration during T_1 . In our experiment, $R_m = 9.2 \pm 0.5$ mm. ± 0.5 is the standard deviation (SD) of the mean for the measured R_m . Both τ and T_1 are functions of R_m . Ideally, $\tau \propto R_m$ and $T_1 = 2\tau$, based on the Rayleigh-Plesset equation for spherical bubbles[1].

We chose R_m as the characteristic length, which reflects the bubble energy. The corresponding timescale τ is selected as the characteristic time. In our experiment, T_1/τ remains almost constant. $T_1/\tau = 1.9 \pm 0.1$ (SD), a value slightly smaller than 2. The presence of the free surface accelerated the collapse of the spark bubble[2].

The free surface motion is driven by the the radiated energy density, e , originated from the spark bubble. As mentioned above, the parameters R_m and τ are chosen as the characteristic length and time to describe the spark bubble. The key dimensionless numbers \hat{h} and t_c are identified to determine the formation of different jet modes.

We have included this information in the revised manuscript.

Comment 3: Non-axisymmetric modes and height of the water jets.

Going through the movies, a comparison of Mode III : Ventilation with reference shows formation of the jet, while the reference jet is symmetric and the jet formed in the Ventilation mode shows non-axisymmetric instabilities (from 5.3 to 5.9 ms). Can the authors comment on their physical origin?

Further, the authors should present an analysis on the height attained by the jets formed in different modes and its dependence on t_c and γ .

A detailed physical analysis of the non-axisymmetric nature of the jet and its maximum height would provide this study an added value beyond the current academic exercise and cater to the various plausible application scenarios.

Thank you for pointing out the non-axisymmetric instability at the end of Supplementary Video 3. The instability of a jet and its subsequent breakup into droplets or waves is a complex interaction of surface tension, aerodynamic forces, and initial disturbances.[3] In this study, the jetting speed is high enough to suppress the development of instability on the jet during the spark bubble's first two oscillation periods $0 < t < T_1 + T_2$, unless the initial disturbance is strong enough. The non-axisymmetric wavy of the jet in Mode III is observed after $T_1 + T_2$. In this mode, the open channel between the spark bubble and the atmosphere induces high-speed airflow into the spark bubble, introducing strong disturbances on the channel wall. Additionally, random factors such as fragmented air bubbles and the horizontal motion of the sphere exist. These result in sinuous wave shape of the jet. The description of this instability observation is added in the description for the snapshots of Mode III in Figure 2 and the corresponding supplementary videos.

In the revised manuscript, we have added the measurements and analyses of the featured geometries of the jets and cavities from typical examples of the five jet modes. Their positions and velocities are illustrated in Figure 3 of the revised manuscript. For comparison, the results of jet/hump formation from flat water surfaces at the same \hat{h} are also presented. Based on Figure 3, we observed that the presence of particles significantly alters jet development. This leads to increased variability in jet types and enhanced jet velocity.

References

- [1] J. P. Franc and J. M. Michel. *Fundamentals of Cavitation*, pages p.36–39. Springer, Dordrecht, 2005.
- [2] S. Zhang, S. P. Wang, and A. M. Zhang. Experimental study on the interaction between bubble and free surface using a high-voltage spark generator. *Physics of Fluids*, 28(3):032109, 2016.

- [3] Constantin Weber. Zum zerfall eines flüssigkeitsstrahles. *ZAMM-Journal of Applied Mathematics and Mechanics/Zeitschrift für Angewandte Mathematik und Mechanik*, 11(2):136–154, 1931.

Responses to Reviewers' Comments for Manuscript NCOMMS-24-09227-T

Particulate Reshapes Surface Jet Dynamics Induced by a Cavitation Bubble

Addressed Comments for Manuscript to

by

Xianggang Cheng, Xiao-Peng Chen, Zhi-Ming Yuan and Laibing Jia

Authors' Response to Reviewer 2

General Comments: The experimental part of the paper was nicely done. However the manuscript can be further improved. Here are my queries:

Thank you for your positive feedback on the experimental section of this study. We appreciate your suggestions. In the revised manuscript, we have made the following major changes:

1. The title was changed from 'Particulates Reshape Jet Dynamics on Water Surface' to 'Particulate Reshapes Surface Jet Dynamics Induced by a Cavitation Bubble' to emphasise the role of cavitation bubbles in the research.
2. A more thorough theoretical analyses on the key dimensionless numbers, the surface defect evolution underlying the five jet modes, and the boundaries between different jet modes have been analysed.
3. The key dimensionless parameters \hat{h} (dimensionless spark bubble depth) and t_c (dimensionless chasing time) were introduced in the main text, explaining their influence on the formation and dynamics of jets. (Notation \hat{h} replaced γ to avoid confusion with surface tension)
4. More experimental data was added in the updated phase diagram (Figure 5a) to systematically categorise the jet modes based on the dimensionless parameters.
5. Quantitative measurements and comparisons of results with and without particulates were presented in Figure 3 to show quantitative differences among jet modes, highlighting the differences in their kinematics.
6. Expanded details on the size, physical properties, and preparation methods of the spheres are provided in the Results and Methods sections.
7. Experimental snapshots of different jet modes were added as Figure 2, providing a direct view of the experimental results in different modes.
8. The jet shapes at $t = T_1 + T_2$ are added in Figures 2 and 4, and descriptions are added in the main text of the revised manuscript to reflect a longer-term evolution of the jets.
9. New references to studies on underwater particles driven by cavitation bubbles and sea spray aerosol (Wu et al. [2017], Ren et al. [2022], Sha et al. [2024]) were added in the introduction section.

Following are our point-by-point responses to your comments.

Comment 1: I may have missed out. Can you please specify the physical parameters which affect the five different jet modes?

In our study, the spark bubble's maximum radius, R_m , the duration for the spark bubble to reach its maximum radius, τ , and the density of water, ρ_l , are chosen as characteristic parameters. The five jet modes are primarily determined by two key dimensionless numbers: the spark bubble's dimensionless depth, \hat{h} , and the dimensionless chasing time, t_c (we have replaced the notation γ with \hat{h} to represent the dimensionless depth of the spark bubble, to avoid confusion with the symbol for surface tension).

Jet formation is driven by the underwater spark bubble. The mechanical energy of a spark bubble is determined by its volume and the driving pressure. Theoretically, it can be expressed as $E = \frac{4}{3}\pi R_m^3 (p_{atm} - p_v)$, where p_{atm} and p_v are the atmosphere pressure and the saturated vapour pressure of water, respectively[1]. In this study, the driving energy E remains consistent throughout the experiments. The spark bubble ignites at a distance of h from the water surface and sphere. The radiated energy density on the water surface can be characterised as,

$$e \sim \frac{E}{4\pi h^2} \propto \hat{h}^{-2}. \quad (1)$$

where

$$\hat{h} = h/R_m.$$

The dimensionless depth of the spark bubble, \hat{h} , serves as a key dimensionless number reflecting the strength of the spark bubble.

The sphere set on the water surface is a unique component in this study. The interaction between the water and the sphere, including factors such as the sphere's radius, density, and contact angle, as well as the water's density, surface tension, and viscosity, represents a complex interplay. To simplify the consideration of the physical processes involved, we focus on the initial interaction of water surface with the sphere, specifically on whether a singular jet is generated during the spark bubble's initial expanding stage ($0 < t < \tau$).

The unimmersed portion of the floating sphere is $2r - h_{im}$, where h_{im} is the sphere's immersed depth. The characteristic upward moving speed of the water surface, $V_l = V_b/\hat{h}^2$, where $V_b = R_m/\tau$ is the characteristic velocity of the spark bubble. The characteristic upward moving speed of the sphere, $V_s = \rho^{-\frac{1}{2}}V_l$, where ρ is the relative density of the sphere. A 'chasing time', the duration required for the water surface to catch up with and cover the unimmersed portion of the floating sphere, can be expressed as $t_c^u = (2r - h_{im})/(V_l - V_s)$. Normalising the chasing time with τ , we have

$$t_c = \frac{t_c^u}{\tau} = \frac{(2 - \hat{h}_{im})\hat{r}}{1 - \rho^{-\frac{1}{2}}}\hat{h}^2,$$

where $\hat{h}_{im} = h_{im}/r$, and $\hat{r} = r/R_m$ denotes the radius ratio between the sphere and the spark bubble.

The dimensionless chasing time t_c serves as the other key dimensionless number, showing the dynamics of how the water surface and the sphere respond to the spark bubble during the initial phase.

We have included detailed definitions and analyses on \hat{h} and t_c in the ‘Modes evolution and distribution’ section of the revised manuscript.

Comment 2: In general γ is used to denote surface tension. It will be better to use another symbol for the non-dimensional height.

Thank you for your advice. The notation \hat{h} is used to replace the use of γ in the revision.

Comment 3: Page 2 line 77- spheres in the bracket may be omitted.

Thank you for the suggestion. We have removed the redundant word in the bracket.

Comment 4: Page 3 line 136- How are the sphere surfaces transformed to superhydrophobic surfaces?

The super-hydrophobic spheres were achieved by coating the surface with a thin layer of hydrophobic nanoparticles. The spheres were ultrasonically cleaned in acetone, ethanol, and deionised water for 30 minutes each. Following this, they were immersed in the coating liquid (Ultra Glaco, Soft99 Co., Japan). After coating, the spheres were taken out and left to dry naturally for 12 hours, resulting in super-hydrophobic surfaces. The procedures for treating sphere surfaces have been included in the Methods section.

Comment 5: Page 4 line 178-179: The deviation from the particulate free condition should be described briefly. Currently the “significant departure” is unable to emphasise the differences. Elaboration of the deviations will be helpful for the reader.

Thank you for your advice. In the revised manuscript, we have added several representative experimental snapshots as Figure 2. These snapshots include those from reference experiments without particles (Fig. 2a) and those illustrating the five jet modes with particles (Fig. 2b). Based on these figures, we have provided detailed descriptions of the different jet modes and emphasised their distinctions from the particle-free cases. Additionally, Figure 3 quantitatively illustrates the positions and velocities of jet tips and cavities in five jet modes and reference cases.

These additions in the revised manuscript help clarify the differences of five modes from that in particulate free conditions.

Comment 6: Pages 183- Please elaborate in two-three lines about the “ventilation” process or at least refer figure 3.

A snapshot at the instant the channel formed between the spark bubble and the atmosphere is shown in Mode III, Figure 2b in the revised manuscript.

For Mode III, the concavity on the water surface expands as the spark bubble collapses. Its bottom reaches the upper wall of the spark bubble, then forms a channel that bridges the bubble with the atmosphere. This channel facilitates aerodynamic interaction and creates a ventilation effect. These descriptions on the ventilation process have been included in the revised manuscript.

Comment 7: Figure 2- There seems to be no discussion on the effect of the contact angle of the particulate with the water. Please report if you have noticed any variation due to contact angle (in line 135 to 136 you have mentioned about the contact angles).

Thank you for pointing this out. In this study, the contact angle θ is an important parameter that determines the immersed depth h_{im} of the sphere. h_{im} reflects the balance among gravity, buoyancy force, and surface tension force on the sphere. It can be expressed as a function of the water’s density, surface tension, the sphere’s radius, density, and contact angle[2]. Figure 1c in the revised manuscript shows different h_{im} due to different θ .

h_{im} essentially affects the dimensionless chasing time t_c . We used spheres with wetting abilities from hydrophilic to super-hydrophobic (contact angles from 81° to 154°), and various radii ranging in $100 \sim 2500 \mu\text{m}$, relative densities between 1.4 and 8.5. These physical properties and sphere radii are integrated into t_c . All the data on different contact angles are summarised in the phase plot, as shown in Figure 5a of the revised manuscript.

Comment 8: Line 222 and 226- “substantially lowers” & “significantly reduce”- Please provide quantitative estimate and compare with particle free condition to emphasise your claims.

Thank you for your advice. We have included following analysis on this issue in the revised manuscript:

The jet formation on the water surface is driven by the underwater spark bubble. The mechanical energy of a spark bubble is determined by its volume and the driving pressure. Theoretically, it can be expressed as $E = \frac{4}{3}\pi R_m^3 (p_{atm} - p_v)$, where p_{atm} and p_v are the atmosphere pressure and the saturated vapour pressure of water, respectively[1]. In this

study, the driving energy E remains consistent throughout the experiments. The spark bubble ignites at a distance of h from the water surface and sphere. The radiated energy density on the water surface by the spark bubble can be characterised as,

$$e \sim \frac{E}{4\pi h^2} \propto \hat{h}^{-2}. \quad (2)$$

When a sphere is present on the water surface, a jet forms even as \hat{h} reaches 3.5, much greater than the typical upper boundary of $\hat{h} \approx 1.2$ for jet formation on a flat water surface[1, 3]. With the sphere, the energy density on the water surface required to generate a jet is roughly $\propto (3.5/1.2)^{-2} \approx 12\%$ of that required without a sphere, which is approximately an order of magnitude lower. The presence of particles significantly lowers the energy threshold required for jet formation, enhancing the system's sensitivity.

Comment 9: Page 271-273: “the importance ofroperties, surface dynamics, and bubble strength.”- Please elaborate how the chasing time is correlated with the given properties.

In this study, the dimensionless chasing time t_c is a key dimensionless number, showing the dynamics of how the water surface and the sphere respond to the spark bubble during the initial phase.

t_c is derived from the duration required for the water surface to catch up with and cover the unimmersed portion of the floating sphere. The details on how t_c is derived are given in the above response to Comment 1. It is influenced by several factors:

1. Sphere Properties: The radius, density, and contact angle of the sphere directly affect the immersed depth h_{im} . h_{im} determines t_c .
2. Surface Dynamics: The characteristic velocities of the water surface (V_l) and the sphere (V_s) are based on energy and volume conservation principles. These velocities are essential in determining the time t_c .
3. Spark Bubble Strength: The dimensionless depth of the spark bubble (\hat{h}) impacts the initial conditions of the spark bubble's expansion and collapse phases, is a part in the expression of t_c .

The detailed definition and analysis on t_c is included in the ‘Modes evolution and distribution’ section of the revised manuscript.

Comment 10: Page275-276: Subsequently you have bypassed the elaboration by mentioning “However, the phase plot only provides an initial understanding, and a deeper analysis is needed to fully grasp their combined effects.” For high impact journals like Nature Communications I think the deeper analysis is needed.

To comprehensively explore the sphere’s influence on jet dynamics, we conducted more experiments using spheres with varying properties. These spheres have the radii ranging in $100 \sim 2500 \mu\text{m}$, relative densities between 1.4 and 8.5, and wetting abilities from hydrophilic to super-hydrophobic (with contact angles from 81° to 154°).

With the additional experimental data, the resolution of the phase plot (Fig. 5a) for the two key dimensionless numbers \hat{h} and t_c has been highly improved, and the range of parameters has been broadened. The distributions of five jet modes and their boundaries are now clearly presented in the phase plot. Six boundary lines were identified compared to two in the previous version. We found that the horizontal line in previous manuscript actually consists of an oblique line and a horizontal line.

Furthermore, we have included a detailed analysis on physical mechanisms underlying the five jet modes. The evolving dynamics are modelled as a competition between the velocities and movements of the water surface and the sphere. We also considered the growth of concavities and the closure of cavity apertures, providing a more comprehensive understanding of how these processes determine jet modes, as shown in Figure 4 and 5b. The physics revealed here offer a more comprehensive understanding of how a sphere induced defect influences the resulting jet formations.

References

- [1] S. Zhang, S. P. Wang, and A. M. Zhang. Experimental study on the interaction between bubble and free surface using a high-voltage spark generator. *Physics of Fluids*, 28(3):032109, 2016.
- [2] D. Vella. Floating versus sinking. *Annual Review of Fluid Mechanics*, 47(1):115–135, 2015.
- [3] Y. J. Kang and Y. Cho. Gravity-capillary jet-like surface waves generated by an underwater bubble. *Journal of Fluid Mechanics*, 866:841–864, 2019.

Responses to Reviewers' Comments for Manuscript NCOMMS-24-09227-T

Particulate Reshapes Surface Jet Dynamics Induced by a Cavitation Bubble

Addressed Comments for Manuscript to

by

Xianggang Cheng, Xiao-Peng Chen, Zhi-Ming Yuan and Laibing Jia

Authors' Response to Reviewer 3

General Comments: Report to Manuscript NCOMMS-24-09227-T entitled “Particulates Reshape Jet Dynamics on Water Surface” by Cheng et al.

The authors reported the jet dynamics of a cavitation bubble near the water surface under the effect of a particle rest at the water surface. Depending on the distance of the cavitation bubble from the free surface and the particle density and wettability, the authors identified five different jetting modes, provided the corresponding regime map, and discussed the evolution of the jet, considering two dimensionless numbers, the dimensionless bubble depth and the dimensionless chasing time. The presence of the rest particle at the liquid surface significantly modifies the jet dynamics from the cavitation bubble, which not only yields interesting phenomena as reported by this manuscript, but indicates unexplored physics regarding the interplay between the particle dynamics and the cavitation bubble dynamics.

This manuscript is organized well, and presents new phenomena with significant impact over a wide range of natural and industrial processes. However, the authors mainly described and discussed different jet phenomena without enough physical insights and theoretical analyses. I thus suggest the authors to improve the theoretical part of the manuscript, as I suggest in the below comments.

Thank you for your insightful feedback and constructive comments. In the revised manuscript, we have made the following major changes:

1. The title was changed from ‘Particulates Reshape Jet Dynamics on Water Surface’ to ‘Particulate Reshapes Surface Jet Dynamics Induced by a Cavitation Bubble’ to emphasise the role of cavitation bubbles in the research.
2. A more thorough theoretical analyses on the key dimensionless numbers, the surface defect evolution underlying the five jet modes, and the boundaries between different jet modes have been analysed.
3. The key dimensionless parameters \hat{h} (dimensionless spark bubble depth) and t_c (dimensionless chasing time) were introduced in the main text, explaining their influence on the formation and dynamics of jets. (Notation \hat{h} replaced γ to avoid confusion with surface tension)
4. More experimental data was added in the updated phase diagram (Figure 5a) to systematically categorise the jet modes based on the dimensionless parameters.
5. Quantitative measurements and comparisons of results with and without particulates were presented in Figure 3 to show quantitative differences among jet modes, highlighting the differences in their kinematics.

6. Expanded details on the size, physical properties, and preparation methods of the spheres are provided in the Results and Methods sections.
7. Experimental snapshots of different jet modes were added as Figure 2, providing a direct view of the experimental results in different modes.
8. The jet shapes at $t = T_1 + T_2$ are added in Figures 2 and 4, and descriptions are added in the main text of the revised manuscript to reflect a longer-term evolution of the jets.
9. New references to studies on underwater particles driven by cavitation bubbles and sea spray aerosol (Wu et al. [2017], Ren et al. [2022], Sha et al. [2024]) were added in the introduction section.

Following are our point-by-point responses to your comments.

Comment 1: I suggest to revise the title of the manuscript to describe this study more precise. Here is a suggestion: “Particulates reshape the jet dynamics of a cavitation bubble near a water surface”.

Thank you for the advice. The title of this manuscript has been changed to ‘Particulate Reshapes Surface Jet Dynamics Induced by a Cavitation Bubble’.

Comment 2: There are some studies discussing the interplay between the cavitation bubble and the particle, such as Ren, PRL, 2022, 128: 044501 and Wu, PRL, 2017, 119: 084501. Though these studies did not involve a free water surface, the theoretical analyses on the interplay between the cavitation dynamics and the particle dynamics may help.

Thank you for recommending these relevant papers. These studies introduce two scenarios of cavitation bubble and underwater particle interactions: liquid inertia driven and bubble contact driven. They provide valuable insight for our research on how cavitation bubbles interact with particles on the water surface. Following their definition, our study focuses on the liquid inertia driven one during the spark bubble expanding phase. We have included an introduction of these studies in the literature review section.

Comment 3: Line 131-133: “. . . spheres of varying sizes, ranging from 100 μm to 2.5 mm in radius, exhibited distinct phenomena compared to a clean water surface”. I am confused with this sentence. Which phenomenon is distinct? The authors stated that they used 500 μm particles in the following sentence. Did they really fixed the particle size?

In the revised manuscript, we have added several representative experimental snapshots as Figure 2. These snapshots include those from reference experiments without particles (Fig. 2a) and those illustrating the five jet modes with particles (Fig. 2b) to show differences between them. Based on these figures, we have provided detailed descriptions of the different jet modes and emphasised their distinctions from the particle-free cases. Figure 3 quantitatively illustrates the positions and velocities of jets and cavities in five jet modes and reference cases. These additions in the revised manuscript help clarify the differences of five modes from that in particulate free conditions.

In our previous experiments, we tested some spheres with radii ranging from 100 μm to 2.5 mm to ensure the phenomena were applicable to a wide range of spheres. However, our primary focus was using spheres with a radius of $r = 500 \mu\text{m}$. The data shown in Fig. 2 of the previous version were drawn solely from experiments using $r = 500 \mu\text{m}$ spheres.

To thoroughly investigate the influence of sphere radius on jet dynamics, we conducted systematic experiments using spheres with different radii, ranging from 100 μm to 2.5 mm. These spheres have relative densities between 1.4 and 8.5, and wetting abilities from hydrophilic to super-hydrophobic (with contact angles from 81° to 154°). The results are presented in Fig. 5a of the revised manuscript. Following a reviewer’s suggestion, we have replaced the notation γ with \hat{h} to denote the dimensionless depth of the spark bubble, to avoid confusion with the symbol for surface tension.

With the additional experimental data, the resolution of the phase plot for the two key dimensionless numbers \hat{h} and t_c has been highly improved, and the range of parameters has been broadened. The distributions of the five jet modes and their boundaries are now clearly presented in the phase plot. We identified six boundary lines compared to two in the previous version. We found that the horizontal line in the previous manuscript actually consists of an oblique line and a horizontal line. These boundaries have been analysed with respect to the evolution of the water surface in the revised manuscript.

Comment 4: Can the authors indicate the particle materials for each data points in Fig. 2, using different point shapes or colors?

Thank you for the suggestion. In the revised phase plot shown in Fig. 5a, we use varying colour saturation to represent spheres’ densities, different symbol interiors to indicate their wetting abilities (contact angles), and varying symbol sizes to denote their radii.

Comment 5: Some important information in the Methods part should be removed to the main text. For example, the fixed maximum bubble size. The definition and

the physical meaning of the two dimensionless numbers are crucial for the readers to understand the regime map and jet evolution schematics.

We have reorganised the manuscript to introduce the descriptions and definitions in a logical sequence. Several key pieces of information, including the fixed maximum radius of the spark bubble, the definitions of the two key dimensionless numbers, \hat{h} and t_c , as well as the definitions of the five jet modes, have now been incorporated into the main text.

Comment 6: The authors identified the jetting modes based on the jet feature at $t = T_1$. However, I think the global features of the jet are more important and more reasonable. For example, in figure 3, it seems the cavity in mode V will collapse after T_1 and may generate a jet, showing a global feature same as Mode IV. I suggest the authors to check the jet dynamics during the whole jetting process instead of only an oscillation period.

We have re-examined the jet dynamics over the entire jetting process for all five modes. Specifically, we analysed the evolution of the jet in the first two oscillation periods, including subsequent cavity collapses, jet and crown formations. The jet shapes at $t = T_1 + T_2$ have been added in Figures 2 and 4. The corresponding descriptions and analyses for the five jet modes have been added in the revised manuscript to reflect a longer-term evolution of the jets.

The main features of the jets are exhibited in the first two oscillation periods. Further evolution over an even longer term results in possible outcomes of instability and eventual breakup. The discussion have been included in ‘Jet modes’ section to provide a comprehensive understanding of the jet dynamics.

Comment 7: I suggest the authors to give some typical time sequences from experiments to support the schematics in Fig. 3.

We have added several typical experimental snapshots at $t = \tau$, T_1 , and $T_1 + T_2$, as shown in Fig. 2 of the revised manuscript. These snapshots display the five jet modes with particles (Fig. 2b) and the reference groups without particles (Fig. 2a). These figures, along with their corresponding descriptions, have been integrated into the revised manuscript to provide a more comprehensive understanding of the diverse jet modes.

Comment 8: The dimensionless chasing time is not a parameter that apparently decided by the initial parameters of the system, which makes it inconvenience to use in practical situations for the prediction and control of the jet morphology. I prefer dimensionless numbers based on the initial parameters of the system, with clear physical meanings.

Why not using dimensionless numbers that represent the comparison of different forces, such as the inertial, viscous, and capillary forces, etc.?

Thank you for your comments. We appreciate your concern regarding the practical use of dimensionless parameters for predicting and controlling jet mode. In this study, the spark bubble's maximum radius, R_m , the duration for the spark bubble to reach its maximum radius, τ , and the density of water, ρ_l , are chosen as characteristic parameters. Two key dimensionless numbers were employed: the spark bubble's dimensionless depth \hat{h} and the dimensionless chasing time t_c . Both of these parameters are essentially determined by the initial physical parameters of the system.

The dimensionless depth of the spark bubble is defined as

$$\hat{h} = h/R_m,$$

where h denotes the initial depth of the spark bubble centre, and R_m is the maximum radius of the spark bubble. Under the condition of a fixed energy input E of the spark, R_m is found to be a constant ($R_m \propto E^{1/3}$, see reference [1]). This suggests that R_m is also an initial parameter of the system, and \hat{h} is a dimensionless parameter that is based on the initial parameters of the system.

The sphere setting on the water surface is a unique component in this study. Here we define a 'chasing time', the duration required for the water surface to catch up with and cover the unimmersed portion of the floating sphere, $t_c^u = (2r - h_{im})/(V_l - V_s)$, where r is the sphere radius, h_{im} is the initial immersed depth of the sphere, V_l and V_s are the characteristic velocities of the water surface and the sphere, respectively. h_{im} is an initial parameter of the system, which is determined by the sphere's radius, density and contact angle, and the water's density and surface tension[2]. Based on energy and volume conservation, V_s and V_l can be estimated as $V_s = \rho^{-\frac{1}{2}}V_l$, $V_l = V_b/\hat{h}^2$. V_b is the characteristic velocity of the spark bubble, $V_b = R_m/\tau$. By normalising with the time scale of the spark bubble, τ , a dimensionless chasing time is obtained:

$$t_c = \frac{t_c^u}{\tau} = \frac{(2 - \hat{h}_{im})\hat{r}}{1 - \rho^{-\frac{1}{2}}}\hat{h}^2,$$

where \hat{h}_{im} represents the dimensionless immersed depth of the sphere, $\hat{h}_{im} = h_{im}/r$; and \hat{r} denotes the radius ratio between the sphere and the spark bubble, $\hat{r} = r/R_m$. t_c depends on the sphere's radius r , relative density ρ , initial immersed depth h_{im} , and the spark bubble's radius R_m and depth h . Therefore, t_c is a also dimensionless parameter that is based on the initial parameters of the system, and has clear physical meanings.

The definition and analyses of the key dimensionless numbers are given in the main text of the revised manuscript.

Comment 9: I am wondering that if the two dimensionless numbers are enough to describe the dynamics of the jet dynamics under the influence of particle. For example, the initial configuration of the particle are considered in the dimensionless chasing time, by involving h_{im} . However, the particle density and wettability also significantly affect the motion of the particle at interface (Ji, Chemical Engineering Science, 2019, 207: 17-29; Ding JFM, 2015, 783: 504-525), and thus may change the dynamics of the cavity and jet.

Thank you for your valuable feedback and constructive suggestions. In this study, the process is dominated by inertial forces and can be categorised as liquid inertia driven. This simplifies the analysis of the physical processes.

In the initial expanding phase of the spark bubble, the evolution of the surface defect is primarily affected by the velocity difference between the solid and liquid, which determines whether a singular jet is formed. During the collapsing phase of the spark bubble, inertial effects are mainly reflected in two aspects: 1) the growth of the cavity under the impact of the collapsing spark bubble, and 2) the pressure difference across the cavity rim induced by the airflow rushing into the cavity.

With these perspectives on the physical processes, we proposed two key dimensionless numbers: the spark bubble's dimensionless depth \hat{h} and the dimensionless chasing time t_c . \hat{h} reflects the strength of the spark bubble, while t_c captures the competitive movement between the water surface and the sphere during the initial expanding phase of the spark bubble. These parameters capture the key dynamics of the process and provide a reasonable explanation for the observed phenomena.

The sphere's density and wetting ability are accounted for in h_{im} , which further affects t_c . The contribution of surface tension is only considered in h_{im} and not in other aspects of surface evolution, limiting the applicability of \hat{h} and t_c for small radii spheres. These limitations have been included in the revised manuscript.

Comment 10: My major concern is that there are no quantitative analyses (even simple scaling) on the experimental findings. Taking the regime map as an example, the authors can derive some theoretical scaling to describe the boundaries between different modes. The characteristic parameters of the jet, such as size, height, and velocity are also deserved to study, and the related theoretical models can be proposed. Of course, all the above suggestions rely on critical physical insight of the underlying physics regarding

the interplay between the particle motion and the flow field generated by the cavitation bubble, which is lacked in the present form and should be included in the revision.

Thank you for your constructive feedback. In the revised manuscript, we have included quantitative measurements and analysis on the jet and cavity kinematics, and theoretical analyses on the five jet modes and their boundaries.

We have included measurements and analyses of the featured geometries of the jets and cavities from typical examples of the five jet modes. Their positions and velocities are illustrated in Figure 3 of the revised manuscript. For comparison, the results of jet/hump formation from flat water surfaces at the same \hat{h} are also presented. Based on Figure 3, we observed that the presence of particles significantly alters jet development, which leads to increased variability in jet types and enhanced jet velocity.

We have included further theoretical analyses on the five jet modes and their boundaries, and conducted systematic experiments using spheres with different radii, densities, and wetting abilities. Based on dimensional analysis, we identified two key dimensionless numbers, the dimensionless depth of the spark bubble (\hat{h}) and the dimensionless chasing time (t_c). The five jet modes and their boundaries are presented in the phase plot of these parameters (Figure 5a of the revised manuscript). Six boundary lines were identified compared to two in the previous version. We found that the horizontal line in previous manuscript actually consists of an oblique line and a horizontal line.

Furthermore, we have included a detailed analysis on physical mechanisms underlying the five jet modes. The evolving dynamics are modelled as a competition between the velocities and movements of the water surface and the sphere. We also considered the growth of concavities and the closure of cavity apertures, providing a more comprehensive understanding of how these processes determine jet modes, as shown in Figure 4 and 5b. The physics revealed here offer a more comprehensive understanding of how a sphere induced defect influences the resulting jet formations.

We hope these additions will help to enhance the understanding of the diverse jet modes and improve the quality of the manuscript.

References

- [1] S. Zhang, S. P. Wang, and A. M. Zhang. Experimental study on the interaction between bubble and free surface using a high-voltage spark generator. *Physics of Fluids*, 28(3):032109, 2016.
- [2] D. Vella. Floating versus sinking. *Annual Review of Fluid Mechanics*, 47(1):115–135, 2015.

Responses to Reviewers' Comments for Manuscript NCOMMS-24-09227-T

Particulate Reshapes Surface Jet Dynamics Induced by a Cavitation Bubble

Addressed Comments for Manuscript to

by

Xianggang Cheng, Xiao-Peng Chen, Zhi-Ming Yuan and Laibing Jia

Authors' Response to Reviewer 4

General Comments: The authors investigate liquid jet formation in presence of a solid particle floating on the liquid surface. While previous studies have been conducted with particles of nano or micrometric size, the authors aim to extend the problem to macroscopic particles. By means of an oscillating bubble triggering the jet formation and spheres of varying properties (density and contact angle), the authors show that the jet shape and dynamics are affected by the floating sphere. In particular, they describe the five type of jets that can emerged depending on the initial configuration and the energy transferred by the spark bubble. The key parameters governing the problem are compiled in two characteristics quantities whose ratio determines the type of jet. In addition, a tentative qualitative explanation for the mechanisms leading to each type of jet is provided.

While these experimental observations are interesting, the paper lacks clarity and quantitative results that would support its claims, which limits the broad impact of this study. In the present form, the paper can not be recommended for publication.

Below are detailed major concerns as well as minor comments.

Thank you for your valuable feedback and constructive comments. In the revised manuscript, we have made the following major changes:

1. The title was changed from 'Particulates Reshape Jet Dynamics on Water Surface' to 'Particulate Reshapes Surface Jet Dynamics Induced by a Cavitation Bubble' to emphasise the role of cavitation bubbles in the research.
2. A more thorough theoretical analyses on the key dimensionless numbers, the surface defect evolution underlying the five jet modes, and the boundaries between different jet modes have been analysed.
3. The key dimensionless parameters \hat{h} (dimensionless spark bubble depth) and t_c (dimensionless chasing time) were introduced in the main text, explaining their influence on the formation and dynamics of jets. (Notation \hat{h} replaced γ to avoid confusion with surface tension)
4. More experimental data was added in the updated phase diagram (Figure 5a) to systematically categorise the jet modes based on the dimensionless parameters.
5. Quantitative measurements and comparisons of results with and without particulates were presented in Figure 3 to show quantitative differences among jet modes, highlighting the differences in their kinematics.

6. Expanded details on the size, physical properties, and preparation methods of the spheres are provided in the Results and Methods sections.
7. Experimental snapshots of different jet modes were added as Figure 2, providing a direct view of the experimental results in different modes.
8. The jet shapes at $t = T_1 + T_2$ are added in Figures 2 and 4, and descriptions are added in the main text of the revised manuscript to reflect a longer-term evolution of the jets.
9. New references to studies on underwater particles driven by cavitation bubbles and sea spray aerosol (Wu et al. [2017], Ren et al. [2022], Sha et al. [2024]) were added in the introduction section.

Following are our point-by-point responses to your comments.

Comment 1: The paper is confusing and needs a thorough reorganisation before being accepted for publication. Descriptions and definitions are given in different parts of the paper and methods, and one need to look for information in many places to make sense of the paper. In particular:

1. the different jet modes need to be described as they are presented. Fig.3 is a good guide but should arrive before the diagram Fig.2.
2. line 183, it is unclear what "ventilation" is; in general more explanations are needed when terms are introduced.
3. the definition of the chasing time t_c , which contains all the physics of the problem, needs to be better described and in the main text.
4. line 158: it is unclear at that point why the temporal parameters are fundamentals in the study. Furthermore, while the characteristic time of the spark bubble τ appears in t_c , it is not the case of the first oscillation period T_1 , which is then absent from the phase diagram Fig.2.

Thank you for your thorough review and constructive feedback. We have reorganised the manuscript to ensure that descriptions and definitions are introduced in a logical sequence.

1. We have added several typical experimental snapshots of the five jet modes, as shown in Figure 2 of the revised manuscript. Based on the experimental snapshots, we have provided detailed descriptions of the different jet modes. The schematic of the five jet modes (Figure 4 in the revised manuscript) is moved in front of the phase plot (Figure 5a in the revised manuscript).

2. A snapshot at the instant the channel forms between the spark bubble and the atmosphere is shown in Mode III, Figure 2b in the revised manuscript. For Mode III, the concavity on the water surface expands as the spark bubble collapses. Its bottom reaches the upper wall of the spark bubble, then forms a channel that bridges the bubble with the atmosphere. This channel facilitates aerodynamic interaction and creates a ventilation effect. These descriptions of the ventilation process have been included in the revised manuscript.

3. The definition of t_c has been moved into the main text of the revised manuscript.

4. Thank you for pointing this out. The spark bubble ignites at $t = 0$, then expands to its maximum radius R_m at $t = \tau$, and collapses to its minimum volume at $t = T_1$. T_1 denotes the spark bubble's first oscillation period, and τ is the spark bubble's expansion duration during T_1 . Ideally, $\tau \propto R_m$ and $T_1 = 2\tau$, based on the Rayleigh-Plesset equation for spherical bubbles[1]. We choose R_m as the characteristic length, which reflects the bubble energy. The corresponding timescale τ is selected as the characteristic time.

In the previous manuscript, we aimed to emphasise the surface development during the spark bubble's expanding stage ($t < \tau$) and collapsing stage ($\tau < t < T_1$). The statement has been replaced by a more in-depth analysis of t_c in the 'Modes evolution and distribution' section.

Comment 2: The authors claim that the presence of the particle lowers the energy needed for jet formation. There are no measurement in the paper that support this claim. In general, there are no quantitative measurements. In particular:

1. it is said that the amplitude of the jet varies with the dimensionless depth of the spark bubble γ (line 168), but no graph is provided. Giving a plot of the amplitude vs γ , both for a particle-free interface and with a particle would be interesting. Furthermore, such a plot would be useful to support the fact that "A pivotal finding of our study is the notable impact of spheres on the water surface, significantly extending the effective range of γ from typically up to 1.2 on a flat water surface to values surpassing 4" and that particles "substantially lowers the energy requirement for jet generation". Here only qualitative comparison are provided in the supplementary videos.

2. Measurements (e.g. evolution of the radius of the cavity or jet height with time) could be provided to better describe the different modes, in particular to support sentences like "the modes III to V depend substantially on the bubble depth γ . A smaller γ results in more rapid expansion of the cavity" (line 303).

Thank you for your constructive suggestions.

1. The jet formation on the water surface is driven by the underwater spark bubble. The mechanical energy of a spark bubble, E , is determined by its volume and the driving pressure. Theoretically, $E = \frac{4}{3}\pi R_m^3 (p_{atm} - p_v)$, where p_{atm} and p_v are the atmosphere pressure and the saturated vapour pressure of water, respectively[2]. In this study, the driving energy E remains consistent throughout the experiments. The spark bubble ignites at a distance of h from the water surface and sphere. The radiated energy density on the water surface by the spark bubble can be characterised as,

$$e \sim \frac{E}{4\pi h^2} \propto \hat{h}^{-2}. \quad (1)$$

When a sphere is present on the water surface, a jet forms even as \hat{h} reaches 3.5, exceeding the typical upper boundary of $\hat{h} \approx 1.2$ for jet formation on a flat water surface[2, 3]. With the sphere, the energy density on the water surface required to generate a jet is roughly $\propto (3.5/1.2)^{-2} \approx 12\%$ of that required without a sphere, which is approximately an order of magnitude lower.

The analysis on the energy reduction for jet generation with the presence of particulates has been incorporated into the ‘Modes evolution and distribution’ section of the revised manuscript.

2. We have included quantitative measurements and analysis on the featured geometries of the jets and cavities from typical examples of the five jet modes. Their positions and velocities are illustrated in Figure 3 of the revised manuscript. For comparison, the results of jet/hump formation from flat water surfaces are also presented as references.

Comment 3: In many places there is too much emphasis without substantial results.

1. line 76: why is the approach "unique and innovative" ? How does this experimental setup allows to do something that has never be done before ? The spark bubble is not a novel method to produce jets (please refer to ref. 17 here).

2. line 321: "not merely academic, it has tangible implications in various fields", line 333 "contributes to the advancements of fluid dynamics", line 335 "unlocks the potential for jet manipulations", "have far-reaching impacts" are all overselling; either develop, or provide at least one clear substantial result that support this. Currently the paper, while being a nice experimental study, remains a qualitative description of an interesting phenomenon in a specific range of parameters without full insight into physical mechanisms or modelling.

Thank you for your constructive feedback. We have examined the manuscript and removed the overgeneralised statements.

1. Although the method of using a spark bubble to produce jets is not new, this study is the first to examine the interaction between a surface defect induced by a floating particle and an underwater spark bubble. This approach offers insights into jet formation mechanisms that have not been explored in previous research. The observed process indicates that a great diversity of natural events can be caused by a tiny initial perturbation.

2. The overgeneralised statements have been removed. In the revised manuscript, we have included further theoretical analyses on the five jet modes and their boundaries, and conducted systematic experiments using spheres with different radii, densities, and wetting abilities.

Based on dimensional analysis, we identified two key dimensionless numbers, including the dimensionless depth of the spark bubble (\hat{h}) and the dimensionless chasing time (t_c). Based on these two parameters, the five jet modes and their boundaries are presented in the phase plot (Figure 5a of the revised manuscript). We identified six boundary lines compared to two in the previous version. We found that the horizontal line in the previous manuscript actually consists of an oblique line and a horizontal line.

Furthermore, we have included a detailed analysis on physical mechanisms underlying the five jet modes. The evolving dynamics are modelled as a competition between the velocities and movements of the water surface and the sphere. We also considered the growth of concavities and the closure of cavity apertures, providing a more comprehensive understanding of how these processes determine jet modes, as shown in Figure 4 and 5b. The physics revealed here offer a more comprehensive understanding of how a sphere induced defect influences the resulting jet formations.

Comment 4: l. 115 : The dotted lines are hardly visible (maybe better in red ?)

We have changed the colour of the dotted lines to red and increased their thickness to make them more distinguishable (Figure 1c of the revised manuscript).

Comment 5: l. 132 : It says that particles of different sizes are used while both in methods and figures only particles of 500 microns are used.

Thank you for your comments. In our previous experiments, we tested some spheres with radii ranging from 100 μm to 2.5 mm to ensure the phenomena were applicable to a wide range of spheres. However, our primary focus was using spheres with a radius of $r = 500 \mu\text{m}$. The data shown in Figure 2 of the previous version were drawn from the data obtained using $r = 500 \mu\text{m}$ spheres only.

To thoroughly investigate the influence of sphere radius on jet dynamics, we conducted additional experiments using spheres with various properties. These spheres had radii ranging from 100 to 2500 μm , relative densities between 1.4 and 8.5, and wetting abilities from hydrophilic to super-hydrophobic (with contact angles from 81° to 154°). The results are presented in Figure 5 of the revised manuscript.

Comment 6: l. 176 : "While this study primarily focuses on the dynamics of liquid jet formation within the spark bubble's first oscillation period." Rephrase/ remove "while" ?

Thank you for your advice. We have revised the sentence as you suggested.

Comment 7: l. 201 : Caption of fig. 2: Add the meaning of the dashed lines. Recall what γ and t_c represent ?

We have revised the caption of the phase diagram as you suggested (Figure 5a of the revised manuscript).

Comment 8: l. 201 : Fig. 2 Are there any error bars on the measurements ? Are the experiments repeated ?

We conducted numerous experiments to ensure the repeatability of our observations. Each data point in the phase diagram (Figure 5a) represents the result of a single experiment. In Figure 3 of the revised manuscript, we have included error bars to represent the experimental uncertainties. Each working condition was repeated at least three times, and the featured geometries of the jets and cavities from typical examples of the five jet modes are presented, along with their positions and velocities.

Comment 9: l. 433 : "Studies on cavitation bubbles reveal that the force is significantly greater at the onset of expansion and at the end of collapse compared to other times". Could you add some references ?

Thank you for the suggestion. We have added the appropriate citation to this statement.

Added citation: J. P. Franc and J. M. Michel. *Fundamentals of Cavitation*, page p.36-39. Springer, Dordrecht, 2005.

References

- [1] J. P. Franc and J. M. Michel. *Fundamentals of Cavitation*, pages p.36–39. Springer, Dordrecht, 2005.

- [2] S. Zhang, S. P. Wang, and A. M. Zhang. Experimental study on the interaction between bubble and free surface using a high-voltage spark generator. *Physics of Fluids*, 28(3):032109, 2016.
- [3] Y. J. Kang and Y. Cho. Gravity-capillary jet-like surface waves generated by an underwater bubble. *Journal of Fluid Mechanics*, 866:841–864, 2019.

Responses to Reviewers' Comments for Manuscript NCOMMS-24-09227-T

Particulate Reshapes Surface Jet Dynamics Induced by a Cavitation Bubble

Addressed Comments for Manuscript to

by

Xianggang Cheng, Xiao-Peng Chen, Zhi-Ming Yuan and Laibing Jia

Authors' Response to Reviewer 5

General Comments: I co-reviewed this manuscript with one of the reviewers who provided the listed reports. This is part of the Nature Communications initiative to facilitate training in peer review and to provide appropriate recognition for Early Career Researchers who co-review manuscripts.

Thank you for sharing this information. We appreciate your involvement in the peer review process and your contributions to improving the quality of the manuscript. We value the insights and feedback provided by you and your co-reviewer, and have made point-to-point revisions to the manuscript based on the suggestions.

Responses to Reviewers' Comments for Manuscript NCOMMS-24-09227A

Particulate Reshapes Surface Jet Dynamics Induced by a Cavitation Bubble

Addressed Comments for Manuscript to

by

Xianggang Cheng, Xiao-Peng Chen, Zhi-Ming Yuan and Laibing Jia

Authors' Response to Reviewer 1

General Comments: The revised manuscript have addressed the comments that I have given. Further, I can see that new figures with results from additional experiments have really improved the presentation and also showcases a deeper analysis which was missing in their previous version. Also the authors have taken time to explain the different time lines in their experiment which is indeed welcome.

Thank you for your thoughtful review and positive feedback on our revised manuscript. Your constructive comments has significantly enhanced the quality of the manuscript.

In the revised manuscript, we have made the following major changes:

1. The dynamic wetting process of the sphere during immersion has been analysed, considering sphere surface wettabilities and wetting failure phenomena. Additionally, a thorough theoretical analysis of the sphere's dimensionless immersion time t_c has been included.
2. Quantitative measurements and theoretical predictions, including water surface velocity and cavity expansion velocity, have been incorporated. Cavity dynamics, primarily influenced by \hat{h} (the spark bubble's dimensionless depth), were identified as the governing factor in the second branching process of jet modes.
3. Building on insights from the two branching processes of jet modes, more detailed theoretical analyses of the boundaries between different jet modes were conducted.
4. Figure 5b of the previous manuscript was integrated into Fig. 4 to better illustrate the two branching processes in the evolution of the jet modes.
5. Jet mode III has been renamed from 'Ventilation' to 'Cavity Venting' to better represent the process in which airflow from the atmosphere flows through the cavity-induced channel into the underlying spark bubble as advised by a reviewer.
6. A dimensional analysis to identify the governing dimensionless parameters and a theoretical analysis of the sphere dynamics to derive its characteristic velocity are added in the Methods section.

Following are our detailed responses to your comments.

Comment 1: The authors have commented on the impact of not accounting for surface tension in their analysis which has impacted their results particularly for small radius sphere. While, I would have ideally preferred a detailed analysis with adding the surface tension terms in their analysis, which would have modified the boundaries in Figure 5. But they have presented an argument on dimensionless numbers, which also present a

form of explanation highlighting the impact of surface tension. Overall I am fine with the modifications.

Thank you for your constructive comments. In response to your concerns, we have included a detailed force analysis of the sphere during its immersion process. The derived theoretical expression for the sphere velocity now accounts for both liquid inertia and surface tension effects. Following is the detailed analysis of the sphere dynamics.

Through dimensional analysis, we simplify the system in this study into seven dimensionless parameters, which are the top seven dimensionless numbers listed in Tab. 1. They are the Weber number We , representing the ratio of liquid inertia to capillary force, the Froude number Fr , which compares liquid inertial to gravitational force, the Reynolds number Re , indicating the ratio of liquid inertia to viscous force, the density ratio ρ , the contact angle θ , the radius ratio between the sphere and the spark bubble, \hat{r} , and a dimensionless depth, h^* . The table also includes the Bond number $Bo = We/Fr^2$, which compares gravitational to capillary force, as well as the spark bubble's dimensionless depth, $\hat{h} = h/R_m = h^*\hat{r}$, a parameter commonly used in studies on cavitation bubble interactions with flat water surfaces[1, 2]. In this study, we use \hat{h} instead of h^* .

The ranges of these dimensionless parameters are summarised in Tab. 1. The Reynolds number ranges from 10^2 to 10^4 , indicating that the inertial force significantly outweighs the viscous force, and the viscous force can be neglected. The Froude number $Fr > 30$ suggests that the inertial force dominates over the gravitational force. The Weber number ranges from 10^0 and 10^3 . At lower Weber numbers, surface tension effects become pronounced and should be considered, whereas at higher values, inertial forces dominate and surface tension can be neglected. The Bond number $Bo < 1$, suggesting that gravitational and buoyancy forces are negligible compared to capillary force[3, 4, 5, 6].

The sphere's motion equation in the vertical direction can be expressed as,

$$(m_s + m_a)a_s = f_d + f_c + f_v + f_g + f_b, \quad (1)$$

where m_s is the sphere's mass, m_a is the added mass, and a_s is the sphere's acceleration. The terms f_d , f_c , f_v , f_g , and f_b represent the form drag, capillary force, viscous force, gravitational force, and buoyancy force, respectively.

Based on the dimensional analysis above, we find that the liquid inertia and capillary force are dominant forces for the sphere. Therefore, by neglecting f_v , f_g , and f_b , and defining a hydrodynamic force $f_h = f_d - m_a a_s$ following previous studies[6, 7, 8], we obtain,

$$m_s a_s = f_h + f_c. \quad (2)$$

Dimensionless numbers	Symbol	Definition	Range
Weber number	We	$\frac{\rho_l V_l^2 r}{\sigma}$	2-6126
Froude number	Fr	$\frac{V_l}{\sqrt{gr}}$	30-412879
Reynolds Number	Re	$\frac{\rho_l V_l r}{\mu}$	112-14938
Density ratio	ρ	$\frac{\rho_s}{\rho_l}$	1.4-8.5
Contact angle	θ	θ	$81^\circ - 154^\circ$
Radius ratio	\hat{r}	$\frac{r}{R_m}$	0.01-0.27
Dimensionless depth	h^*	$\frac{h}{r}$	6-268
Bond number	Bo	$\frac{\rho_l r^2 g}{\sigma} = \frac{We}{Fr^2}$	0.001-0.8
Spark bubble's dimensionless depth	\hat{h}	$\frac{h}{R_m} = h^* \hat{r}$	0.6-3.5

Table 1: Relevant dimensionless parameters and their characteristic values.

The hydrodynamic force f_h can be represented as[6, 7, 8]

$$f_h = \frac{\pi}{2} C_h \rho_l V_l^2 r^2, \quad (3)$$

where C_h is the hydrodynamic force coefficient. For liquid flow around the sphere with $Re > 10^2$, it can be assumed as potential flow, with C_h being of order unity[9, 10, 7, 8]. The capillary force is influenced by the wettability of the sphere[5], with a characteristic value of

$$f_c = \pi r (1 - \cos \theta) \sigma. \quad (4)$$

By substituting f_h and f_c into Eq. 2, we can give the scale of sphere acceleration:

$$a_s \sim (C_h + 4 \sin^2 \frac{\theta}{2} We^{-1}) \rho^{-1} \frac{V_l^2}{r}. \quad (5)$$

Further, with the time scale for sphere acceleration, $\Delta t \sim r/V_l$, the characteristic velocity of the sphere is obtained:

$$V_s \sim a_s \Delta t \sim (C_h + 4 \sin^2 \frac{\theta}{2} We^{-1}) \rho^{-1} V_l, \quad (6)$$

where the first term in the brackets corresponds to the influence of liquid inertia, and the second term represents the effect of surface tension. When $We \gg 1$, surface tension effects can be neglected, resulting in $V_s \sim C_h \rho^{-1} V_l$.

References

- [1] S. Zhang, S. P. Wang, and A. M. Zhang. Experimental study on the interaction between bubble and free surface using a high-voltage spark generator. *Physics of Fluids*, 28(3):032109, 2016.
- [2] Y. J. Kang and Y. Cho. Gravity-capillary jet-like surface waves generated by an underwater bubble. *Journal of Fluid Mechanics*, 866:841–864, 2019.
- [3] D. G. Lee and H. Y. Kim. Impact of a superhydrophobic sphere onto water. *Langmuir*, 24(1):142–145, 2008.
- [4] B. Ji, Q. Song, and Q. Yao. Numerical study of hydrophobic micron particle’s impaction on liquid surface. *Physics of Fluids*, 29(7), 2017.
- [5] H. Chen, H. Liu, X. Lu, and H. Ding. Entrapping an impacting particle at a liquid–gas interface. *Journal of Fluid Mechanics*, 841:1073–1084, 2018.
- [6] B. Ji, Q. Song, A. Wang, and Q. Yao. Critical sinking of hydrophobic micron particles. *Chemical Engineering Science*, 207:17–29, 2019.
- [7] B. Ji, Q. Song, K. Shi, J. Liu, and Q. Yao. Oblique impact of microspheres on the surface of quiescent liquid. *Journal of Fluid Mechanics*, 900, 2020.
- [8] B. Ji, Z. Tang, and Q. Song. Oblique impact dynamics of micron particles onto a liquid surface. *Physical Review Fluids*, 5(11), 2020.
- [9] M. Shiffman and Donald C. Spencer. The force of impact on a sphere striking a water surface. AMP Technical Reports 42. 1R, 42. 2R, AMG-NYU Nos. 105, 133, 1945.
- [10] M. Moghisi and P. T. Squire. An experimental investigation of the initial force of impact on a sphere striking a liquid surface. *Journal of Fluid Mechanics*, 108:133–146, 1981.

Responses to Reviewers' Comments for Manuscript NCOMMS-24-09227A

Particulate Reshapes Surface Jet Dynamics Induced by a Cavitation Bubble

Addressed Comments for Manuscript to

by

Xianggang Cheng, Xiao-Peng Chen, Zhi-Ming Yuan and Laibing Jia

Authors' Response to Reviewer 2

General Comments: I think the authors have done substantial revisions to the revised manuscript. I am happy with the changes incorporated this time. I think the article can be accepted now.

Thank you for your positive feedback and recommendation. We are pleased that the revisions have addressed your concerns and that you are satisfied with the changes. We appreciate for your thorough review and valuable suggestions, which have significantly improved the quality of our manuscript.

In the revised manuscript, we have made the following major changes:

1. The dynamic wetting process of the sphere during immersion has been analysed, considering sphere surface wettabilities and wetting failure phenomena. Additionally, a thorough theoretical analysis of the sphere's dimensionless immersion time t_c has been included.
2. Quantitative measurements and theoretical predictions, including water surface velocity and cavity expansion velocity, have been incorporated. Cavity dynamics, primarily influenced by \hat{h} (the spark bubble's dimensionless depth), were identified as the governing factor in the second branching process of jet modes.
3. Building on insights from the two branching processes of jet modes, more detailed theoretical analyses of the boundaries between different jet modes were conducted.
4. Figure 5b of the previous manuscript was integrated into Fig. 4 to better illustrate the two branching processes in the evolution of the jet modes.
5. Jet mode III has been renamed from 'Ventilation' to 'Cavity Venting' to better represent the process in which airflow from the atmosphere flows through the cavity-induced channel into the underlying spark bubble as advised by a reviewer.
6. A dimensional analysis to identify the governing dimensionless parameters and a theoretical analysis of the sphere dynamics to derive its characteristic velocity are added in the Methods section.

Responses to Reviewers' Comments for Manuscript NCOMMS-24-09227A

Particulate Reshapes Surface Jet Dynamics Induced by a Cavitation Bubble

Addressed Comments for Manuscript to

by

Xianggang Cheng, Xiao-Peng Chen, Zhi-Ming Yuan and Laibing Jia

Authors' Response to Reviewer 3

General Comments: The authors have addressed some of my concerns carefully, but the revised manuscript still lacks enough physical insights and some of the physical arguments seems unreasonable.

Thank you for your continued review and constructive feedback. In response to your concerns, we have incorporated a theoretical analysis of the sphere dynamics and derived its characteristic velocity. We provided detailed theoretical analyses regarding the dimensionless parameters for the phase plot. Additionally, we included further quantitative measurements on the velocities of the water surface and the cavity across various \hat{h} values, which support our theoretical predictions on the transitions between distinct jet modes. These additions enhance our understanding of the physical mechanisms underlying the different jet modes.

In the revised manuscript, we have made the following major changes:

1. The dynamic wetting process of the sphere during immersion has been analysed, considering sphere surface wettabilities and wetting failure phenomena. Additionally, a thorough theoretical analysis of the sphere's dimensionless immersion time t_c has been included.
2. Quantitative measurements and theoretical predictions, including water surface velocity and cavity expansion velocity, have been incorporated. Cavity dynamics, primarily influenced by \hat{h} (the spark bubble's dimensionless depth), were identified as the governing factor in the second branching process of jet modes.
3. Building on insights from the two branching processes of jet modes, more detailed theoretical analyses of the boundaries between different jet modes were conducted.
4. Figure 5b of the previous manuscript was integrated into Fig. 4 to better illustrate the two branching processes in the evolution of the jet modes.
5. Jet mode III has been renamed from 'Ventilation' to 'Cavity Venting' to better represent the process in which airflow from the atmosphere flows through the cavity-induced channel into the underlying spark bubble as advised by a reviewer.
6. A dimensional analysis to identify the governing dimensionless parameters and a theoretical analysis of the sphere dynamics to derive its characteristic velocity are added in the Methods section.

Following are our detailed responses to your comments.

Comment 1: Regarding their response to my previous comments, the dimensionless chasing number t_c is not a parameter that can be easily obtained with initial parameters

of the system - they need to measure the particle immersed depth h_{im} , which is actually a function of the parameters of particle and liquid. Besides, I cannot understand the physical reason for the relation of $V_s = \rho^{-1/2}V_l$. The motion of the particle is set by the acting forces including surface tension, drag force, and added mass force, etc., which cannot be described by $V_s = \rho^{-1/2}V_l$. What is the physical picture of particle immersion? It may be resulted by the movement of contact line (when the contact line moves to the top of the particle) or the pinch off of the liquid surface above the particle, while the authors only consider the difference between the velocities of liquid surface and particle. This should be clearly demonstrated by solid physical arguments.

In addition, many arguments for the derivation of t_c can be validated by experiments, for example the velocities of particle and liquid surface are measurable, as well as the characteristic times for maximum radius and particle immersion. These measurements will well guide and support the interpretations of the jetting behaviour.

Thank you for your constructive comments. In response to your concern on the dimensionless number t_c , here we provide detailed analyses on the dynamic wetting process of the sphere during its immersion. The parameter t_c represents the ratio of the sphere's immersion time to the spark bubble's expansion time. It is utilised to characterise the first branching process of jet modes, indicating whether a singular jet forms (i.e., whether the sphere is fully submerged). Following is our detailed response to your comments.

The sphere immersed depth h_{im} and the arc length s of the unimmersed portion

Building upon dynamic wetting principles, we have replaced the sphere's unimmersed depth, $(2r - h_{im})$ with the arc length s of the sphere's unimmersed portion, as the length scale for the sphere's immersion process (see Fig. 1). The arc length $s = \phi r$, where r is the radius of the sphere and ϕ denotes the azimuthal angle of the contact line. The angle ϕ is governed by the equilibrium among gravity, buoyancy, and surface tension acting on the sphere. The balance of these forces is described by the following equation[1, 2, 3]:

$$6 \sin \phi \sin (\phi - \theta) - (\cos^3 \phi - 3 \cos \phi - 2)(\rho - 1)Bo = 0, \quad (1)$$

where θ is the contact angle of the sphere with water, ρ is the density ratio of the solid sphere to the water, and Bo is a Bond number defined by $Bo = \rho_l r^2 g / \sigma$. Here, ρ_l is the density of water, g is the gravitational acceleration, and σ is the surface tension of water. The angle ϕ is implicitly a function of Bo , ρ , and θ , expressed as $\phi = \phi(Bo, \rho, \theta)$.

Figure 1: A sketch for the floating sphere. The red curve denotes the arc length s of the sphere's unimmersed portion.

The sphere velocity V_s

Through dimensional analysis, we simplify the system in this study into seven dimensionless parameters, which are the top seven dimensionless numbers listed in Tab. 1. They are the Weber number We , representing the ratio of liquid inertia to capillary force, the Froude number Fr , which compares liquid inertial to gravitational force, the Reynolds number Re , indicating the ratio of liquid inertia to viscous force, the density ratio ρ , the contact angle θ , the radius ratio between the sphere and the spark bubble, \hat{r} , and a dimensionless depth, h^* . The table also includes the Bond number $Bo = We/Fr^2$, which compares gravitational to capillary force, as well as the spark bubble's dimensionless depth, $\hat{h} = h/R_m = h^*\hat{r}$, a parameter commonly used in studies on cavitation bubble interactions with flat water surfaces[4, 5]. In this study, we use \hat{h} instead of h^* .

The ranges of these dimensionless parameters are summarised in Tab. 1. The Reynolds number ranges from 10^2 to 10^4 , indicating that the inertial force significantly outweighs the viscous force, and the viscous force can be neglected. The Froude number $Fr > 30$ suggests that the inertial force dominates over the gravitational force. The Weber number ranges from 10^0 and 10^3 . At lower Weber numbers, surface tension effects become pronounced and should be considered, whereas at higher values, inertial forces dominate and surface tension can be neglected. The Bond number $Bo < 1$, suggesting that gravitational and buoyancy forces are negligible compared to capillary force[6, 7, 8, 9].

The sphere's motion equation in the vertical direction can be expressed as,

$$(m_s + m_a)a_s = f_d + f_c + f_v + f_g + f_b, \quad (2)$$

where m_s is the sphere's mass, m_a is the added mass, and a_s is the sphere's acceleration. The terms f_d , f_c , f_v , f_g , and f_b represent the form drag, capillary force, viscous force, gravitational force, and buoyancy force, respectively.

Based on the dimensional analysis above, we find that the liquid inertia and capillary force are dominant forces for the sphere. Therefore, by neglecting f_v , f_g , and f_b , and

Dimensionless numbers	Symbol	Definition	Range
Weber number	We	$\frac{\rho_l V_l^2 r}{\sigma}$	2-6126
Froude number	Fr	$\frac{V_l}{\sqrt{gr}}$	30-412879
Reynolds Number	Re	$\frac{\rho_l V_l r}{\mu}$	112-14938
Density ratio	ρ	$\frac{\rho_s}{\rho_l}$	1.4-8.5
Contact angle	θ	θ	$81^\circ - 154^\circ$
Radius ratio	\hat{r}	$\frac{r}{R_m}$	0.01-0.27
Dimensionless depth	h^*	$\frac{h}{r}$	6-268
Bond number	Bo	$\frac{\rho_l r^2 g}{\sigma} = \frac{We}{Fr^2}$	0.001-0.8
Spark bubble's dimensionless depth	\hat{h}	$\frac{h}{R_m} = h^* \hat{r}$	0.6-3.5

Table 1: Relevant dimensionless parameters and their characteristic values.

defining a hydrodynamic force $f_h = f_d - m_a a_s$ following previous studies[9, 10, 11], we obtain

$$m_s a_s = f_h + f_c. \quad (3)$$

The hydrodynamic force f_h can be represented as[9, 10, 11]

$$f_h = \frac{\pi}{2} C_h \rho_l V_l^2 r^2, \quad (4)$$

where C_h is the hydrodynamic force coefficient. For liquid flow around the sphere with $Re > 10^2$, it can be assumed as potential flow, with C_h being of order unity[12, 13, 10, 11]. The capillary force is influenced by the wettability of the sphere[8], with a characteristic value of

$$f_c = \pi r (1 - \cos \theta) \sigma. \quad (5)$$

By substituting f_h and f_c into Eq. 3, we can give the scale of sphere acceleration:

$$a_s \sim (C_h + 4 \sin^2 \frac{\theta}{2} We^{-1}) \rho^{-1} \frac{V_l^2}{r}. \quad (6)$$

Further, with the time scale for sphere acceleration, $\Delta t \sim r/V_l$, the characteristic velocity of the sphere is obtained:

$$V_s \sim a_s \Delta t \sim (C_h + 4 \sin^2 \frac{\theta}{2} We^{-1}) \rho^{-1} V_l, \quad (7)$$

where the first term in the brackets corresponds to the influence of liquid inertia, and the second term represents the effect of surface tension. Additionally, when $We \gg 1$, surface tension effects can be neglected, resulting in $V_s \sim C_h \rho^{-1} V_l$.

The sphere immersion process

In the revised manuscript, we have included detailed discussions on the dynamic wetting process of the sphere during its immersion, as illustrated in Fig. 2. These include the case where the wetted liquid film follows the sphere, and the other case where wetting failure occurs[14, 15, 16] and the film detaches from the sphere. The expression for the dimensionless chasing time (dimensionless immersion time of the sphere), t_c , is applicable in both cases. The detailed descriptions of the sphere immersion process are presented as follows.

As the spark bubble expands, the water surface rises with a velocity V_l . The water impacts and wets the sphere, and the sphere accelerates to a velocity V_s . This scenario is analogous to water entry problems, where a characteristic feature of liquid-solid impacts is the formation of a liquid film during the interaction[14, 15, 16, 17]. If the relative speed between the water surface and the sphere, $v = V_l - V_s$, is below a critical velocity, the liquid film adheres the sphere and climbs to its north pole. If v exceeds this critical velocity, the film becomes unstable and detaches from the sphere, resulting in wetting failure[14, 15, 16].

The transition between wetting and non-wetting regimes occurs when the relative speed of liquid-solid surpasses the maximum contact line speed allowed[14]. For hydrophilic spheres ($\theta < 90^\circ$), the critical velocity $U^* \approx 0.1\sigma/\mu$, which is basically independent of the contact angle. For hydrophobic spheres ($\theta > 90^\circ$), U^* is lower and dependent on the contact angle, $U^* \approx (7/270)\sigma/\mu(\pi - \theta)^3$. In this study, three types of spheres with different wetting abilities were used. Based on the wetting transition model by Duez et al.[14], the values of U^* for the hydrophilic ($\theta = 81^\circ$), hydrophobic ($\theta = 108^\circ$), and super-hydrophobic ($\theta = 154^\circ$) spheres used in our experiments are 7.3, 3.8, and 0.2 m/s, respectively.

We measured the average water surface velocity V_l from $t = 0$ to τ , across different values of \hat{h} . As shown in Fig. 3, the normalised water surface velocity, V_l/V_b , follows a power-law relationship with \hat{h} . Assuming the spark bubble expands spherically, V_l can be estimated using volume conservation, leading to $V_l h^2 \sim V_b R_m^2$. This yields $V_l = a V_b \hat{h}^{-2}$. The theoretical predictions align well with the experimental data, with a best-fitting prefactor of $a = 0.64$. By neglecting the sphere's speed at the initial phase ($V_s \approx 0$), the

Figure 2: Immersion processes for spheres with either hydrophilic (**a**, blue circles) or hydrophobic (**b**, orange circles) surfaces during the spark bubble's expansion phase ($t \leq \tau$). These represent the first branching process of jet modes, determined by whether a singular jet forms before $t = \tau$. Solid lines indicate the air-water interfaces.

critical values of \hat{h} for wetting transition can be determined by setting V_l equal to U^* . The predicted critical values of \hat{h} are 1.0, 1.4, and 6.3 for hydrophilic, hydrophobic, and super-hydrophobic spheres, respectively. In our experiments, \hat{h} ranges from 0.6 to 3.5. It means the wetting transition will occur for both hydrophilic and hydrophobic spheres, while super-hydrophobic spheres will consistently undergo wetting failure. In Tab. 2, we summarised the critical values and potential immersion regimes for three types of spheres with different wetting abilities.

For hydrophilic spheres, three wetting regimes can occur (Fig. 2a). In the first case, the relative speed v exceeds the critical speed U^* , and a cavity forms above the sphere. The rim of the liquid film eventually gathers above the sphere and ejects a singular jet, leaving air bubbles between the jet root and the sphere[18] (Fig. 2-a1). In the second case, the relative speed v is below U^* , the liquid film follows the sphere and climbs up[14]. Before the spark bubble collapses, the film converges at the north pole of the sphere and ejects a singular jet (Fig. 2-a2). The third case occurs when the relative velocity v is low

Figure 3: The dimensionless water surface velocity on flat water surfaces, V_l/V_b , follows a power-law relationship with \hat{h} , $V_l/V_b \sim \hat{h}^{-2}$.

enough that the wetting film cannot fully immerse the sphere at $t = \tau$, and no singular jet is ejected, as illustrated in Fig. 2-a3.

For hydrophobic spheres, three distinct wetting regimes will appear. When the relative speed v exceeds U^* , wetting failure occurs, and the contact line tends to pin near the sphere's equator[14, 19, 18, 20]. This results in a larger cavity above the sphere (Fig. 2-b1,b2), in contrast to the narrow air channel formed in the hydrophilic case[18, 17] (Fig. 2-a1). If v is sufficiently high, the rim of the detached film converges above the sphere and ejects a singular jet, leaving behind an air bubble comparable in size to the sphere[14, 17] (Fig. 2-b1). When v is not fast enough (still larger than U^*), an air cavity forms above the sphere with its aperture remaining open (Fig. 2-b2). Another case is that when v is smaller than U^* , the film wets the sphere more slowly, and the sphere remains unimmersed at $t = \tau$ (Fig. 2-b3).

In the experiment, we observed singular jets occurring in Modes I, II, and III.a before $t = \tau$, indicating that the sphere was fully submerged by this time, and the corresponding immersion regimes should be a1, a2, or b1 in Fig. 2. In contrast, no singular jets were present in Modes III.b, IV, and V until $t = \tau$, suggesting that the sphere was not yet fully submerged at $t = \tau$, and the corresponding immersion regimes should be a3, b2, or b3 in Fig. 2.

To characterise this branching process, we define an ‘immersion time’, t_c^u , which represents the duration required for the sphere to become fully submerged. To theoretically determine t_c^u , two cases are considered: one where wetting failure does not occur and one where it does.

θ ($^\circ$)	81	108	154
U^* (m/s)	7.3	3.8	0.2
\hat{h}^*	1.0	1.4	6.3
Potential immersion regimes	a1, a2, a3	b1, b2, b3	b1, b2

Table 2: Critical values for three types of spheres with different wetting abilities. θ is the sphere's contact angle with water, U^* is the critical velocity for wetting transition, and \hat{h}^* is the corresponding critical dimensionless spark bubble depth. In our experiments, \hat{h} ranges in 0.6 – 3.5. The immersion regimes are illustrated in Fig. 2.

During the sphere's immersion, the liquid film and contact line advance at a velocity $V_f \approx \zeta v$, where ζ is a prefactor on the order of unity [21, 14]. In the case without wetting failure, the wetted film and contact line move at a speed comparable to v . Using the arc length s as the length scale, the immersion time can be estimated as $t_c^u \approx s/v$. In the case where wetting failure occurs, the detached liquid film advances more quickly due to reduced viscous dissipation at the sphere's surface (with $\zeta = 2$ in Duez et al. [14]). The distance travelled by the detached film before converging is also increased. Here we introduce a factor ζ' to estimate the travel distance as $\zeta' s$, where ζ' is greater than one. The immersion time in this case is given by $t_c^u \approx (\zeta' s)/(\zeta v)$. Both the travel distance and the travel speed are increased. However, these increases tend to cancel each other out, making the ratio ζ'/ζ close to unity. Therefore, the expression $t_c^u \approx s/v$ holds for both cases, as supported by subsequent experimental results (Fig. 5).

The critical condition for fully immersion is that the sphere completely submerged just before the spark bubble collapses, i.e., $t_c^u = \tau$. Normalising t_c^u by τ , one obtains the dimensionless immersion duration:

$$t_c = \frac{t_c^u}{\tau}. \quad (8)$$

By substituting V_l ($\sim V_b/\hat{h}^2$) and V_s (Eq. 7) into Eq. 8, the dimensionless immersion duration of the sphere is derived,

$$t_c \sim \frac{\phi \hat{r}}{\alpha} \hat{h}^2 \sim \frac{\phi(Bo, \rho, \theta) \hat{r}}{1 - (C_h + 4 \sin^2 \frac{\theta}{2} We^{-1}) \rho^{-1}} \hat{h}^2, \quad (9)$$

where the factor α is expressed as $\alpha \sim 1 - (C_h + 4 \sin^2 \frac{\theta}{2} We^{-1}) \rho^{-1}$. The parameter t_c is determined by six dimensionless numbers: Bo , We , ρ , θ , \hat{r} and \hat{h} . Theoretically, if $t_c \leq 1$, the sphere is fully immersed before the spark bubble collapses, resulting in Modes

I, II or III.a. Conversely, if $t_c > 1$, the sphere remains unimmersed at $t = \tau$, leading to Modes III.b, IV or V.

Quantitative measurements

In response to your concerns, we have included further quantitative measurements, including the average water surface velocity V_l (Fig. 3), and the average cavity expansion velocity V_c (Fig. 4), across different values of \hat{h} . The experimental data on V_l validates the theoretical expression of $V_l \sim V_b/\hat{h}^2$. The experimental results of V_c indicate that the cavity dynamics are primarily determined by \hat{h} . Additionally, these quantitative measurements support our theoretical predictions for jet mode transitions, which will be discussed in our responses to ‘Comment 3’.

Due to experimental constraints, including the intense light produced by the spark bubble and the obscuring effects of curved interfaces, we are currently unable to track the movement of the sphere before its immersion and determine its exact immersion duration. However, we provided a theoretical analysis on the sphere dynamics during its immersion process. The derived dimensionless chasing time (immersion time) t_c based on the theoretically determined sphere velocity, effectively distinguishes between jet modes with or without singular jets before $t = \tau$. We appreciate your constructive suggestions on the quantitative measurements of sphere kinematics. Future work will involve numerical simulations to investigate the sphere dynamics in detail that are not accessible experimentally.

Comment 2: My second concern is about the interpretation of another dimensionless number \hat{h} . The radiated energy density $e \sim E/(4\pi h^2) \sim (p_{atm} - p_v)R_m\hat{h}^{-2}$, which means that \hat{h} cannot solely represent the radiated energy density (actually e is a dimensional parameter while \hat{h} is dimensionless). Using radiated energy to explain why \hat{h} divides different modes seems not reasonable. Thus, the physical meaning of the proposed dimensionless numbers need to be re-examined carefully.

Thank you for your comments. The energy density on the water surface radiated by the spark bubble, e , is determined by the driving pressure, $(p_{atm} - p_v)$, the maximum radius of the spark bubble, R_m , and the dimensionless depth of the spark bubble, \hat{h} . In this study, the driving pressure $(p_{atm} - p_v)$ remains constant, and R_m is fixed at 9.2 ± 0.5 mm. As a result, the energy density e depends solely on \hat{h} and is inversely proportional to \hat{h}^2 .

In this study, we selected \hat{h} to characterise the second branching process of jet modes, which occurs during the spark bubble’s collapse phase. During this phase, both the

Figure 4: The dimensionless cavity expansion velocity, V_c/V_b , versus the dimensionless spark bubble depth, \hat{h} . The solid line represents theoretical predictions.

concavity at the junction between the singular jet and the water bulge in Modes I, II, and III.a, as well as the pit over the sphere in Modes III.b, IV, and V, expand into downward cavities. To further analyse this process, we measured the cavity expansion velocity V_c , as illustrated in Fig. 4. The results show that V_c is primarily determined by \hat{h} , suggesting that \hat{h} is an appropriate parameter for characterising the cavity dynamics, further determining mode transitions during the second branching process. Detailed theoretical analyses of the second branching process will be discussed in our responses to ‘Comment 3’.

Additionally, \hat{h} is commonly used in studies on cavitation bubble interactions with flat water surfaces, where distinct surface jet modes arise at different values of \hat{h} [4, 5]. Although the size of the surface jet increases with a larger cavitation bubble radius (R_m), similar jet modes are observed at the same \hat{h} [4, 5].

Comment 3: My third concern is about the proposed criteria for different jet modes. Besides the physical meaning of two dimensionless numbers, the critical values of the dimensionless numbers for different modes lack physical reasons. Why $t_c = 0.2$ separates modes II and IV? Why $\hat{h} = 2.1$ marks the boundary separating mode V from mode IV? The same for the critical \hat{h} for the boundaries between modes I, II, and V.

Thank you for your feedback. In response to your concerns, we conducted a dimensional analysis to identify the appropriate dimensionless parameters for building the phase diagram, as discussed in our responses to ‘Comment 1’.

The dimensional analysis identified seven dimensionless parameters that characterise

the system in this study (Tab. 1): the Weber number We , the Froude number Fr , the Reynolds number Re , the density ratio ρ , the contact angle θ , the radius ratio \hat{r} , and a dimensionless depth h^* . Two additional dimensionless parameter, the Bond number $Bo = We/Fr^2$, and the spark bubble's dimensionless depth $\hat{h} = h^*\hat{r}$ are also included. These parameters capture the interactions between inertial, viscous, surface tension, and gravitational forces, along with the geometric and interfacial properties that affect jet formation. We found that using any combination of two among these parameters was insufficient to distinctly classify the various jet modes observed in this study.

In this study, we selected t_c and \hat{h} for the phase digram as they have clear physical meanings. These two dimensionless parameters characterise the two branching processes of jet modes. From experiments, we observed two primary branching processes from a sphere resting on the water surface to five distinct jet modes. The first branching process occurs during the expansion phase of the spark bubble ($t \leq \tau$), depending on whether a singular jet forms. The second branching process occurs during the collapse phase of the spark bubble ($\tau \leq t \leq T_1$), the cavity expands and further evolves into five distinct jet modes.

To characterise the first branching process, we introduced the dimensionless chasing time t_c . It represents the ratio of the sphere's immersion time to the spark bubble's expansion time. Theoretically, if $t_c \leq 1$, the sphere becomes fully submerged before the spark bubble collapses, resulting in Modes I, II or III.a with singular jets. Conversely, if $t_c > 1$, the sphere remains unimmersed at $t = \tau$, leading to Modes III.b, IV or V without singular jets at $t = \tau$. The second branching process is primarily governed by the cavity dynamics. As shown in Fig. 4, the cavity expansion velocity V_c is primarily determined by \hat{h} . Therefore, \hat{h} is a suitable parameter for characterising the second branching process.

Since t_c is influenced by \hat{h} (Eq. 9), to decouple these two parameters, we constructed the phase diagram based on \hat{h} and $\phi\hat{r}/\alpha$ (where $\phi\hat{r}/\alpha = t_c/\hat{h}^2$, with $C_h = 1$ in all cases to calculate α). As shown in Fig. 5, the five jet modes cluster distinctly. In the phase plot, the oblique line represents the boundary of modes with and without singular jets at $t = \tau$. It corresponds to the first branching process, applicable to experiments using spheres with various wettabilities. Along this line, $t_c = 0.24$. It is lower than the theoretical value of $t_c = 1$. This discrepancy may be attributed to an overestimation of the liquid film velocity V_f by using $(V_l - V_s)$ when calculating t_c , due to the absence of a coefficient for accurate quantification of V_f . Despite this, the phase diagram effectively delineates the first branching process without requiring an additional fitting factor. This demonstrates that our theoretical model using t_c successfully distinguishes between the jet modes.

Figure 5: The plot of $\phi \hat{r} / \alpha (= t_c / \hat{h}^2)$ versus \hat{h} categorises the five jet modes. \hat{h} is the spark bubble's dimensionless depth. t_c is the dimensionless immersion time, defined as the ratio of the sphere's immersion time to the spark bubble's expansion time. The formula $t_c = \phi \hat{r} \hat{h}^2 / \alpha$ incorporates the azimuthal angle ϕ of the contact line and the radius ratio \hat{r} between the sphere and the spark bubble. The factor α is determined by the density ratio ρ , Weber number We , and contact angle θ , expressed as $\alpha = 1 - (1 + 4 \sin^2 \frac{\theta}{2} We^{-1}) \rho^{-1}$. The oblique line marks the boundary at $t_c = 0.24$, distinguishing jet modes with singular jets (Modes I, II, and III.a below the line) from those without singular jets (Modes III.b, IV, and V above the line). Additional lines are included to indicate the boundaries between different jet modes.

The cavity expansion velocity V_c

As discussed above, the cavity dynamics are crucial in the second branching process of jet modes. To estimate V_c , we analyse the physical process during the spark bubble's collapse. The spark bubble is centred at a depth h . At time $t = \tau$, the bubble reaches its maximum radius R_m . We consider a spherical shell of water extending from R_m to h at the water surface. As the spark bubble collapses over the duration τ_{coll} (approximated as τ [22, 4, 5]), the cavity begins to expand downward at a velocity V_c from the water surface to a depth of $V_c\tau_{coll}$. This collapsing causes the spherical shell to contract, reducing its inner radius from R_m to zero, and outer radius from h to $h - V_c\tau_{coll}$. With the conservation of water volume within this spherical shell, we establish the relationship: $h^3 - R_m^3 \sim (h - V_c\tau_{coll})^3$, which yields

$$V_c \approx b \left(\hat{h} - \sqrt[3]{\hat{h}^3 - 1} \right) V_b. \quad (10)$$

As shown in Fig. 4, the theoretical predictions align well with the experimental data, with a best fitting prefactor of $b = 4.35$.

The quantitative measurements on V_c guide the theoretical predictions on the second branching process of jet modes. In Fig. 4, two critical values, $\hat{h} \approx 1.3$ and 2.1, differentiate Modes III.b and IV, and Modes IV and V, respectively. These suggest that the transitions among these modes are primarily governed by V_c . When V_c is sufficiently high, the cavity reaches the spark bubble before $t = T_1$, a channel forms, resulting in Mode III.b. As V_c decreases at larger values of \hat{h} , the cavity seals at its neck, leading to Mode IV. Further decreases in V_c prevent cavity sealing, resulting in Mode V.

The boundary between Modes III.b and IV

In Fig. 5, the regime map shows a vertical line at $\hat{h} \approx 1.3$, which separates Modes III.b and IV. Mode III.b is observed for smaller values of \hat{h} , while larger values correspond to Mode IV. This is consistent with the experimental finding that Mode III.b exhibits a larger cavity expansion velocity than Mode IV (see Fig. 4). The critical condition separating these two modes occurs when the cavity bottom reaches the spark bubble at $t = T_1$. Disregarding the uplift of the water surface at $t = \tau$ and the migration of the spark bubble's centroid at $t = T_1$, we find $V_c\tau_{coll} \approx h$. This yields a critical value of $\hat{h} \approx 1.23$, which closely matches the experimental result of $\hat{h} \approx 1.3$.

The boundary between Modes IV and V

The distinction between Modes IV and V depends on whether the cavity rim seals at $t = T_1$. This sealing process resembles the surface sealing phenomenon observed in water entry problems[19, 16, 23]. As the spark bubble collapses, the cavity expands, allowing air to flow in and fill the void. Based on the geometric relationship, the air flow speed through the cavity rim is estimated as $V_a \sim (r_c/r_{rim})^2 V_c$, where r_c and r_{rim} denote the radii of the cavity and its rim, respectively.

According to Bernoulli's principle, this inflowing air induces an aerodynamic pressure difference across the rim, $\Delta p \propto -\rho_a V_a^2$, where ρ_a is the air density. This pressure difference drives the collapse of the cavity rim, and a modified Rayleigh-Plesset equation can be obtained[19, 23]: $\rho_l \left(r_{rim} \ddot{r}_{rim} + \frac{3}{2} \dot{r}_{rim}^2 \right) = \Delta p$. Further, the collapse duration of the rim is derived, $\tau_{rim} \sim r_{rim}^0 \sqrt{\rho_l / |\Delta p|}$, where r_{rim}^0 is the initial radius of the cavity rim. If $\tau_{rim} \leq \tau_{coll}$, the cavity seals before the spark bubble rebounds, resulting in Mode IV. Otherwise, the cavity remains open at $t = T_1$, leading to Mode V. The value of τ_{rim} is primarily governed by Δp , which depends on \hat{h} . This suggests a critical \hat{h} that determines cavity sealing, consistent with experimental observations showing that the vertical line at $\hat{h} \approx 2.1$ marks the boundary between Modes IV and V.

The boundary between Modes I and II

In the case where the singular jet forms before $t = \tau$, the concavity at the junction between the singular jet and the water bulge moves downward and develops into a cavity as the spark bubble collapses. The critical condition distinguishing Modes I and II arises when the concavity reaches the sphere at $t = T_1$. At $t = \tau$, two distances affect the transition between these two modes: the distance between the concavity and the sphere, $d_1 \approx (V_l - V_s)\tau$, and the distance between the concavity and the spark bubble, $d_2 \approx V_l\tau + h$. Here, d_1 represents the initial gap the concavity needs to close to reach the sphere, and d_2 determines the downward speed of the concavity, V_c . Ignoring the sphere's motion during the spark bubble's collapse phase, we find $V_c\tau_{coll} \approx d_1$, yielding a critical value of $\hat{h} \approx 1.04$ for $C_h = 1$, $We \gg 1$, and $\rho \rightarrow \infty$. This theoretical value is close to the experimental result of $\hat{h} \approx 1.15$ that separates Modes I and II.

The boundary between Modes II and V

In the phase diagram, the oblique boundary at $t_c = 0.24$ separates jet modes with and without singular jets before $t = \tau$. However, several experiments, classified as Mode V

without singular jets, fall below this boundary (Fig. 5). These exceptions occur with sphere radii r ranging in $100 - 175 \mu\text{m}$ and \hat{h} between 1.9 and 2.8.

We hypothesize that the spheres in these experiments are fully submerged at $t = \tau$. The singular jet radius r_{jet} is roughly an order of magnitude smaller than the sphere's radius, with $r_{jet} = O(10) \mu\text{m}$ for spheres with radii between $100 - 175 \mu\text{m}$. By balancing the jet's kinetic energy per unit length ($\sim \pi \rho_l r_{jet}^2 V_r^2 / 2$) with its surface energy per unit length ($\sim 2\pi \sigma r_{jet}$), the reduction in jet velocity due to surface tension is $V_r \sim [4\sigma / (\rho_l r_{jet})]^{1/2}$ [24, 25]. For $r_{jet} = 10 \mu\text{m}$, $V_r \sim 5.4 \text{ m/s}$. In the experiments, the singular jet velocity at $\hat{h} \approx 1.7$ is $V_{jet} = O(1) \text{ m/s}$. V_r is of the same order as V_{jet} , suggesting that surface tension significantly suppresses the jet's development. Additionally, the Weber number based on the jet parameters, $We_{jet} = \rho_l V_{jet}^2 r_{jet} / \sigma$ is of $O(10^{-1})$. It suggests that for small r and large \hat{h} , surface tension plays a key role in singular jet dynamics, resulting in the absence of the singular jet.

Comment 4: Fourthly, as the author stated, they focus on the initial branching of modes in Fig. 4 during the bubble's initial expanding stage when proposing t_c , then t_c should describe the boundary between modes I, II and modes IV, V. However, the regime map in Fig. 5a does not support this.

Actually, the boundaries between different modes in Fig. 5a are blur and we can use different lines to describe them. For example, the boundaries between III and IV, as well as between IV and V, can also be roughly described by two oblique lines perpendicular to the current oblique line. The boundaries between I and II, as well as between II and V, show a "V" shape.

Thank you for your comments. As shown in Fig. 5, the oblique line represents the boundary of modes with and without singular jets at $t = \tau$. It corresponds to the first branching process, applicable to experiments using spheres with various wettabilities. Along this line, $t_c = 0.24$. It is lower than the theoretical value of $t_c = 1$. This discrepancy may be attributed to an overestimation of the liquid film velocity V_f by using $(V_l - V_s)$ when calculating t_c , due to the absence of a coefficient for accurate quantification of V_f . Despite this, the phase diagram effectively delineates the first branching process without requiring an additional fitting factor. This demonstrates that our theoretical model using t_c successfully distinguishes between the jet modes.

We conducted additional measurements of the cavity expansion velocity and provided theoretical predictions (Fig. 4). Our findings indicate that the second branching process of jet modes is primarily governed by the cavity dynamics, which are mainly influenced by \hat{h} . We proposed critical conditions separating jet modes during the second branching

process. The theoretical predictions for mode transitions have been discussed in our responses to ‘Comment 3’.

Comment 5: Additionally, I also cannot understand the interpretation in the paragraph before section “Conclusion and Discussion”. Why “the omission of surface tension effects for small-radii spheres underestimates their velocities”? The small radii spheres correspond to a region below an oblique line instead of the horizontal line $t_c = 0.2$.

From the supplementary videos, one can find that the local dynamics around the particle matters for the jet dynamics. Thus I suggest the authors to look into the detail on the dynamics between the particle and liquid surface, either experimentally or theoretically, which may be a feasible route for convincing physical explanations.

Thank you for your comments. To address your concerns, we have provided a theoretical analysis of the sphere dynamics, as discussed in our responses to ‘Comment 1’. We find that the liquid inertia and capillary force are dominant forces acting on the sphere during its immersion process. Through the force analysis, we obtain the characteristic sphere velocity, $V_s \sim (C_h + 4 \sin^2 \frac{\theta}{2} We^{-1}) \rho^{-1} V_l$. The first term in the brackets corresponds to the influence of liquid inertia, and the second term represents the surface tension effects. When $We \gg 1$, the surface tension effect becomes negligible, which yields $V_s \sim C_h \rho^{-1} V_l$.

Noting that $We = \rho_l V_l^2 r / \sigma \sim \rho_l V_b^2 r / (\sigma \hat{h}^4)$, the Weber number depends on the sphere radius r and the dimensionless depth \hat{h} . Experiments with smaller-radius spheres and larger values of \hat{h} result in smaller Weber numbers, which strengthen the effects of surface tension. In our previous manuscript, we estimated the sphere velocity considering only the liquid inertia effects, assuming that $We \gg 1$. This assumption, with neglecting surface tension effects, underestimates the sphere velocity.

We appreciate your constructive suggestions on examining the detailed dynamics of the sphere and its surrounding air-liquid interfaces. In response to your concerns, we have incorporated a theoretical analysis of the sphere dynamics, and derived its characteristic velocity. The dynamic wetting process of the sphere during immersion has been analysed, considering sphere surface wettabilities and wetting failure phenomena. A thorough theoretical analysis of the dimensionless chasing time t_c (the sphere’s dimensionless immersion time) has been included. Additionally, we included further quantitative measurements on the velocities of the water surface and the cavity across various \hat{h} values, which support our theoretical predictions on the transitions between distinct jet modes. These additions deepen our understanding of the physical mechanisms underlying the different jet modes.

References

- [1] D. Vella, D. G. Lee, and H. Y. Kim. The load supported by small floating objects. *Langmuir*, 22(14):5979–5981, 2006.
- [2] C. W. Extrand and S. I. Moon. Using the flotation of a single sphere to measure and model capillary forces. *Langmuir*, 25(11):6239–44, 2009.
- [3] D. Vella. Floating versus sinking. *Annual Review of Fluid Mechanics*, 47(1):115–135, 2015.
- [4] S. Zhang, S. P. Wang, and A. M. Zhang. Experimental study on the interaction between bubble and free surface using a high-voltage spark generator. *Physics of Fluids*, 28(3):032109, 2016.
- [5] Y. J. Kang and Y. Cho. Gravity-capillary jet-like surface waves generated by an underwater bubble. *Journal of Fluid Mechanics*, 866:841–864, 2019.
- [6] D. G. Lee and H. Y. Kim. Impact of a superhydrophobic sphere onto water. *Langmuir*, 24(1):142–145, 2008.
- [7] B. Ji, Q. Song, and Q. Yao. Numerical study of hydrophobic micron particle’s impaction on liquid surface. *Physics of Fluids*, 29(7), 2017.
- [8] H. Chen, H. Liu, X. Lu, and H. Ding. Entrapping an impacting particle at a liquid–gas interface. *Journal of Fluid Mechanics*, 841:1073–1084, 2018.
- [9] B. Ji, Q. Song, A. Wang, and Q. Yao. Critical sinking of hydrophobic micron particles. *Chemical Engineering Science*, 207:17–29, 2019.
- [10] B. Ji, Q. Song, K. Shi, J. Liu, and Q. Yao. Oblique impact of microspheres on the surface of quiescent liquid. *Journal of Fluid Mechanics*, 900, 2020.
- [11] B. Ji, Z. Tang, and Q. Song. Oblique impact dynamics of micron particles onto a liquid surface. *Physical Review Fluids*, 5(11), 2020.
- [12] M. Shiffman and Donald C. Spencer. The force of impact on a sphere striking a water surface. AMP Technical Reports 42. 1R, 42. 2R, AMG-NYU Nos. 105, 133, 1945.
- [13] M. Moghisi and P. T. Squire. An experimental investigation of the initial force of impact on a sphere striking a liquid surface. *Journal of Fluid Mechanics*, 108:133–146, 1981.
- [14] Cyril Duez, Christophe Ybert, Christophe Clanet, and Lydéric Bocquet. Making a splash with water repellency. *Nature Physics*, 3(3):180–183, 2007.
- [15] M. H. Zhao, X. P. Chen, and Q. Wang. Wetting failure of hydrophilic surfaces promoted by surface roughness. *Scientific Reports*, 4:5376, 2014.

- [16] Tadd T. Truscott, Brenden P. Epps, and Jesse Belden. Water entry of projectiles. *Annual Review of Fluid Mechanics*, 46(1):355–378, 2014.
- [17] Nathan B. Speirs, Mohammad M. Mansoor, Jesse Belden, and Tadd T. Truscott. Water entry of spheres with various contact angles. *Journal of Fluid Mechanics*, 862, 2019.
- [18] Minh Do-Quang and Gustav Amberg. The splash of a solid sphere impacting on a liquid surface: Numerical simulation of the influence of wetting. *Physics of Fluids*, 21(2), 2009.
- [19] J. M. Aristoff and J. W. M. Bush. Water entry of small hydrophobic spheres. *Journal of Fluid Mechanics*, 619:45–78, 2009.
- [20] H. Ding, B. Q. Chen, H. R. Liu, C. Y. Zhang, P. Gao, and X. Y. Lu. On the contact-line pinning in cavity formation during solid–liquid impact. *Journal of Fluid Mechanics*, 783:504–525, 2015.
- [21] James Mark Oliver. *Water entry and related problems*. Thesis, University of Oxford, 2002.
- [22] J. P. Franc and J. M. Michel. *Fundamentals of Cavitation*, pages p.36–39. Springer, Dordrecht, 2005.
- [23] J. Eshraghi, S. Jung, and P. P. Vlachos. To seal or not to seal: The closure dynamics of a splash curtain. *Physical Review Fluids*, 5(10), 2020.
- [24] I. R. Peters, Y. Tagawa, N. Oudalov, C. Sun, A. Prosperetti, D. Lohse, and D. van der Meer. Highly focused supersonic microjets: Numerical simulations. *Journal of Fluid Mechanics*, 719:587–605, 2013.
- [25] J. M. Gordillo and Stephan Gele. Generation and breakup of Worthington jets after cavity collapse. Part 2. Tip breakup of stretched jets. *Journal of Fluid Mechanics*, 663:331–346, 2010.

Responses to Reviewers' Comments for Manuscript NCOMMS-24-09227A

Particulate Reshapes Surface Jet Dynamics Induced by a Cavitation Bubble

Addressed Comments for Manuscript to

by

Xianggang Cheng, Xiao-Peng Chen, Zhi-Ming Yuan and Laibing Jia

Authors' Response to Reviewer 4

General Comments: The presentation of the results has been improved, in particular with the addition of Fig. 3 and 5 and the reorganisation of the text. However, the paper still lacks a clear message linked to quantitative analysis. We believe the measurements are not exploited enough and could be used to obtain a precise characterisation of the regimes and a better understanding of the physical mechanisms.

Thank you for your positive feedback and constructive suggestions. In response to your concerns, we conducted further quantitative measurements, including the water surface velocity V_l , the cavity expansion velocity V_c , and the jet velocity V_j , across various values of the dimensionless spark bubble depths \hat{h} . We also incorporated theoretical predictions on V_l and V_c . We found that the second branching process of jet modes is primarily governed by the cavity dynamics, and the quantitative measurements support our theoretical predictions on jet mode transitions. Additionally, we included detailed theoretical arguments for the dimensionless parameters in this study. These additions deepen our understanding of the physical mechanisms behind the different jet modes.

In the revised manuscript, we have made the following major changes:

1. The dynamic wetting process of the sphere during immersion has been analysed, considering sphere surface wettabilities and wetting failure phenomena. Additionally, a thorough theoretical analysis of the sphere's dimensionless immersion time t_c has been included.
2. Quantitative measurements and theoretical predictions, including water surface velocity and cavity expansion velocity, have been incorporated. Cavity dynamics, primarily influenced by \hat{h} (the spark bubble's dimensionless depth), were identified as the governing factor in the second branching process of jet modes.
3. Building on insights from the two branching processes of jet modes, more detailed theoretical analyses of the boundaries between different jet modes were conducted.
4. Figure 5b of the previous manuscript was integrated into Fig. 4 to better illustrate the two branching processes in the evolution of the jet modes.
5. Jet mode III has been renamed from 'Ventilation' to 'Cavity Venting' to better represent the process in which airflow from the atmosphere flows through the cavity-induced channel into the underlying spark bubble as advised by a reviewer.
6. A dimensional analysis to identify the governing dimensionless parameters and a theoretical analysis of the sphere dynamics to derive its characteristic velocity are added in the Methods section.

Following are our point-by-point responses to your comments.

Comment 1: In the phase diagram Fig. 5, there is a clear dependence on the particle's radius which is not taken into account in neither dimensionless number (it cancels in t_c). This should be discussed with the physical description of the regimes.

Thank you for your comments. We would like to clarify that the dimensionless chasing time, t_c , is dependent on the sphere radius r . The parameter t_c represents the ratio of the sphere's immersion time to the spark bubble's expansion time. $t_c = t_c^u/\tau$, where t_c^u is the time required for the sphere to become fully immersed, and τ is the expansion duration of the spark bubble. t_c is utilised to characterise the first branching process of jet modes, indicating whether the sphere is fully submerged at $t = \tau$.

In the revised manuscript, we considered the dynamic wetting process of the sphere to determine t_c^u (Fig. 1). Similar to water entry problems, an essential characteristic of solid to liquid impacts is that a film develops during the impact and climbs up the sphere[1, 2]. The time for the sphere to be fully submerged corresponds to when the liquid film converges above the sphere, which gives $t_c^u \sim s/V_f$. Here, s is the arc length of the sphere's unimmersed portion, and V_f is the liquid film velocity (see Fig. 1).

The arc length $s = \phi r$, where ϕ is the azimuthal angle of the contact line and r is the sphere radius. The angle ϕ is governed by the balance of gravity, buoyancy, and surface tension acting on the sphere. The balance of these forces is described by the following equation[3, 4, 5]: $6 \sin \phi \sin(\phi - \theta) - (\cos^3 \phi - 3 \cos \phi - 2)(\rho - 1)Bo = 0$, where θ is the sphere's contact angle, ρ is the density ratio of the solid sphere to the water, and Bo is a Bond number defined by $Bo = \rho_l r^2 g / \sigma$. Here, ρ_l is the density of water, g is the gravitational acceleration, and σ is the surface tension. The angle ϕ is implicitly a function of Bo , ρ , and θ , expressed as $\phi = \phi(Bo, \rho, \theta)$.

The liquid film moves at a velocity $V_f \sim V_l - V_s$ [6, 1], where V_l and V_s denote the velocities of the water surface and the sphere, respectively. Assuming the spark bubble expands spherically, V_l can be estimated through $V_l h^2 \sim V_b R_m^2$, which yields $V_l \sim V_b \hat{h}^{-2}$. The sphere velocity V_s was determined through a force analysis. We considered the dominant forces acting on the sphere, including the liquid inertia and surface tension forces, which yields $V_s \sim (C_h + 4 \sin^2 \frac{\theta}{2} We^{-1}) \rho^{-1} V_l$ (a detailed derivation is included in the revised manuscript). Here, We is the Weber number, $We = \rho_l V_l^2 r / \sigma$. C_h is the hydrodynamic force coefficient, which is on the order of unity[7, 8, 9].

By substituting V_l and V_s into t_c , the dimensionless immersion duration is derived,

$$t_c \sim \frac{\phi \hat{r}}{\alpha} \hat{h}^2 \sim \frac{\phi(Bo, \rho, \theta) \hat{r}}{1 - (C_h + 4 \sin^2 \frac{\theta}{2} We^{-1}) \rho^{-1}} \hat{h}^2, \quad (1)$$

Figure 1: Immersion processes for spheres with either hydrophilic (**a**, blue circles) or hydrophobic (**b**, orange circles) surfaces during the spark bubble's expansion phase ($t \leq \tau$). These represent the first branching process of jet modes, determined by whether a singular jet forms before $t = \tau$. Solid lines indicate the air-water interfaces.

where \hat{r} is the radius ratio between the sphere and the spark bubble, $\hat{r} = r/R_m$. The factor α is expressed as $\alpha \sim 1 - (C_h + 4 \sin^2 \frac{\theta}{2} We^{-1}) \rho^{-1}$.

Equation 1 illustrates that t_c is dependent on the sphere radius ($t_c \propto \hat{r}$). A larger sphere radius results in a longer length for the liquid film to travel, further leading to an increased immersion duration t_c .

Comment 2: From the description of the regimes it seems like the relevant time scale is T_1 rather than τ . Since T_1 is roughly 2τ the limit at $t_c = 0.2$ would be closer to $t_c^u = T_1$.

Thank you for your feedback. In this study, we observed the first branching process of jet modes during the expansion phase of the spark bubble ($t \leq \tau$), which depends on whether a singular jet generates. A singular jet forms when the liquid film above the sphere converges (i.e., the sphere is fully submerged; see Fig. 1).

To characterise the first branching process, we introduced the sphere's immersion time, t_c^u . The critical condition for this process is that the sphere becomes fully submerged

Figure 2: The plot of $\phi\hat{r}/\alpha$ ($= t_c/\hat{h}^2$) versus \hat{h} categorises the five jet modes. \hat{h} is the spark bubble's dimensionless depth. t_c is the dimensionless immersion time, defined as the ratio of the sphere's immersion time to the spark bubble's expansion time. The formula $t_c = \phi\hat{r}\hat{h}^2/\alpha$ incorporates the azimuthal angle ϕ of the contact line and the radius ratio \hat{r} between the sphere and the spark bubble. The factor α is determined by the density ratio ρ , Weber number We , and contact angle θ , expressed as $\alpha = 1 - (1 + 4\sin^2\frac{\theta}{2}We^{-1})\rho^{-1}$. The oblique line marks the boundary at $t_c = 0.24$, distinguishing jet modes with singular jets (Modes I, II, and III.a below the line) from those without singular jets (Modes III.b, IV, and V above the line). Additional lines are included to indicate the boundaries between different jet modes.

just before the spark bubble collapses, i.e., $t_c^u = \tau$. By normalising t_c^u with τ , we obtain the dimensionless immersion duration, $t_c = t_c^u/\tau$. Theoretically, if $t_c \leq 1$, the sphere is fully submerged before the spark bubble collapses, resulting in Modes I, II or III.a with singular jets. Conversely, if $t_c > 1$, the sphere remains unimmersed at $t = \tau$, leading to Modes III.b, IV or V without singular jets at $t = \tau$. Therefore, it is more physically appropriate using τ as the relevant time scale than T_1 .

Since t_c is influenced by \hat{h} (Eq. 1), to decouple these two parameters, we constructed the phase diagram based on \hat{h} and $\phi\hat{r}/\alpha$ (where $\phi\hat{r}/\alpha = t_c/\hat{h}^2$, with $C_h = 1$ in all cases to calculate α). In Fig. 2, the oblique line represents the boundary of modes with and without singular jets at $t = \tau$. It corresponds to the first branching process, applicable to experiments using spheres with various wettabilities. Along this line, $t_c = 0.24$. It

is lower than the theoretical value of $t_c = 1$. This discrepancy may be attributed to an overestimation of the liquid film velocity V_f by using $(V_l - V_s)$ when calculating t_c , due to the absence of a coefficient for accurate quantification of V_f . Despite this, the phase diagram effectively delineates the first branching process without requiring an additional fitting factor. This demonstrates that our theoretical model using t_c successfully distinguishes between the jet modes.

Comment 3: The two parameters t_c and \hat{h} are not independent. The slope of the separation between regimes III and I simply reflects this dependence as clearly seen from all data points. There is more to understand to have an efficient collapse of the data and separation of the regimes. A log-log plot and power law scaling are hard to interpret when there is less than half a decade.

Thank you for your comments. In response to your concerns, we conducted a dimensional analysis to identify the appropriate dimensionless parameters for constructing the phase diagram. Through dimensional analysis, we simplify the system in this study into seven dimensionless parameters, which are the top seven dimensionless numbers listed in Tab. 1. They are the Weber number We , representing the ratio of liquid inertia to capillary force, the Froude number Fr , which compares liquid inertial to gravitational force, the Reynolds number Re , indicating the ratio of liquid inertia to viscous force, the density ratio ρ , the contact angle θ , the radius ratio between the sphere and the spark bubble, \hat{r} , and a dimensionless depth, h^* . The table also includes the Bond number $Bo = We/Fr^2$, which compares gravitational to capillary force, as well as the spark bubble's dimensionless depth, $\hat{h} = h/R_m = h^*\hat{r}$, a parameter commonly used in studies on cavitation bubble interactions with flat water surfaces[10, 11]. In this study, we use \hat{h} instead of h^* .

Table 1 summarises the range of these parameters within the study. These parameters capture the interactions between inertial, viscous, surface tension, and gravitational forces, along with the geometric and interfacial properties that affect jet formation. We found that using any combination of two among these parameters was insufficient to distinctly classify the various jet modes observed in this study.

In this study, we selected t_c and \hat{h} for the phase diagram as they have clear physical meanings. These two dimensionless parameters characterise the two branching processes of jet modes. From experiments, we observed two primary branching processes from a sphere resting on the water surface to five distinct jet modes. The first branching process occurs during the expansion phase of the spark bubble ($t \leq \tau$), depending on whether a singular jet forms. The second branching process occurs during the collapse phase of the

Dimensionless numbers	Symbol	Definition	Range
Weber number	We	$\frac{\rho_l V_l^2 r}{\sigma}$	2-6126
Froude number	Fr	$\frac{V_l}{\sqrt{gr}}$	30-412879
Reynolds Number	Re	$\frac{\rho_l V_l r}{\mu}$	112-14938
Density ratio	ρ	$\frac{\rho_s}{\rho_l}$	1.4-8.5
Contact angle	θ	θ	$81^\circ - 154^\circ$
Radius ratio	\hat{r}	$\frac{r}{R_m}$	0.01-0.27
Dimensionless depth	h^*	$\frac{h}{r}$	6-268
Bond number	Bo	$\frac{\rho_l r^2 g}{\sigma} = \frac{We}{Fr^2}$	0.001-0.8
Spark bubble's dimensionless depth	\hat{h}	$\frac{h}{R_m} = h^* \hat{r}$	0.6-3.5

Table 1: Relevant dimensionless parameters and their characteristic values.

spark bubble ($\tau \leq t \leq T_1$), the cavity expands and further evolves into five distinct jet modes.

To characterise the first branching process, we introduced the dimensionless chasing time t_c . It represents the ratio of the sphere's immersion time to the spark bubble's expansion time. Theoretically, if $t_c \leq 1$, the sphere becomes fully submerged before the spark bubble collapses, resulting in Modes I, II or III.a with singular jets. Conversely, if $t_c > 1$, the sphere remains unimmersed at $t = \tau$, leading to Modes III.b, IV or V without singular jets.

The second branching process is primarily governed by the cavity dynamics. Driven by the negative radiated pressure from the spark bubble, the concavity at the junction between the singular jet and the water bulge in Modes I, II, and III.a, as well as the pit above the sphere in Modes III.b, IV, and V, expands into downward cavities. As shown in Fig. 3, the cavity expansion velocity, V_c , is primarily determined by \hat{h} . Therefore, \hat{h} is a suitable parameter for characterising the second branching process. Additionally, \hat{h} is widely used in studies on cavitation bubble interactions with a flat water surface, where distinct surface jet modes arise at different values of \hat{h} [10, 11].

Since t_c is influenced by \hat{h} (Eq. 1), to decouple these two parameters, we constructed the phase diagram based on \hat{h} and $\phi \hat{r} / \alpha$ (where $\phi \hat{r} / \alpha = t_c / \hat{h}^2$, with $C_h = 1$ in all cases to calculate α). As shown in Fig. 2, the five jet modes cluster distinctly. The oblique line represents the boundary of modes with and without singular jets at $t = \tau$. It

Figure 3: The dimensionless cavity expansion velocity, V_c/V_b , versus the dimensionless spark bubble depth, \hat{h} . The solid line represents theoretical predictions.

corresponds to the first branching process, applicable to experiments using spheres with various wettabilities. Along this line, $t_c = 0.24$. It is lower than the theoretical value of $t_c = 1$. This discrepancy may be attributed to an overestimation of the liquid film velocity V_f by using $(V_l - V_s)$ when calculating t_c , due to the absence of a coefficient for accurate quantification of V_f . Despite this, the phase diagram effectively delineates the first branching process without requiring an additional fitting factor. This demonstrates that our theoretical model using t_c successfully distinguishes between the jet modes.

Vertical lines at $\hat{h} \approx 1.15, 1.3, \text{ and } 2.1$ demarcate the boundaries between Modes I and II, III.b and IV, and IV and V, respectively. In the subsequent responses to ‘Comment 4’, we included discussions on the critical conditions for these mode transitions, along with corresponding theoretical predictions.

Additionally, we observed a horizontal line at $\phi\hat{r}/\alpha \approx 0.15$ that separates Modes I and III.a, and this boundary is independent of \hat{h} (Fig. 2). To investigate this further, we replotted the phase diagram for these two modes based on ρ and θ , as shown in Fig. 4. Mode III.a tends to occur with spheres that have larger contact angles and lower densities, while Mode I is more likely with spheres having smaller contact angles and higher densities. Note that wetting failure tends to occur on surfaces with lower wettability, leading to the formation of surface cavities[1, 12, 13, 2]. Additionally, the size of the sealed air bubble increases as the contact angle increases[12, 2]. We hypothesise that the sealed air bubble trapped between the concavity and the sphere triggers the cavity venting phenomenon observed in Mode III.a (see b1 in Fig. 1). When the downward-moving concavity ruptures the liquid film between its base and the upper wall of the sealed air

Figure 4: Phase plot of Modes I (grey squares) and III.a (blue triangles) based on the density ratio ρ and contact angle θ .

bubble, it causes a rapid cavity expansion as in Mode III.b, creating a channel between the spark bubble and the atmosphere.

Another factor influencing the transition between Modes I and III.a is the sphere's density. A sphere with a lower density moves more quickly during the spark bubble's expansion phase, reducing the thickness of the liquid film between the concavity and the sealed air bubble at $t = \tau$. This thinner liquid film is more easily ruptured as the concavity expands, facilitating cavity venting.

Due to experimental constraints, including the intense light produced by the spark bubble and the obscuring effects of curved interfaces, we are currently unable to track the sphere's movement and the surrounding air-liquid interfaces during the initial phase. This restricts our ability to provide sufficient quantitative data to support our analysis on the mode transition for Modes I and III.a. Although a detailed theory to explain these results is still lacking, we propose an effective method to delineate the boundary, suggesting the importance of sphere properties and wetting failure phenomena in influencing the mode transition. Future work will involve numerical simulations to gain a deeper understanding of the cavity venting phenomenon observed in Mode III.a.

Comment 4: While it is good to have some quantitative description of the regimes, Fig. 3 is hard to understand. There are more to extract from these measurements. For example, the jet velocity with or without particle seems to depend on the dimensionless depth \hat{h} . Could you plot velocity vs \hat{h} for the different modes? Would it help understand the evolution of the depth of the cavity? Fig. 3f is not discussed thoroughly in the main

text and it is hard to use the information it provides to gain an understanding of the physical mechanisms.

Thank you for your valuable suggestions. In response to your feedback, we conducted additional quantitative measurements, including the cavity expansion velocity V_c , water surface velocity V_l , and jet velocity V_j , across various values of \hat{h} . We also included theoretical predictions for V_l and V_c . We found that the second branching process of jet modes is primarily driven by the cavity dynamics, with these measurements supporting our theoretical predictions for jet mode transitions. These additions enhance our understanding of the physical mechanisms behind the different jet modes. Below are our detailed responses to your comments.

The cavity expansion velocity V_c

We measured the average cavity expansion velocity V_c from $t = \tau$ to T_1 , as illustrated in Fig. 3. The results show that V_c is primarily determined by \hat{h} , suggesting that \hat{h} is an appropriate parameter for characterising the cavity dynamics, further determining mode transitions during the second branching process.

To estimate V_c , we analyse the physical process during the spark bubble's collapse. The spark bubble is centred at a depth h . At time $t = \tau$, the bubble reaches its maximum radius R_m . We consider a spherical shell of water extending from R_m to h at the water surface. As the spark bubble collapses over the duration τ_{coll} (approximated as τ [14, 10, 11]), the cavity begins to expand downward at a velocity V_c from the water surface to a depth of $V_c\tau_{coll}$. This collapsing causes the spherical shell to contract, reducing its inner radius from R_m to zero, and outer radius from h to $h - V_c\tau_{coll}$. With the conservation of water volume within this spherical shell, we establish the relationship: $h^3 - R_m^3 \sim (h - V_c\tau_{coll})^3$, which yields

$$V_c \approx b \left(\hat{h} - \sqrt[3]{\hat{h}^3 - 1} \right) V_b. \quad (2)$$

As shown in Fig. 3, the theoretical predictions align well with the experimental data, with a best fitting prefactor of $b = 4.35$.

The quantitative measurements on V_c guide the theoretical predictions on the second branching process of jet modes. In Fig. 3, two critical values, $\hat{h} \approx 1.3$ and 2.1 , differentiate Modes III.b and IV, and Modes IV and V, respectively. These suggest that the transition among these modes are primarily governed by V_c . When V_c is sufficiently high, the cavity reaches the spark bubble before $t = T_1$, a channel forms, resulting in Mode III.b. As V_c decreases at larger values of \hat{h} , the cavity seals at its neck, leading to Mode IV. Further decreases in V_c prevent cavity sealing, resulting in Mode V.

The boundary between Modes III.b and IV

In Fig. 2, the regime map shows a vertical line at $\hat{h} \approx 1.3$, which separates Modes III.b and IV. Mode III.b is observed for smaller values of \hat{h} , while larger values correspond to Mode IV. This is consistent with the experimental finding that Mode III.b exhibits a larger cavity expansion velocity than Mode IV (see Fig. 3). The critical condition separating these two modes occurs when the cavity bottom reaches the spark bubble at $t = T_1$. Disregarding the uplift of the water surface at $t = \tau$ and the migration of the spark bubble's centroid at $t = T_1$, we find $V_c \tau_{coll} \approx h$. This yields a critical value of $\hat{h} \approx 1.23$, which closely matches the experimental result of $\hat{h} \approx 1.3$.

The boundary between Modes IV and V

The distinction between Modes IV and V depends on whether the cavity rim seals at $t = T_1$. This sealing process resembles the surface sealing phenomenon observed in water entry problems[15, 13, 16]. As the spark bubble collapses, the cavity expands, allowing air to flow in and fill the void. Based on the geometric relationship, the air flow speed through the cavity rim is estimated as $V_a \sim (r_c/r_{rim})^2 V_c$, where r_c and r_{rim} denote the radii of the cavity and its rim, respectively.

According to Bernoulli's principle, this inflowing air induces an aerodynamic pressure difference across the rim, $\Delta p \propto -\rho_a V_a^2$, where ρ_a is the air density. This pressure difference drives the collapse of the cavity rim, and a modified Rayleigh-Plesset equation can be obtained[15, 16]: $\rho_l \left(r_{rim} \ddot{r}_{rim} + \frac{3}{2} \dot{r}_{rim}^2 \right) = \Delta p$. Further, the collapse duration of the rim is derived, $\tau_{rim} \sim r_{rim}^0 \sqrt{\rho_l / |\Delta p|}$, where r_{rim}^0 is the initial radius of the cavity rim. If $\tau_{rim} \leq \tau_{coll}$, the cavity seals before the spark bubble rebounds, resulting in Mode IV. Otherwise, the cavity remains open at $t = T_1$, leading to Mode V. The value of τ_{rim} is primarily governed by Δp , which depends on \hat{h} . This suggests a critical \hat{h} that determines cavity sealing, consistent with experimental observations showing that the vertical line at $\hat{h} \approx 2.1$ marks the boundary between Modes IV and V.

The boundary between Modes I and II

In the case where the singular jet forms before $t = \tau$, the concavity at the junction between the singular jet and the water bulge moves downward and develops into a cavity as the spark bubble collapses. The critical condition distinguishing Modes I and II arises when the concavity reaches the sphere at $t = T_1$. At $t = \tau$, two distances affect the transition between these two modes: the distance between the concavity and the

Figure 5: The dimensionless water surface velocity on flat water surfaces, V_l/V_b , follows a power-law relationship with \hat{h} , $V_l/V_b \sim \hat{h}^{-2}$.

sphere, $d_1 \approx (V_l - V_s)\tau$, and the distance between the concavity and the spark bubble, $d_2 \approx V_l\tau + h$. Here, d_1 represents the initial gap the concavity needs to close to reach the sphere, and d_2 determines the downward speed of the concavity, V_c . Ignoring the sphere's motion during the spark bubble's collapse phase, we find $V_c\tau_{coll} \approx d_1$, yielding a critical value of $\hat{h} \approx 1.04$ for $C_h = 1$, $We \gg 1$, and $\rho \rightarrow \infty$. This theoretical value is close to the experimental result of $\hat{h} \approx 1.15$ that separates Modes I and II.

The water surface velocity V_l

We measured the average water surface velocity V_l from $t = 0$ to τ , as illustrated in Fig. 5. The normalised water surface velocity, V_l/V_b , follows a power-law relationship with \hat{h} . Assuming the spark bubble expands spherically, V_l can be estimated through $V_l h^2 \sim V_b R_m^2$, which yields $V_l = a V_b \hat{h}^{-2}$. The theoretical predictions align well with the experimental data, with a best-fitting prefactor of $a = 0.64$.

The jet velocity V_j

Figure 6 presents the measured jet velocities, including the singular jet velocities in Modes I, II, and III.a, the primary jet velocities in Modes II and V, and the primary jet velocities on flat water surfaces. The velocities of both the singular and primary jets exceed those of flat water surfaces at the same \hat{h} , particularly for the primary jets in Modes II and V. The jet velocity from flat water surfaces is governed by the dimensionless depth of the spark bubble, decreasing to nearly zero as \hat{h} approaches approximately

Figure 6: Dimensionless jet velocity V_j/V_b versus the dimensionless depth of the spark bubble, \hat{h} . The singular jet velocity in Modes I, II, and III.a, the primary jet velocity in Modes II and V, and the primary jet velocity on flat water surfaces are included.

1.2. In contrast, in the experimental group, the jet persists until \hat{h} reaches around 3.5, significantly extending the effective range of \hat{h} for jet generation. Additionally, we found that in the experimental group, the velocities of both singular and primary jets are primarily controlled by \hat{h} , and the sphere properties also have an impact on the jet velocity. This is beyond the scope of the current study and will be discussed further in future research.

Comment 5: Minor remark: ventilation is still an odd name that does not describe well the fact that there is an airflow from the cavity.

Thank you for your feedback. We have renamed Mode III from ‘Ventilation’ to ‘Cavity Venting’ to better represent the process in which airflow from the atmosphere flows through the cavity-induced channel into the underlying spark bubble.

References

- [1] Cyril Duez, Christophe Ybert, Christophe Clanet, and Lydéric Bocquet. Making a splash with water repellency. *Nature Physics*, 3(3):180–183, 2007.
- [2] Nathan B. Speirs, Mohammad M. Mansoor, Jesse Belden, and Tadd T. Truscott. Water entry of spheres with various contact angles. *Journal of Fluid Mechanics*, 862, 2019.

- [3] D. Vella, D. G. Lee, and H. Y. Kim. The load supported by small floating objects. *Langmuir*, 22(14):5979–5981, 2006.
- [4] C. W. Extrand and S. I. Moon. Using the flotation of a single sphere to measure and model capillary forces. *Langmuir*, 25(11):6239–44, 2009.
- [5] D. Vella. Floating versus sinking. *Annual Review of Fluid Mechanics*, 47(1):115–135, 2015.
- [6] James Mark Oliver. *Water entry and related problems*. Thesis, University of Oxford, 2002.
- [7] M. Moghisi and P. T. Squire. An experimental investigation of the initial force of impact on a sphere striking a liquid surface. *Journal of Fluid Mechanics*, 108:133–146, 1981.
- [8] B. Ji, Q. Song, K. Shi, J. Liu, and Q. Yao. Oblique impact of microspheres on the surface of quiescent liquid. *Journal of Fluid Mechanics*, 900, 2020.
- [9] B. Ji, Z. Tang, and Q. Song. Oblique impact dynamics of micron particles onto a liquid surface. *Physical Review Fluids*, 5(11), 2020.
- [10] S. Zhang, S. P. Wang, and A. M. Zhang. Experimental study on the interaction between bubble and free surface using a high-voltage spark generator. *Physics of Fluids*, 28(3):032109, 2016.
- [11] Y. J. Kang and Y. Cho. Gravity-capillary jet-like surface waves generated by an underwater bubble. *Journal of Fluid Mechanics*, 866:841–864, 2019.
- [12] Minh Do-Quang and Gustav Amberg. The splash of a solid sphere impacting on a liquid surface: Numerical simulation of the influence of wetting. *Physics of Fluids*, 21(2), 2009.
- [13] Tadd T. Truscott, Brenden P. Epps, and Jesse Belden. Water entry of projectiles. *Annual Review of Fluid Mechanics*, 46(1):355–378, 2014.
- [14] J. P. Franc and J. M. Michel. *Fundamentals of Cavitation*, pages p.36–39. Springer, Dordrecht, 2005.
- [15] J. M. Aristoff and J. W. M. Bush. Water entry of small hydrophobic spheres. *Journal of Fluid Mechanics*, 619:45–78, 2009.
- [16] J. Eshraghi, S. Jung, and P. P. Vlachos. To seal or not to seal: The closure dynamics of a splash curtain. *Physical Review Fluids*, 5(10), 2020.

Responses to Reviewers' Comments for Manuscript NCOMMS-24-09227A

Particulate Reshapes Surface Jet Dynamics Induced by a Cavitation Bubble

Addressed Comments for Manuscript to

by

Xianggang Cheng, Xiao-Peng Chen, Zhi-Ming Yuan and Laibing Jia

Authors' Response to Reviewer 5

General Comments: I co-reviewed this manuscript with one of the reviewers who provided the listed reports. This is part of the Nature Communications initiative to facilitate training in peer review and to provide appropriate recognition for Early Career Researchers who co-review manuscripts.

Thank you for providing this information. We are grateful for your participation in the peer review process and for your contributions to enhancing the quality of our manuscript. We value the insights and feedback from you and your co-reviewer, and we have made point-to-point revisions to the manuscript in response to your suggestions.

In the revised manuscript, we have made the following major changes:

1. The dynamic wetting process of the sphere during immersion has been analysed, considering sphere surface wettabilities and wetting failure phenomena. Additionally, a thorough theoretical analysis of the sphere's dimensionless immersion time t_c has been included.
2. Quantitative measurements and theoretical predictions, including water surface velocity and cavity expansion velocity, have been incorporated. Cavity dynamics, primarily influenced by \hat{h} (the spark bubble's dimensionless depth), were identified as the governing factor in the second branching process of jet modes.
3. Building on insights from the two branching processes of jet modes, more detailed theoretical analyses of the boundaries between different jet modes were conducted.
4. Figure 5b of the previous manuscript was integrated into Fig. 4 to better illustrate the two branching processes in the evolution of the jet modes.
5. Jet mode III has been renamed from 'Ventilation' to 'Cavity Venting' to better represent the process in which airflow from the atmosphere flows through the cavity-induced channel into the underlying spark bubble as advised by a reviewer.
6. A dimensional analysis to identify the governing dimensionless parameters and a theoretical analysis of the sphere dynamics to derive its characteristic velocity are added in the Methods section.

Responses to Reviewers' Comments for Manuscript NCOMMS-24-09227B

Particulate Reshapes Surface Jet Dynamics Induced by a Cavitation Bubble

Addressed Comments for Manuscript to

by

Xianggang Cheng, Xiao-Peng Chen, Zhi-Ming Yuan and Laibing Jia

Dear Editor and Reviewers,

We sincerely thank you for your careful review and constructive comments. Your feedback has been invaluable in improving the clarity and scientific rigour of the manuscript. In the revised version, we have made the following major changes in direct response to your comments:

1. Immersion and jetting dynamics refined

- The immersion process has been clarified by replacing *liquid film* with *liquid layer* and contrasting the dynamics with classical water-entry.
- The physical basis for the immersion time t_c and its critical value t_c^* are now provided.
- The suppression of singular jets at small sphere sizes is analysed via Ohnesorge and Capillary numbers, demonstrating surface tension dominance.
- The discussion on the Mode I-III.a boundary has been expanded to emphasize the role of sphere properties, acknowledge current experimental limitations, and propose potential approaches to address this open question.

2. Cavity sealing analysis extended

The cavity sealing transition between Modes IV and V is now described by introducing the aerodynamic-to-Laplace pressure ratio We^* , suggesting aerodynamic pressure dominance.

3. Methods clarification and minor revisions

The definition of the cavity expansion velocity V_c is clarified as the linear slope from $t = \tau$ to $t = T_1$. The radius of the rim itself is now defined as a , and its distance to the cavity axis centre is changed $r_{\text{rim}} \rightarrow b$. Figure 4 is updated with legend to distinguish hydrophilic and hydrophobic spheres.

4. Supplementary material additions

Supplementary Figures 1-10 have been added to provide detailed supporting data, snapshots, and phase maps corresponding to the above revisions.

In the following pages, we provide point-by-point responses to all issues raised. We hope the revisions have satisfactorily addressed all concerns and improved the manuscript accordingly.

Thank you again for your time and consideration.

Sincerely,

Laibing Jia (on behalf of all authors)

Authors' Response to Reviewer 3

General Comments: The authors provide an explanation using the water entry theory to support the criterion for the first branch, but this theory still cannot well explain the formation of all the final five jet modes. I am still worried that the dimensionless immersion time is not an initial parameter and lacks universality. Without rigorous theoretical arguments, as well as a broad enough impact, my feeling is, therefore, the manuscript would be better suited to a specialized journal in its present form. I believe it could be improved further by considering the following questions.

Thank you for your constructive review and insightful comments. In this revision, we have carefully addressed the three key issues you raised: (i) the applicability of classical water-entry theory to the five jet modes, (ii) the universality and intrinsic nature of the dimensionless immersion time t_c , and (iii) the clarity and broader impact of our findings. Our detailed responses are provided below.

(i) Applicability of water-entry theory to five jet modes. We appreciate your critical observation regarding our use of water-entry theory. The classical water-entry theory was introduced as a qualitative analogy for the first branch. However, due to differences in both geometry and driving mechanisms between our experiments and classical water-entry problems, this analogy does not extend to the second branch, and cannot account for the formation of the five final jet modes.

- In classical water entry, a solid object impacts and penetrates a quiescent liquid surface. The initial contact between the sphere and the liquid surface induces strong radial spreading of a splashing liquid film. The resulting cavity is primarily driven by the solid's motion, with pinch-off often occurring below the free surface.
- In contrast, in our configuration, the sphere initially rests in a partially submerged equilibrium at the interface, with no downward penetration through the free surface as seen in classical water-entry problems. Instead, the liquid layer rises along/near the sphere, driven by the expansion of the spark bubble, without obvious radial splashing (**Supplementary Figure 6**).
- During the early collapse stage of the spark bubble, negative pressure drives the cavity evolution. The cavity dynamics observed in our experiments, driven by the spark bubble, differ substantially from the cavity pinch-off behaviour in classical water-entry problems.

To clarify these distinctions and avoid potential confusion, we have revised the manuscript accordingly, providing explicit statements in the updated Fig. 4 and its accompanying text in the section "Two branching processes of jet modes".

(ii) Universality and intrinsic nature of the dimensionless immersion time t_c . t_c is a universal and intrinsic parameter determined *a priori*, as it depends only on the initial experimental configuration. The reasoning is as follows:

Since the entire process is driven by the evolution of the spark bubble, we define t_c as the ratio of the sphere’s immersion time t_c^u to the spark bubble’s expansion duration τ . Physically, it also characterises the ratio between the unimmersed arc length of the sphere and the distance travelled by the liquid layer during τ ,

$$t_c = \frac{t_c^u}{\tau} = \frac{s}{v\tau}. \quad (1)$$

- The arc length s of the initially unimmersed sphere portion is determined by initial geometry and material parameters: Bond number Bo , density ratio ρ , contact angle θ , and sphere radius r .
- The relative speed $v = V_l - V_s$ depends on:
 - V_l : the velocity of the water surface driven by spark bubble expansion, governed by the dimensionless spark bubble depth \hat{h} .
 - V_s : the sphere’s velocity, determined by initial parameters We , ρ , and θ .
- The characteristic spark bubble time scale τ is determined by the spark bubble’s maximum radius R_m and external pressure conditions.

The specific expression for t_c is given in Eq. (4) of the manuscript. A Buckingham- Π dimensional analysis, using ρ_l , V_l , and r as repeating variables, shows that t_c depends on a set of dimensionless parameters:

$$t_c = F\left(Bo, \rho, \theta, \hat{r}, \hat{h}, We\right), \quad (2)$$

where all parameters are determined by the initial configuration of the sphere and spark bubble.

We have revised the manuscript in the section “Governing parameters of jet modes” and clarified the intrinsic universality of t_c in the discussion of Fig. 5.

(iii) Clarity and broader impact of our findings. This work carries both fundamental and practical significance, revealing a previously unrecognised energy-focusing mechanism induced by tiny surface defects, with broad potential for applications across multiple fields.

Our experiments uncover *new physics* in jet dynamics driven by spark bubble-particulate interactions: tiny surface defects created by floating particulates serve as energy-focusing sites, lowering the jet-formation threshold by an order of magnitude and giving rise to five previously unreported jet modes. This rich spectrum of phenomena emerges from deceptively simple mechanisms, the interplay between a spark bubble and a single surface defect, demonstrating how small-scale heterogeneities can reshape free-surface dynamics.

Beyond advancing fundamental principles in interfacial physics and multiphase flow dynamics, these findings offer potential applications such as controlling aerosol release of pollutants and pathogens, tailoring droplet size in biomedical cavitation, and improving the precision of needle-free injection jets.

We have discussed these advances and application prospects in the revised “Conclusion and Discussion” section to underline both fundamental physics and engineering applications.

Thank you again for your constructive feedback, which has helped us to improve both the clarity and quality of the manuscript.

Following are our detailed responses to your comments.

Comment 1: It is oversimplified for the derivation of t_c . The travel distance ζ' s here should be the pinch-off depth of sphere in water entry problem (Aristoff & Bush 2008 JFM). It was reported that this value can be much higher than the sphere size (ζ' can be much larger than 1 in deep seal mode). Is it reasonable to consider ζ'/ζ as order of unity?

Thank you for bringing up this point. As discussed in our response to the general comment, our scenario differs fundamentally from classical water-entry problems.

(i) Physical distinction from classical water entry. In classical water-entry, the initial contact between the sphere and the liquid surface induces strong radial spreading of a splashing liquid film. In contrast, our configuration lacks such initial solid-liquid impact and is governed instead by the internal pressure variations of the oscillating spark bubble. These distinct driving mechanisms lead to different flow dynamics. Consequently, the classical cavity pinch-off mechanism is not applicable to our system.

(ii) Experimental confirmation of the travel distance. In our case, the travel distance of the upward-moving liquid layer is limited by the short spark bubble expansion duration τ (approximately 1 ms in all our experiments). Although we were unable to precisely measure the value of ζ'/ζ due to optical distortion caused by the curved air-water interface, time-resolved image sequences consistently show that the upward travel distance of the liquid layer is comparable to the sphere size (**Supplementary Figure 6**). We did not observe any cases of deep cavity elongation similar to the deep-seal mode reported in classical water-entry problems.

We have clarified this point in the revised manuscript and provided Supplementary Figure 6 to support our statement that ζ'/ζ remains of order unity in our configuration.

Comment 2: When describing the particles immersion, “liquid film” is odd and I cannot understand what it refers to. The closure of the cavity above the particle can be resulted by the pinch off of the cavity instead of a liquid film.

Thank you for raising this point. In the previous version, the term “liquid film” was intended to describe the fluid structure that moves upward near/along the sphere during the expansion of the spark bubble and contributes to the immersion of the sphere (**Supplementary Figure 6**). This terminology was borrowed from the classical water entry literature (Duez et al., 2007[1]), where it refers to the splashing layer. However, we realise that this usage may lead to confusion in the present context.

Supplementary Figure 6: Experimental snapshots illustrate the formation and development of the liquid layer during the expansion phase of the spark bubble. (see full caption in Supplementary Information)

As discussed in our above responses, the immersion process in our study differs from classical water entry scenarios, and the pinch-off mechanism commonly described the closure of a cavity in that context does not capture the liquid layer convergence and closure observed during $t \leq \tau$.

To avoid confusion, we have replaced the term “liquid film” with “liquid layer” throughout the manuscript and clarified its meaning in the section “Two branching processes”. Supplementary Figure 6 has also been annotated to illustrate this layer.

Comment 3: How do the authors measure the velocity V_c in Fig. 3h? Is it the average velocity during the whole expansion process?

V_c denotes the characteristic expansion velocity of the cavity, defined as the slope of a linear regression fitted to the cavity bottom position $z(t)$ over the interval $\tau \leq t \leq T_1$. As shown in **Supplementary Figures 1-5**, $z(t)$ exhibits an approximately linear trend over this interval, and the fitted slope is taken as the value of V_c .

In the experiments, side-view image sequences were processed to extract the cavity contour using the Canny edge-detection algorithm. The cavity bottom position $z(t)$ at each frame was identified by selecting the lowest edge point within the detected contour. To reduce sensitivity to occasional outliers, the linear regression was performed using the `robustfit` function in MATLAB.

We have included a sentence describing this measurement procedure in the section “Kinematics of jets and cavities” of the revised manuscript.

Comment 4: Line 499: the phase diagram effectively delineates the first branching process without requiring an additional fitting factor. 0.24 itself is a fitting parameter that smaller than 1 by one order of magnitude.

Thank you for pointing this out. We agree that the wording in the earlier version may have led to confusion and offer the following clarification.

The critical dimensionless immersion time $t_c^* = 0.24$ is not a fitting parameter in the usual sense, but rather an experimentally determined critical value arising from dimensional analysis. t_c^* is not introduced or adjusted during model construction or data analysis to improve the agreement between the model and the data. Instead, its value is determined experimentally under the current system configuration and remains fixed once the physical parameters of the system are specified.

In our study, dimensional analysis yields the scaling $t_c \sim \phi \hat{r} \hat{h}^2 / \alpha$, but does not determine the numerical prefactor. The critical value $t_c^* = 0.24$, which separates regimes with and without singular jet formation, was directly identified from experimental measurements. This threshold emerged independently across multiple experimental geometries and configurations, consistently separating singular from non-singular regimes without further adjustment.

The observed deviation of t_c^* from unity reflects the combined effects of system geometry and relative speed between the sphere and surrounding liquid, which contribute to

the prefactor but not captured by dimensional analysis. Such deviations are common in fluid mechanics, where experimentally determined critical values often emerge even for well-defined dimensionless groups. For example, the Reynolds number defines the ratio of inertial to viscous forces, yet the transition from laminar to turbulent flow in pipe flow occurs at a critical Reynolds number of approximately 2300 (Reynolds, 1883[2]; Schlichting and Gersten, 2017[3]), and the onset of vortex shedding behind a circular cylinder occurs at approximately 49 (Williamson, 1996[4]).

In the revised manuscript, we have revised the discussion on the critical threshold of t_c , removed the phrase “without requiring an additional fitting factor” to avoid misunderstanding, and clarified that $t_c^* = 0.24$ is an experimentally determined constant arising from the dimensional analysis framework.

Comment 5: I cannot understand the derivations when discussing the boundary between Modes IV and V. Did the author neglect the contribution of Laplace pressure?

Thank you for pointing out the need to clarify our reasoning regarding the boundary between Modes IV and V, particularly concerning the role of Laplace pressure.

Figure for Comment 5: A sketch for the cavity. The grey circle denotes the solid sphere. The black curve denotes the air-water interface. The radius of the rim itself is a , and its distance to the cavity axis is b .

We evaluate the aerodynamic pressure drop across the cavity rim, Δp , with the Laplace pressure p_s induced by surface tension. According to Bernoulli’s principle, $\Delta p \sim \rho_a V_a^2$, where ρ_a is the air density and V_a is the velocity of airflow entering the cavity. The Laplace pressure is given by $p_s = \sigma(\frac{1}{a} - \frac{1}{b})$, where a is the radius of the rim itself, and b is the distance from the cavity axis to the rim (see **Figure for Comment 5**. In response to a reviewer’s suggestion, we have replaced the original notation r_{rim} with b to avoid confusion). Based on the geometric characteristics of the liquid rim in Modes IV and V, both a and b are on the order of r , leading to an estimation of $p_s \sim \sigma/r$.

A local Weber number is defined as,

$$We^* = \frac{\Delta p}{p_s} \sim \frac{\rho_a V_a^2 r}{\sigma}. \quad (3)$$

The estimation of We^* is detailed in the Methods section “Aerodynamic pressure vs. Laplace pressure in Modes IV and V”. As shown in **Supplementary Figure 7**, We^* ranges from $O(10^1)$ to $O(10^5)$ at the boundary between Modes IV and V. This

Supplementary Figure 7: Phase diagram of Modes IV and V plotted against \hat{h} and We^* . \hat{h} denotes the dimensionless depth of the spark bubble, and We^* is a Weber number defined by the ratio of aerodynamic pressure to Laplace pressure.

indicates that the aerodynamic pressure is at least an order of magnitude greater than the Laplace pressure throughout the observed regime. Based on this scaling, we consider aerodynamic pressure to be the primary contributor in determining the IV/V transition, and surface tension is not required to explain the observed boundary.

Comment 6: To explain the boundary between Modes III.a and I, the authors discuss the influences of particle density and wettability. It is not clear why a horizontal line divides these two modes. Can the authors explain it more directly?

Thank you for your insightful feedback. Regarding the boundary between Modes I and III.a, we attempted an analysis and here we present our thinking.

In the phase diagram (Fig. 5 in the main text), the boundary between Modes I and III.a is a horizontal line at $\phi\hat{r}/\alpha \approx 0.15$, independent of the dimensionless spark bubble depth \hat{h} . These modes appear below the oblique line $t_c = 0.24$, meaning the sphere is fully submerged at $t = \tau$.

The vertical axis is $\phi\hat{r}/\alpha$, where $\alpha \sim 1 - (C_h + 4 \sin^2 \frac{\theta}{2} We^{-1})\rho^{-1}$. Here, C_h is the hydrodynamic force coefficient of order unity, r , θ and ρ are the sphere's radius, contact angle, and density ratio, respectively. For these two modes, with $We \sim 10^3 - 10^4$, the boundary simplifies to:

$$\frac{\phi\hat{r}}{1 - \rho^{-1}} \approx 0.15.$$

This result suggests that the sphere's physical properties, rather than \hat{h} , primarily influence the transition. We confirmed this by replotted the phase diagram in terms of the density ratio ρ and contact angle θ (**Supplementary Figure 9**), showing that denser and more wettable spheres tend towards Mode I, while less wettable and lower-density spheres are associated with Mode III.a.

Based on our observations in experiments, such as air venting and rapid channel formation, we hypothesize that the difference between Modes I and III.a lies in whether

a large air bubble forms and seals atop the sphere during the rapid immersion: Mode III.a features a sealed air bubble, whereas Mode I does not.

Consider two important mechanisms from water entry studies that could lead to sealed bubbles:

1. Wetting failure mechanism: When the liquid layer on the sphere moves faster than the maximum speed the contact line can sustain, wetting failure occurs[1, 5, 6]. This leads to the separation of the liquid layer from the sphere’s surface, allowing air to be entrained above the sphere.

2. Contact-line pinning: Following wetting failure, the dynamic contact line tends to pin above the sphere’s equator[7, 8]. The position of this pinning is influenced by the sphere’s wettability. With increasing contact angle and Weber number, the pinning position shifts towards the equator, ultimately leading to the sealing of a larger air bubble.

In addition to these mechanisms, the density of the sphere can also influence this transition. A lower-density sphere exhibits better flow-following behaviour, enhancing its movement through the liquid. This characteristic can impact both the dynamics of wetting failure and contact-line pinning. Furthermore, if an air bubble forms atop such a sphere, the enhanced movement can cause the bubble’s top layer to thin out. This thinning makes the bubble more susceptible to rupture, thus facilitating the venting of the cavity.

While these mechanisms provide a useful framework, they still cannot directly explain why the boundary is a horizontal line with respect to \hat{h} . More research is necessary to fully understand the transition mechanisms between these two modes.

Due to the constraints of current experimental techniques, strong light emission and surface deformation obscure measurements of the sphere and interface dynamics. These challenges limit our capacity to gather enough quantitative data for a detailed analysis. Consequently, the specific physical mechanisms behind the appearance of the horizontal line remain an open question. To address this, future work will involve numerical simulations to investigate the sphere’s and contact line’s dynamics in governing this particular boundary.

Comment 7: When it comes to the failure of $t_c = 0.24$ to describe the occurrence of Mode V for small particles, the authors attribute it to the influence of surface tension. I am wondering that if the viscosity plays a role here. Viscosity becomes important and inhibits the jetting significantly at small length scales with a large Ohnesorge number as reported for bubble bursting jets. Besides, I suggest the authors to prove this argument by presenting the experimental images that indicate the cases of Mode V below $t_c = 0.24$ are from branch 4 in Fig. 4.

Thank you for raising this point. To evaluate whether viscosity plays a role in suppressing singular jet formation for small particles ($r = 100 - 175 \mu\text{m}$, $1.9 < \hat{h} < 2.8$), we analyse two dimensionless numbers:

Supplementary Figure 9: Phase diagram showing Modes I (grey squares) and III.a (blue triangles) plotted against the sphere's density ratio ρ and contact angle θ .

1. Ohnesorge number:

$$Oh = \frac{\mu}{\sqrt{\rho_l \sigma r}} \approx 0.01,$$

where ρ_l , μ and σ are the liquid density, viscosity and surface tension, respectively. We use the sphere radius r as the characteristic length - consistent with bubble-bursting literature, which typically uses bubble radius rather than jet radius as the reference scale[9–11]. Previous studies on bubble-bursting jets[10, 11] report that jet formation is strongly affected by viscosity when $Oh > 0.037$, and the jet formation being completely inhibited when $Oh = 0.1$, clearly above the range in our measurements. This quantitative assessment confirms that viscosity plays a minor role in singular jet dynamics at these length scales.

2. Capillary number:

$$Ca = \frac{\mu V_{jet}}{\sigma} \approx 0.01,$$

where V_{jet} is the singular jet velocity. According to experimental observations, $V_{jet} = O(1)$ m/s at $\hat{h} \approx 1.7$, and we adopt an upper bound of $V_{jet} = 1$ m/s to estimate the Ca number. The small value of $Ca \ll 1$ also indicates that viscous force is negligible compared to surface tension.

These two dimensionless numbers confirm that viscous effects are not significant in this regime. Instead, we attribute the suppression of singular jet formation in these small-particle cases to surface tension.

We have provided **Supplementary Figure 10** to further verify that these Mode V cases below the $t_c^* = 0.24$ threshold originate from branch 4 (where the sphere is fully submerged at $t = \tau$). As shown in these snapshots, a micro-droplet appears above the sphere at $t = T_1$, indicating that the sphere has been fully submerged by this time. Additionally, no observable singular jet forms before $t = T_1$, supporting the reason that surface tension suppresses jet formation at these small scales.

Supplementary Figure 10: Experimental observation of a micro-scale droplet above the sphere at $t = T_1$. Scale bars: 1 mm. (see full caption in Supplementary Information)

In the revision, we added these discussions in the section “Distribution of the five jet modes”.

References for Reviewer 3

1. Duez, C., Ybert, C., Clanet, C. & Bocquet, L. Making a splash with water repellency. *Nature Physics* **3**, 180–183 (2007).
2. Reynolds, O. An experimental investigation of the circumstances which determine whether the motion of water shall be direct or sinuous, and of the law of resistance in parallel channels. *Philosophical Transactions of the Royal Society of London* **174**, 935–982 (1883).
3. Schlichting, H. & Gersten, K. *Boundary-Layer Theory* 12–14 (Springer, Berlin, 2017).
4. Williamson, C. H. K. Vortex Dynamics in the Cylinder Wake. *Annual Review of Fluid Mechanics* **28**, 477–539 (1996).
5. Truscott, T. T., Epps, B. P. & Belden, J. Water Entry of Projectiles. *Annual Review of Fluid Mechanics* **46**, 355–378 (2014).
6. Speirs, N. B., Mansoor, M. M., Belden, J. & Truscott, T. T. Water entry of spheres with various contact angles. *Journal of Fluid Mechanics* **862** (2019).
7. Do-Quang, M. & Amberg, G. The splash of a solid sphere impacting on a liquid surface: Numerical simulation of the influence of wetting. *Physics of Fluids* **21** (2009).

8. Ding, H. *et al.* On the contact-line pinning in cavity formation during solid-liquid impact. *Journal of Fluid Mechanics* **783**, 504–525 (2015).
9. Lee, J. S. *et al.* Size limits the formation of liquid jets during bubble bursting. *Nature Communications* **2** (2011).
10. Krishnan, S., Hopfinger, E. J. & Puthenveetil, B. A. On the scaling of jetting from bubble collapse at a liquid surface. *Journal of Fluid Mechanics* **822**, 791–812 (2017).
11. Ji, B., Yang, Z. & Feng, J. Compound jetting from bubble bursting at an air-oil-water interface. *Nature Communications* **12**, 6305 (2021).

Authors' Response to Reviewer 6

General Comments: The authors have adequately addressed Reviewer #4's concerns regarding the physical mechanisms. They have incorporated substantial quantitative analysis to clarify the physical picture of the jetting and the associated particle motion. Additionally, the critical condition proposed for distinguishing different jet modes is well-reasoned and supported by detailed explanations and literatures. However, I can only recommend publication given the following issues are appropriately addressed.

Thank you for your thoughtful review and positive feedback on our revised manuscript. Your constructive comments have been very helpful in further improving the clarity and quality of the manuscript.

Below, we provide detailed responses to your comments.

Comment 1: Figure 3(a-e) contains curves with seven different line styles, making these panels difficult to read clearly. Additionally, the data presented in Fig. 3(a-e) are not referenced in the main texts, and the essential information seem to be adequately conveyed by Fig. 3(f-h) already. I suggest the authors consider simplifying or removing Fig. 3(a-e) to enhance readability and clarity.

Thank you for this helpful suggestion. Figures 3a-e originally illustrated the kinematics of the featured geometries in the five jet modes. We agree that these figures contain some overlapping information with that in Figs. 3f-h.

In the revision, we have relocated Figs. 3a-e to the Supplementary Information, where they are now labelled as Supplementary Figures 1-5. In addition to the original quantitative measurements, these supplementary figures include corresponding experimental snapshots, which facilitate clearer interpretation of the measured results.

We hope this modification improves the clarity of the main text while preserving the full dataset for interested readers.

Comment 2: The term r_{rim} typically denotes the radius of the frontier of liquid domain, as shown in Figure 2 in Reference [50]. However, in the current manuscript, r_{rim} appears to represent the distance between the rim and the center of the cavity. This discrepancy may lead to confusion, as it deviates from the established convention. I suggest the authors clarify or reconcile this difference.

Thank you for your advice. To avoid confusion, the term r_{rim} has been renamed as b to represent the distance from the cavity rim to the cavity axis, as illustrated in **Figure for Comment 2**. This updated notation has been consistently applied throughout the revised manuscript to ensure clarity and avoid potential misunderstanding.

Comment 3: Reference [43] considers the contribution of surface tension in the Rayleigh-Plesset equation. Can the authors explain the reason why you neglected this term in the current analysis? Any consideration of Weber numbers here?

Figure for Comment 2: A sketch for the cavity. The grey circle denotes the solid sphere. The black curve denotes the air-water interface. The radius of the rim itself is a , and its distance to the cavity axis is b .

Thank you for highlighting the need for clearer reasoning regarding the boundary between Modes IV and V, specifically concerning the role of surface tension.

We evaluate the aerodynamic pressure drop across the cavity rim, Δp , with the Laplace pressure p_s induced by surface tension. According to Bernoulli’s principle, $\Delta p \sim \rho_a V_a^2$, where ρ_a is the air density and V_a is the velocity of airflow entering the cavity. The Laplace pressure is given by $p_s = \sigma(\frac{1}{a} - \frac{1}{b})$, where a is the radius of the rim itself and b is its distance to the cavity axis (see **Figure for Comment 2**). Based on the geometric characteristics of the liquid rim in Modes IV and V, both a and b are on the order of r , leading to $p_s \sim \sigma/r$.

To quantify the competition between aerodynamic and surface tension effects, we introduce a local Weber number:

$$We^* = \frac{\Delta p}{p_s} \sim \frac{\rho_a V_a^2 r}{\sigma}. \quad (4)$$

The estimation of We^* is detailed in the Methods section “Aerodynamic pressure vs. Laplace pressure in Modes IV and V”. As shown in **Supplementary Figure 7**, We^* ranges from $O(10^1)$ to $O(10^5)$ at the boundary between Modes IV and V. This indicates that the aerodynamic pressure is at least an order of magnitude greater than the Laplace pressure throughout the observed regime. Based on this scaling, we consider aerodynamic pressure to be the primary contributor in determining the IV/V transition, and surface tension is not required to explain the observed boundary.

We hope this modification clarifies why surface tension was omitted in this part of the analysis.

Comment 4: The authors state that the critical criterion for distinguishing between Mode I and Mode II is whether the bottom of the cavity contacts the particle at $t = T_1$. However, based on the provided images and videos, it is difficult to clearly observe or confirm this contact condition in Mode I. Can the authors clarify or provide additional evidence to support this claim?

Thank you for your valuable feedback. To address your concern, we have added **Supplementary Figure 8**, which includes both global views and close-up snapshots for representative cases of Modes I and II. These images capture both the jetting behaviour above the free surface and the sphere below it.

Supplementary Figure 7: Phase diagram of Modes IV and V plotted against \hat{h} and We^* . \hat{h} denotes the dimensionless depth of the spark bubble, and We^* is a Weber number defined by the ratio of aerodynamic pressure to Laplace pressure.

In **Supplementary Figure 8a** (Mode I), the arrows indicate that the concavity remains above the sphere and does not contact it at $t = T_1$. In contrast, in **Supplementary Figure 8b** (Mode II), the concavity reaches and contacts the sphere prior to $t = T_1$, as highlighted by the arrows.

We hope these additional images provide direct experimental confirmation of the criterion used to distinguish between Modes I and II.

Supplementary Figure 8: Experimental snapshots of (a) Mode I and (b) Mode II, including global views, close-ups above the water surface, and underwater close-ups. Arrows indicate that in Mode I, the concavity does not reach the sphere at $t = T_1$, while in Mode II, it contacts the sphere before $t = T_1$. (see full caption in Supplementary Information)

Comment 5: Minor remark: Please label or annotate the orange and blue circles in the panel of Figure 4 for hydrophobic/hydrophilic particles. Currently, it is difficult for

readers to understand what these colors represent without referring to the figure caption or accompanying explanation.

Thank you for your helpful suggestion. We have revised Fig. 4 in the main text by adding annotations to the blue and orange circles to indicate hydrophilic and hydrophobic spheres, respectively.

Responses to Reviewers' Comments for Manuscript NCOMMS-24-09227C

Particulate Reshapes Surface Jet Dynamics Induced by a Cavitation Bubble

Addressed Comments for Manuscript to

by

Xianggang Cheng, Xiao-Peng Chen, Zhi-Ming Yuan and Laibing Jia

Dear Editor and Reviewers,

We sincerely thank you for your careful review and constructive comments. Your feedback has been invaluable in improving the clarity and scientific rigour of the manuscript. In the revised version, we have made the following changes in direct response to your comments:

The classification criteria for Modes I and II have been clarified, and Supplementary Figure 8 has been updated with additional images that show the differences between the two modes at $t = \tau$ and $T_1 + T_2$.

In the following pages, we provide point-by-point responses to all issues raised. We hope the revisions have satisfactorily addressed all concerns and improved the manuscript accordingly.

Thank you again for your time and consideration.

Sincerely,

Laibing Jia (on behalf of all authors)

Authors' Response to Reviewer 3

Reviewer Comments: The authors have carefully revised the manuscript in response to my previous concerns and I am happy to recommend publication now.

Thank you for your positive feedback and recommendation. We are pleased that the revisions have addressed your concerns. We appreciate for your thorough review and valuable suggestions, which have significantly improved the quality of our manuscript.

Authors' Response to Reviewer 6

Reviewer Comments: I thank the authors addressing my concerns, and I appreciate the authors' clarification regarding the relative magnitudes of aerodynamic pressure and Laplace pressure. The provided analysis convincingly demonstrates that the surface tension can be neglected in the current modelling framework. However, I still have one problem regarding the supplementary Figure 8. Based on the jet morphology observed during the first period (i.e., prior to the time T_1) the two sets of results presented in Supplementary Figure 8 appear to represent an identical jet mode. However, a discrepancy arises when examining Figure 4 in the main manuscript: the distinction between Mode I and Mode II seems to originate before the cavitation bubble collapses during the first period. This observation raises concerns about the consistency between Figure 4 and Supplementary Figure 8. I kindly request the authors to clarify or address this apparent inconsistency.

Thank you for your helpful comment and for highlighting the apparent inconsistency between Figure 4 and Supplementary Figure 8.

Figure 4 in the manuscript summarises typical jet-evolution patterns based on a large set of experiments. The first branching criterion is whether a singular jet forms. Within the lower branch, differences appear before the first collapse of the spark bubble. These include the height of the water bulge and whether a large air bubble is sealed above the sphere. These early differences subsequently influence the second branching into Modes I, II, and III.a.

To clearly distinguish Modes I and II, we summarise their major observable differences, D1: $t \leq \tau$, height of the water bulge above the initial waterline. Mode I (high), Mode II (low).

D2: $t = T_1$, surface concavity contacts the sphere. Mode I (no), Mode II (yes).

D3: $t = T_1 + T_2$, crown structure appears. Mode I (present), Mode II (absent).

Among these differences, D1 lacks a sharp threshold that separates these two modes. D2 is a yes-or-no event that provides reliable identification and directly captures the particle-surface interaction. Although D3 is also a yes-or-no event, a crown can form even without particles, making it less particle-specific. For these reasons, we adopted D2 as the sole criterion for distinguishing Modes I and II.

In the previous version of Supplementary Figure 8, we intentionally selected conditions close to the mode-transition threshold, with the same sphere properties and similar spark bubble depths, aiming to highlight the key difference D2 (concavity-sphere contact). Because the chosen parameters were close, the visual differences were subtle, leading to confusion, for which we apologise. More pronounced visual differences between Modes I and II, particularly at early stages, can be found in Figure 2 of the main manuscript, as well as in Supplementary Figures 1 & 2, and Supplementary Videos 1 & 2.

In the revised Supplementary Figure 8, we retain the original experimental images for clarity but now include additional images that show differences D1 and D3. It should be

Supplementary Figure 8: Experimental snapshots of Mode I and Mode II, including global views, close-ups above the water surface, and underwater close-ups. Three observable differences (D1-D3) between the two modes are shown. D1: Height of the water bulge above the initial waterline at $t = \tau$. Mode I (high), Mode II (low); D2: Surface concavity contacts the sphere at $t = T_1$. Mode I (no), Mode II (yes); D3: Crown structure appears at $t = T_1 + T_2$. Mode I (present), Mode II (absent). Among these, D2 is adopted as the sole classification criterion, as it provides a clear binary event directly reflecting the particle-surface interaction. The dimensionless depth of the spark bubble is $\hat{h} = 1.11$ for Mode I and $\hat{h} = 1.35$ for Mode II. Sphere properties: $r = 500 \mu\text{m}$, $\rho = 4.4$, $\theta = 80.8^\circ$. Scale bars: 5 mm.

noted, however, that difference D1 remains visually subtle due to similar experimental conditions. For difference D3, an early-stage crown structure appears in Mode I but is absent in Mode II.

We hope that the above explanations, together with the revised Supplementary Figure 8, address your concern and clarify its consistency with the evolution paths summarised in Figure 4.

Responses to Reviewers' Comments for Manuscript NCOMMS-24-09227D

Particulate Reshapes Surface Jet Dynamics Induced by a Cavitation Bubble

Addressed Comments for Manuscript to

by

Xianggang Cheng, Xiao-Peng Chen, Zhi-Ming Yuan and Laibing Jia

Authors' Response to Reviewer 6

Reviewer Comments: The authors have addressed my concerns clearly and provided sufficient explanation for the key differences between Modes I and II. The revisions and the additional images make the manuscript clear and complete. I recommend the manuscript for publication.

Thank you for your positive feedback and recommendation. We are pleased that the revisions addressed your concerns. We are grateful for your thorough review and valuable suggestions, which significantly improved the quality of our manuscript.

The experimental part of the paper was nicely done. However the manuscript can be further improved. Here are my queries:

I may have missed out. Can you please specify the physical parameters which affect the five different jet modes?

In general γ is used to denote surface tension. It will be better to use another symbol for the non-dimensional height.

Page 2 line 77- spheres in the bracket may be omitted.

Page 3 line 136- How are the sphere surfaces transformed to superhydrophobic surfaces?

Page 4 line 178-179: The deviation from the particulate free condition should be described briefly. Currently the “significant departure ” is unable to emphasize the differences. Elaboration of the deviations will be helpful for the reader.

Pages 183- Please elaborate in two-three lines about the “ventilation’ process or at least refer figure 3.

Figure 2- There seems to be no discussion on the effect of the contact angle of the particulate with the water. Please report if you have noticed any variation due to contact angle (in line 135 to 136 you have mentioned about the contact angles).

Line 222 and 226- “substantially lowers” & “significantly reduce”- Please provide quantitative estimate and compare with particle free condition to emphasize your claims.

Page 271-273: “the importance ofroperties, surface dynamics, and bubble strength.”- Please elaborate how the chasing time is correlated with the given properties.

Page 275-276 : Subsequently you have bypassed the elaboration by mentioning “However, the phase plot only provides an initial understanding, and a deeper analysis is needed to fully grasp their combined effects.” For high impact journals like Nature Communications I think the deeper analysis is needed.

Report to Manuscript NCOMMS-24-09227A entitled “Particulate Reshapes Surface Jet Dynamics Induced by a Cavitation Bubble” by Cheng et al.

The authors have addressed some of my concerns carefully, but the revised manuscript still lacks enough physical insights and some of the physical arguments seems unreasonable.

Regarding their response to my previous comments, the dimensionless chasing number t_c is not a parameter that can be easily obtained with initial parameters of the system – they need to measure the particle immersed depth h_{im} , which is actually a function of the parameters of particle and liquid. Besides, I cannot understand the physical reason for the relation of $V_s = \rho^{-1/2}V_l$. The motion of the particle is set by the acting forces including surface tension, drag force, and added mass force, etc., which cannot be described by $V_s = \rho^{-1/2}V_l$. What is the physical picture of particle immersion? It may be resulted by the movement of contact line (when the contact line moves to the top of the particle) or the pinch off of the liquid surface above the particle, while the authors only consider the difference between the velocities of liquid surface and particle. This should be clearly demonstrated by solid physical arguments.

In addition, many arguments for the derivation of t_c can be validated by experiments, for example the velocities of particle and liquid surface are measurable, as well as the characteristic times for maximum radius and particle immersion. These measurements will well guide and support the interpretations of the jetting behavior.

My second concern is about the interpretation of another dimensionless number \hat{h} . The radiated energy density $e \sim E/(4\pi h^2) \sim (p_{atm} - p_v)R_m\hat{h}^{-2}$, which means that \hat{h} cannot solely represent the radiated energy density (actually e is a dimensional parameter while \hat{h} is dimensionless). Using radiated energy to explain why \hat{h} divides different modes seems not reasonable. Thus, the physical meaning of the proposed dimensionless numbers need to be reexamined carefully.

My third concern is about the proposed criteria for different jet modes. Besides the physical meaning of two dimensionless numbers, the critical values of the dimensionless numbers for different modes lack physical reasons. Why $t_c = 0.2$ separates modes II and IV? Why $\hat{h} = 2.1$ marks the boundary separating mode V from mode IV? The same for the critical \hat{h} for the boundaries between modes I, II, and V. Fourthly, as the author stated, they focus on the initial branching of modes in Fig. 4

during the bubble's initial expanding stage when proposing t_c , then t_c should describe the boundary between modes I, II and modes IV, V. However, the regime map in Fig. 5a does not support this.

Actually, the boundaries between different modes in Fig. 5a are blur and we can use different lines to describe them. For example, the boundaries between III and IV, as well as between IV and V, can also be roughly described by two oblique lines perpendicular to the current oblique line. The boundaries between I and II, as well as between II and V, show a "V" shape.

Additionally, I also cannot understand the interpretation in the paragraph before section "Conclusion and Discussion". Why "the omission of surface tension effects for small-radii spheres underestimates their velocities"? The small radii spheres correspond to a region below an oblique line instead of the horizontal line $t_c = 0.2$.

From the supplementary videos, one can find that the local dynamics around the particle matters for the jet dynamics. Thus I suggest the authors to look into the detail on the dynamics between the particle and liquid surface, either experimentally or theoretically, which may be a feasible route for convincing physical explanations.

The authors have adequately addressed Reviewer #4's concerns regarding the physical mechanisms. They have incorporated substantial quantitative analysis to clarify the physical picture of the jetting and the associated particle motion. Additionally, the critical condition proposed for distinguishing different jet modes is well-reasoned and supported by detailed explanations and literatures. However, I can only recommend publication given the following issues are appropriately addressed.

1. Figure 3(a–e) contains curves with seven different line styles, making these panels difficult to read clearly. Additionally, the data presented in Fig. 3(a–e) are not referenced in the main texts, and the essential information seem to be adequately conveyed by Fig. 3(f–h) already. I suggest the authors consider simplifying or removing Fig. 3(a–e) to enhance readability and clarity.
2. The term r_{rim} typically denotes the radius of the frontier of liquid domain, as shown in Figure 2 in Reference [50]. However, in the current manuscript, r_{rim} appears to represent the distance between the rim and the center of the cavity. This discrepancy may lead to confusion, as it deviates from the established convention. I suggest the authors clarify or reconcile this difference.
3. Reference [43] considers the contribution of surface tension in the Rayleigh-Plesset equation. Can the authors explain the reason why you neglected this term in the current analysis? Any consideration of Weber numbers here?
4. The authors state that the critical criterion for distinguishing between Mode I and Mode II is whether the bottom of the cavity contacts the particle at $t = T_1$. However, based on the provided images and videos, it is difficult to clearly observe or confirm this contact condition in Mode I. Can the authors clarify or provide additional evidence to support this claim?
5. Minor remark: Please label or annotate the orange and blue circles in the panel of Figure 4 for hydrophobic/hydrophilic particles. Currently, it is difficult for readers to understand what these colors represent without referring to the figure caption or accompanying explanation.